# Analysis of Natural Actor-Critic with Randomized Low-Discrepancy Sampling

## Abstract

Natural gradient methods are appealing in policy optimization due to their invariance to smooth reparameterization and their ability to account for the local geometry of the policy manifold. These properties often lead to improved conditioning of the optimization problem compared to Euclidean policy gradients. However, their reliance on Monte Carlo estimation introduces high variance and sensitivity to hyperparameters. In this paper, we address these limitations by integrating Randomized Quasi-Monte Carlo (RQMC) sampling into the natural actor-critic (NAC) framework. We revisit the NAC linear system and show that, under imperfect value approximation, the NAC update decomposes exactly into the true natural gradient plus a Fisher-metric projection of the Bellman residual onto the score-feature span. We further develop RQMC-based NAC estimators that replace IID sampling with randomized low-discrepancy trajectories. We provide a variance analysis showing that the RQMC-based estimators reduce estimator variance under regularity conditions, thereby reducing the propagation of Bellman-residual error into the natural-gradient update. Empirical results on certain reinforcement learning benchmarks demonstrate that our RQMC-enhanced algorithms consistently match or improve upon the performance and stability of their vanilla counterparts.

## 1 Introduction

Reinforcement learning (RL) finds optimal solutions to sequential decision-making problems, where an agent seeks a set of decision rules that maximize cumulative reward through repeated interaction with an environment (Sutton et al., 1998; Bertsekas, 2025). Among the spectrum of RL algorithms, policy-gradient actor–critic methods are appealing due to their compatibility with continuous action spaces, their use of flexible function approximators, and their applicability to high-dimensional nonlinear systems (Sutton et al., 1999; Peters et al., 2005). The policy-gradient and actor–critic methods (Williams, 1992; Sutton et al., 1999; Konda & Tsitsiklis, 1999; Konda & Borkar, 1999; Baxter & Bartlett, 2001) avoid explicit computation of exact value functions, instead calibrate the policy parameters directly through averaged or bootstrapped gradient estimates. However, standard Euclidean gradients can become trapped in plateau regions of the objective landscape, where their magnitude is extremely small and provides little directional guidance for further improvement (Kakade, 2001). Moreover, when optimizing over parameterized probability distributions, Euclidean distance between parameter vectors does not necessarily correspond to similarity between the induced policies.

Natural gradient methods (Amari, 1998; Martens, 2020) and their reinforcement-learning variant, the natural policy gradient (NPG) (Kakade, 2001; Peters et al., 2005; Khodadadian et al., 2021), address these limitations by replacing Euclidean geometry with the Fisher–Rao information metric, the canonical Riemannian metric on statistical manifolds. Under this geometry, which is equivalently induced by the Kullback–Leibler (KL) divergence, the steepest-descent direction is not given by the ordinary gradient rather by the natural gradient (Amari, 1998; Martens & Grosse, 2015; Martens, 2020), which preconditions updates using the Fisher information matrix. The resulting updates correspond to meaningful changes in policy behavior, are invariant to reparameterization, and tend to be more stable when optimizing stochastic policies. Lin-

ear convergence of the natural policy gradient has been proven for tabular and softmax parameterizations (Khodadadian et al., 2021), confirming its geometric efficiency.

Classical natural gradient methods, though, provide invariant, geometry-aware updates, they do not specify how to get a good estimate of that update from the data. Monte Carlo (MC) method (Metropolis & Ulam, 1949; Mohamed et al., 2020) has been the most common method for approximating policy gradient and natural policy gradient. Despite their widespread use and asymptotic convergence guarantees, Monte Carlo methods can suffer from slow convergence rates $O(N^{-1/2})$ and high variance, particularly in high-dimensional problems using fewer sample set. Variance reduction for policy gradients has been extensively studied through baseline subtraction (Weaver & Tao, 2001), control variates (Grathwohl et al., 2018; Tucker et al., 2017), generalized advantage estimation (Schulman et al., 2016), importance-sampling corrections (Precup, 2000; Jiang & Li, 2016), and recursive variance reduction methods (Xu et al., 2020a). Analyses of REINFORCE have explored its sample-efficiency properties under Monte Carlo rollouts (Zhang et al., 2021), yet such methods remain fundamentally tied to independent trajectory sampling and therefore do not address the variance arising from the trajectory-generation process itself. While recent works have provided non-asymptotic convergence rates for natural actor-critic (Xu et al., 2020b), the performance of these algorithms in practice remains heavily influenced by the variance of their gradient estimates.

Quasi-Monte Carlo methods (QMC) offer a solution to this problem by replacing the random sampling of Monte Carlo with a more deterministic low-discrepancy point set (Niederreiter, 1978; Papageorgiou, 2003; Faure, 1982; Halton, 1960) such as Sobol, Halton Faure and Niederreiter sequences and provides an efficient approach to numerical integration (Halton, 1960), simulation and optimization. QMC is shown to obtain $O(1/N)$ rate of convergence (Papageorgiou, 2003) for integrands with bounded variation in the sense of Hardy and Krause and lower effective dimension (Owen, 1997b). Moreover, when multiple independent samples are needed, one can introduce random perturbations to a QMC point set and get a set of new random points in a way that preserves the low-discrepancy nature (Owen, 1995; L'Ecuyer, 2018). This new point set is referred to as the randomized quasi-Monte Carlo (RQMC) (Owen, 1995) point set. The efficiency of RQMC sampling for policy evaluation and policy learning has been empirically studied in (Arnold et al., 2022), where RQMC sampling is used as a drop-in replacement of Monte Carlo sampling, in vanilla policy gradient algorithm and soft actor-critic algorithm (Haarnoja et al., 2018). In (Arnold et al., 2022) it is shown that by generating trajectories from actions sampled using RQMC point sets, improved estimates of policy gradient can be achieved while reducing variance in policy value estimates. Furthermore, this enhancement can be accomplished with fewer samples. Array-RQMC (L'Ecuyer et al., 2008) applies RQMC point sets to simulate finite parallel realizations of a Markov chain by sampling next states directly from the state space, rather than actions from the action space. Each RQMC point has dimensionality matching the state space, enabling coordinated, low-discrepancy sampling across trajectories. This leads to substantial variance reduction due to the induced stratification and negative dependence across state transitions (Puchhammer et al., 2021; Abdellah et al., 2019).

## 1.1 Our Contribution

In this paper, we provide a theoretical analysis of the Natural Actor–Critic framework with respect to stochastic estimation error on natural-gradient updates. We revisit NAC from the perspective of the compatible-feature solution and show that, when the critic only approximately satisfies the Bellman equation, the resulting update admits a clear structural decomposition: the estimate produced by the critic consists of the true natural gradient together with a Fisher-metric projection of the Bellman residual onto the score feature space. We further show how randomness in this solution propagates through Fisher preconditioning, directly influencing update magnitude, conditioning, and stability. Prior analyses of bias in natural actor–critic methods (Thomas, 2014; Wen et al., 2021) only examine discounting mismatch or the generic gap between actor–critic and policy-gradient updates, but do not provide this projection-based characterization specific to natural actor–critic with compatible features. Our analysis makes explicit that the behaviour of NAC is not determined solely by information geometry, but also by the variance properties of the estimator used to approximate the natural gradient. Based on this perspective, we propose a variance-reduced natural-gradient estimator based on low-discrepancy sampling. By incorporating RQMC constructions into the estimation of the compatible-feature solution, we obtain an unbiased estimator whose dispersion is prov-

ably reduced relative to standard Monte Carlo sampling. Although RQMC methods have been applied to reinforcement learning (Arnold et al., 2022), specifically for generic policy-gradient and standard actor–critic algorithms and their use has not been examined within the natural actor–critic framework. Ours is the first to integrate Array-RQMC sampling directly into the NAC linear system and to establish that randomized low-discrepancy arrays yield improved unbiased estimators of NAC components.

## 2 Background

In this paper, we consider the standard reinforcement learning (RL) framework (Sutton et al., 1998; Puterman, 2014), where the underlying system is modeled as a discrete-time Markov decision process (MDP). The latent MDP consists of a finite set of states $\mathcal{S}$ and actions $\mathcal{A}$, the values of which at each time step $t \in \{0, 1, 2...\}$ are represented as $\mathbf{s}_t \in \mathcal{S}$ and $\mathbf{a}_t \in \mathcal{A}$. The dynamics of the underlying system are modeled using state transition probabilities $\mathbb{P}(\mathbf{s}_{t+1} = s' \,|\, \mathbf{s}_t = s, \mathbf{a}_t = a)$ and the state transition reward $r(\mathbf{s}_t, \mathbf{a}_t) \in \mathbb{R}$, $\forall s, s' \in \mathcal{S}, a \in \mathcal{A}, t \in \{0, 1, 2....\}$. The agent follows a stationary stochastic policy $\pi_\theta(\cdot|s)$, parameterized by $\theta \in \mathbb{R}^q$, which defines a distribution over actions given state $s$. In general, the goal of RL is to find a policy that optimizes the cost function $J(\theta)$, which is a measure of the performance of the policy $\pi_\theta$. The natural actor-critic algorithm (Peters et al., 2005) optimizes the cost function based on the mean cumulative $\gamma$-discounted rewards, where $\gamma \in [0, 1)$ serves as the discount factor. The objective is given by:

$$J_\gamma(\theta) = \mathbb{E}_{\mathbf{a}_t \sim \pi_\theta(\cdot|\mathbf{s}_t)} \left[ \sum_{t=0}^{\tau-1} \mathbf{r}_t \,\big|\, \mathbf{s}_0 \sim \mu \right], \tag{1}$$

where $\mu$ is the initial distribution and $\tau$ be a geometric stopping time, independent of the trajectory, with tail distribution $\mathbb{P}(\tau > t) = \gamma^t$. By independence of $\tau$ and the trajectory, and by Tonelli's theorem, $\mathbb{E}_{\pi_\theta}\left[\sum_{t=0}^{\tau-1} \mathbf{r}_t \,|\, \mathbf{s}_0 \sim \mu\right] = \sum_{t=0}^{\infty} \mathbb{P}(\tau > t)\, \mathbb{E}_\theta[\mathbf{r}_t \,|\, \mathbf{s}_0 \sim \mu] = \sum_{t=0}^{\infty} \gamma^t\, \mathbb{E}_{\pi_\theta}[\mathbf{r}_t \,|\, \mathbf{s}_0 \sim \mu]$. Hence,

$$J_\gamma(\theta) = \sum_{t=0}^{\infty} \gamma^t\, \mathbb{E}_{\pi_\theta}[\mathbf{r}_t \mid \mathbf{s}_0 \sim \mu] = \sum_{t=0}^{\infty} \gamma^t \sum_{s \in \mathcal{S}} \mathbb{P}_{\pi_\theta}(\mathbf{s}_t = s \mid \mathbf{s}_0 \sim \mu) \sum_{a \in \mathcal{A}} \pi_\theta(a|s) r(s, a)$$

$$= \sum_{s \in \mathcal{S}} \left[ \sum_{t=0}^{\infty} \gamma^t\, \mathbb{P}_{\pi_\theta}(\mathbf{s}_t = s \mid \mathbf{s}_0 \sim \mu) \right] \sum_{a \in \mathcal{A}} \pi_\theta(a|s) r(s, a) = \frac{1}{1-\gamma} \sum_{s \in \mathcal{S}} d_{\gamma,\mu}^{\pi_\theta}(s) \sum_{a \in \mathcal{A}} \pi_\theta(a|s) r(s, a), \tag{2}$$

where $d_{\gamma,\mu}^{\pi_\theta}(s) = (1-\gamma) \sum_{t=0}^{\infty} \gamma^t \mathbb{P}_{\pi_\theta}(\mathbf{s}_t = s | \mathbf{s}_0 \sim \mu)$ is the discounted stationary distribution of state $s$ under the policy $\pi_\theta$. The value function $V^\pi(s)$ is defined as $V_\gamma^{\pi_\theta}(s) = \sum_{a \in \mathcal{A}} \pi_\theta(a|s) Q_\gamma^{\pi_\theta}(s, a)$, where

$$Q_\gamma^{\pi_\theta}(s, a) \;=\; \mathbb{E}_{\pi_\theta} \left[ \sum_{t=0}^{\infty} \gamma^t \mathbf{r}_t \mid \mathbf{s}_0 = s, \mathbf{a}_0 = a \right]. \tag{3}$$

The gradient of the cost function $w.r.t.$ the policy parameters $\theta$ is as follows (Sutton et al., 1999),

$$\nabla_\theta J(\theta) = \sum_{s \in \mathcal{S}} d_{\gamma,\mu}^{\pi_\theta}(s) \sum_{a \in \mathcal{A}} \nabla_\theta \pi_\theta(a|s) Q_\gamma^{\pi_\theta}(s, a), \tag{4}$$

We can also subtract a baseline function for the reduction in variance since a baseline function always has a mean zero with respect to the policy. The gradient of the cost function then becomes, $\nabla_\theta J(\theta) = \sum_s d_{\gamma,\mu}^{\pi_\theta}(s) \sum_a \nabla_\theta \pi_\theta(a|s)(Q_\gamma^{\pi_\theta}(s, a) - b(s))$. Since $\gamma$ and $\mu$ are fixed, we let $d^{\pi_\theta} = d_{\gamma,\mu}^{\pi_\theta}$, $Q^{\pi_\theta} = Q_\gamma^{\pi_\theta}$ and $V^{\pi_\theta} = Q_\gamma^{\pi_\theta}$.

In (Amari, 1998), it is shown that when optimizing a function in a Riemannian space, the natural gradient gives the direction of the steepest descent. This is applicable in reinforcement learning because the cost function $J(\theta)$ can be viewed as a function over the set of probability distributions $\{\pi_\theta, \theta \in \mathbb{R}^q\}$ such that each state $s$ is associated with a probability manifold that has a Riemannian structure. This manifold captures the intrinsic geometry of the policy space, accounting for the fact that small changes in parameters can have different effects depending on the current policy. Also, two parameter vectors that induce the same

probability distribution over the trajectory space should be considered the same point, and the distance between two policies should be measured by how different their probability distributions are, not by the Euclidean distance between their parameters. The natural gradient update can be seen as a first-order update that preconditions the gradient with the Fisher information matrix, which accounts for the curvature of the policy space (Kakade, 2001) and is defined as follows:

$$\nabla_G J(\theta) = G(\theta)^{-1} \nabla J(\theta) \tag{5}$$

where $G$ is the Fisher information matrix associated with the probability distribution $\pi_\theta(a|s)$,

$$G(\theta) = \mathbb{E}_{\substack{s \sim d^{\pi_\theta} \\ a \sim \pi_\theta(\cdot|s)}} \left[ (\nabla_\theta \log \pi_\theta(\mathbf{a}|\mathbf{s}))(\nabla_\theta \log \pi_\theta(\mathbf{a}|\mathbf{s}))^\top \right] \tag{6}$$

The natural gradient preconditions the update with $G(\theta)^{-1}$, which accounts for the intrinsic geometry of the parameter manifold. This ensures that a unit step in the gradient direction corresponds to a unit of change in the model's predictions, not its parameters, leading to more stable and efficient convergence. In (Kakade, 2001), it is shown that if $f_w(s, a)$ is a function used to approximate the state-action value $Q^\pi(s, a)$, such that $f_w(s, a)$ meets the criteria posed in (Sutton et al., 1999), then the parameters for the approximation, gives the natural gradient of the cost function, *i.e.*, $w = G(\theta)^{-1} \nabla J(\theta)$. In (Peters et al., 2005), actor-critic algorithms(Konda & Tsitsiklis, 1999) were introduced, integrating natural policy gradient with linear function approximation within an LSTD-$\lambda$ framework (Boyan, 1999) to estimate the value of $w$. Subsequently, (Bhatnagar et al., 2007) extended these methods by proposing fully incremental algorithms that estimate the inverse Fisher information matrix and the natural gradient in an online setting. It is important to note that the geometry of the policy manifold is locally captured by $G(\theta)$, and any noise in its estimate can lead to biased or unstable natural gradient directions.

## 3 Natural Actor Critic

Here, we establish a rigorous analysis of the natural actor-critic algorithm in the discounted setting, under linear function approximation. Analytically, the natural gradient is defined through the Fisher metric, practical algorithms must estimate this direction from sampled trajectories, where the true advantage function is unknown. The natural actor–critic framework addresses this by employing linear function approximation, which enables the natural gradient to be recovered from a sample-based linear system. Specifically, we consider a linear function approximation for the advantage function: let $f_w(s, a) = \boldsymbol{\psi}(s, a)^T w$ be an approximation of the advantage function $A^{\pi_\theta}(s, a)$, where $w \in \mathbb{R}^q$ is a parameter vector and $\boldsymbol{\psi}(s, a) = \nabla_\theta \log \pi_\theta(a|s)$ is the state-action score (policy gradient) feature vector. Similarly, the value function $V^{\pi_\theta}$ is approximated by a linear function: $V^{\pi_\theta}(s) \approx \boldsymbol{\phi}(s)^\top v$, with state feature vector $\boldsymbol{\phi} : S \to \mathbb{R}^d$ and $v \in \mathbb{R}^d$. In this paper, we assume the following:

**Assumption 1.** For $\theta \in \mathbb{R}^q$, the Markov chain induced by $\pi_\theta$ is ergodic, *i.e.*, aperiodic, and irreducible.

**Assumption 2.** The features and rewards are bounded, *i.e.*, $\exists C_{\boldsymbol{\phi}}, C_{\boldsymbol{\psi}}, C_r > 0, s.t.,$

$$\|\boldsymbol{\phi}(s)\| \le C_{\boldsymbol{\phi}}, \quad \|\boldsymbol{\psi}(s, a)\| \le C_{\boldsymbol{\psi}}, \quad |r(s, a)| \le C_r, \forall s, a.$$

**Assumption 3.** The policy satisfies $\pi_\theta(a \mid s) > 0$ for all $s \in \mathcal{S}$, $a \in \mathcal{A}$, and $\theta \in \Theta$. Moreover, the policy $\pi_\theta(a|s)$ is twice continuously differentiable *w.r.t.* $\theta$ and there exist constants $M_{\nabla \pi}, M_\pi > 0$ satisfying

$$|\pi_\theta(a \mid s)| \le M_\pi, \qquad \|\nabla_\theta \pi_\theta(a \mid s)\| \le M_{\nabla \pi}, \qquad \forall s, a, \theta.$$

**Assumption 4.** The gradient of the cost objective $\nabla_\theta J(\theta)$ is Lipschitz continuous, *i.e.*,

$$\exists L_J > 0, \|\nabla_\theta J(\theta) - \nabla_\theta J(\theta')\| \le L_J \|\theta - \theta'\|, \forall \theta, \theta' \in \Theta.$$

**Assumption 5.** The Fisher information matrix, defined as $G(\theta) = \mathbb{E}_{s \sim d_\pi, a \sim \pi_\theta(\cdot|s)}[\boldsymbol{\psi}(s, a)\boldsymbol{\psi}(s, a)^\top]$, satisfies

(a) **Positive definiteness:** $\exists \lambda_{\min} > 0$ and $\lambda_{\max} < \infty$ such that $\lambda_{\min} I \preceq G(\theta) \preceq \lambda_{\max} I, \quad \forall \theta \in \Theta.$

(b) **Lipschitz continuity:** $\exists L_G > 0$ such that $\|G(\theta) - G(\theta')\| \le L_G \|\theta - \theta'\|, \forall \theta, \theta' \in \Theta$.

**Assumption 6.** For each $\theta \in \Theta$, the augmented feature vector $\widehat{\phi}(s,a) = [\psi(s,a)^\top, \phi(s)^\top]^\top$ is linearly independent $d^{\pi_\theta} \times \pi_\theta$-almost surely.

**Remark 1.** Assumption 2 trivially follows for finite state and action spaces. Further, under Assumptions 2 and 3, and with the state and action spaces finite, it follows that both the policy gradient $\nabla_\theta J(\theta)$ and natural gradient $\nabla_G J(\theta)$ are also bounded. Specifically,

1. The policy gradient satisfies $\|\nabla_\theta J(\theta)\| \le M_J := |\mathcal{A}| \cdot M_{\nabla \pi} \cdot \frac{C_r}{1-\gamma}$ for all $\theta \in \Theta$. This follows from policy gradient theorem and Assumptions 2 and 3, since

$$\|\nabla_\theta J(\theta)\| \le \sum_{s \in \mathcal{S}} d^{\pi_\theta}(s) \sum_{a \in \mathcal{A}} \|\nabla_\theta \pi_\theta(a|s)\| \cdot |Q^{\pi_\theta}(s,a)|$$

$$\le \sum_{s \in \mathcal{S}} d^{\pi_\theta}(s) \sum_{a \in \mathcal{A}} M_{\nabla \pi} \cdot \frac{C_r}{1-\gamma} = M_{\nabla \pi} |\mathcal{A}| \cdot \frac{C_r}{1-\gamma}.$$

2. The natural gradient satisfies $\|\nabla_G J(\theta)\| \le M_{\nabla_G} := M_J / \lambda_{\min}$ for all $\theta \in \Theta$. Indeed,

$$\|\nabla_G J(\theta)\| = \|G(\theta)^{-1} \nabla_\theta J(\theta)\| \le \|G(\theta)^{-1}\| \cdot \|\nabla_\theta J(\theta)\| \le \frac{M_J}{\lambda_{\min}}.$$

The last inequality follows from Part 1 and Assumption 5(a), $(G(\theta) \succeq \lambda_{\min} I$, so $\|G(\theta)^{-1}\| \le 1/\lambda_{\min})$.

To ensure that the curvature of the policy manifold does not distort gradients arbitrarily, we first establish that the natural-gradient operator varies smoothly with the parameters.

**Lemma 1.** *Under Assumptions 4 and 5, the natural gradient $\nabla_G J(\theta)$ is Lipschitz continuous, i.e., there exists $L_{\nabla_G} > 0$ such that*

$$\|\nabla_G J(\theta) - \nabla_G J(\theta')\| \le L_{\nabla_G} \|\theta - \theta'\|, \quad \forall \theta, \theta' \in \Theta,$$

*with Lipschitz constant $L_{\nabla_G} = \frac{L_J}{\lambda_{\min}} + \frac{L_G M_J}{\lambda_{\min}^2}$.*

*Proof.* Let $\theta, \theta' \in \Theta$. Then:

$$\|\nabla_G J(\theta) - \nabla_G J(\theta')\| = \|G(\theta)^{-1} \nabla_\theta J(\theta) - G(\theta')^{-1} \nabla_\theta J(\theta')\|$$

$$\le \|G(\theta)^{-1}(\nabla_\theta J(\theta) - \nabla_\theta J(\theta'))\| + \|(G(\theta)^{-1} - G(\theta')^{-1})\nabla_\theta J(\theta')\|$$

For the first term, using Assumption 4, we get

$$\|G(\theta)^{-1}(\nabla_\theta J(\theta) - \nabla_\theta J(\theta'))\| \le \frac{1}{\lambda_{\min}} L_J \|\theta - \theta'\| \tag{7}$$

For the second term, using the identity $G(\theta')^{-1} - G(\theta)^{-1} = G(\theta')^{-1}(G(\theta) - G(\theta'))G(\theta)^{-1}$:

$$\|(G(\theta)^{-1} - G(\theta')^{-1})\nabla_\theta J(\theta')\| = \|G(\theta)^{-1}(G(\theta') - G(\theta))G(\theta')^{-1}\nabla_\theta J(\theta')\|$$

$$\le \|G(\theta)^{-1}\| \cdot \|G(\theta') - G(\theta)\| \cdot \|G(\theta')^{-1}\nabla_\theta J(\theta')\|$$

$$\le \frac{1}{\lambda_{\min}} \cdot L_G \|\theta - \theta'\| \cdot \frac{M_J}{\lambda_{\min}}$$

$$= \frac{L_G M_J}{\lambda_{\min}^2} \|\theta - \theta'\| \tag{8}$$

The claim follows by combining (7) and (8). $\qquad\square$

The geometric interpretation of the natural gradient is rooted in the intimate connection between the Fisher information matrix and the local structure of the policy manifold. To make this relationship precise, we appeal to a classical result from information geometry (Amari, 1998): for any smooth parameterized family of distributions, the Kullback–Leibler divergence between a distribution and an infinitesimally perturbed version admits a second-order expansion governed entirely by the Fisher information matrix. In the RL setting, the object of interest is the policy-manifold-averaged KL divergence, obtained by integrating the per-state divergence against the discounted state visitation measure $d^{\pi_\theta}$,

$$\mathcal{D}_{\mathrm{KL}}(\pi_\theta \| \pi_{\theta+\delta}) = \mathbb{E}_{\mathbf{s} \sim d^{\pi_\theta}} \left[ \mathcal{D}_{\mathrm{KL}}(\pi_\theta(\cdot|\mathbf{s}) \| \pi_{\theta+\delta}(\cdot|\mathbf{s})) \right] = \frac{1}{2} \delta^\top G(\theta) \delta + o(\|\delta\|^2). \tag{9}$$

This expansion reveals that $G(\theta)$ constitutes the Hessian of the KL divergence in parameter space, averaged under the discounted stationary distribution, and therefore defines the local Riemannian metric on the manifold of stochastic policies (Amari, 1998; Martens, 2020). Two structural consequences are immediate. First, Euclidean distance between parameter vectors is not a faithful measure of dissimilarity between the policies they induce: a displacement of fixed Euclidean magnitude $\|\delta\|$ can correspond to an arbitrarily large or negligible change in policy behavior, depending on the local curvature captured by $G(\theta)$. This decoupling of parameter-space geometry from policy-space geometry is the fundamental source of ill-conditioning in ordinary gradient ascent on $J(\theta)$. Second, and more constructively, the expansion in (9) implies that constraining policy change in terms of KL divergence is equivalent, to second order, to constraining the $G(\theta)$-weighted norm of the update. Formally, as established by (Kakade, 2001) and made algorithmically explicit in the trust-region formulation of (Schulman et al., 2015), further developed in the context of natural actor–critic methods by (Peters et al., 2005) and (Martens, 2020), the solution to the constrained problem

$$\max_\delta \ \nabla_\theta J(\theta)^\top \delta \quad \text{subject to} \quad \mathcal{D}_{\mathrm{KL}}(\pi_\theta \| \pi_{\theta+\delta}) \le \epsilon \tag{10}$$

is locally approximated by the natural-gradient direction $\delta^* = \alpha G(\theta)^{-1} \nabla_\theta J(\theta)$, with the step size $\alpha$ absorbed into the constraint. This identifies the natural gradient as the local direction of steepest ascent on the Riemannian policy manifold, the unique direction along which a unit KL displacement yields the greatest increase in expected return. Because this direction is defined intrinsically through the manifold geometry rather than through the choice of parameterization, the resulting update is invariant to smooth reparameterization of the policy and exhibits superior conditioning relative to its Euclidean counterpart, particularly in flat regions of the objective landscape where ordinary gradients provide little directional guidance (Kakade, 2001; Khodadadian et al., 2021).

Critically, however, the foregoing is an analytic characterization: the natural gradient $G(\theta)^{-1} \nabla_\theta J(\theta)$ is defined through the population Fisher matrix and the true policy gradient, neither of which is directly accessible. In practice, both quantities must be approximated from finite trajectory data, and the resulting stochastic error is subsequently filtered through the Fisher preconditioning operator $G(\theta)^{-1}$ (Bhatnagar et al., 2007; Xu et al., 2020b). Since preconditioning amplifies perturbations along directions of small Fisher curvature, precisely those directions in which the policy is least sensitive to parameter changes and along which sampling noise tends to concentrate, the quality of the natural gradient direction can be substantially degraded even when the raw gradient estimate is statistically adequate. This interaction between estimation variance and geometric preconditioning is not merely an implementation artifact; it is a structural property of the natural actor-critic framework that fundamentally governs update stability and conditioning. Making this propagation precise, and subsequently controlling it through structured low-discrepancy sampling, is our central analytical objective.

### 3.1 The Algorithm

The natural actor–critic framework estimates natural policy gradient specifically through compatible function approximation, exploiting the structural compatibility between the policy score functions and the natural-gradient. This methodology (Kakade, 2001; Peters et al., 2005), considers the policy score function $\psi(s, a) = \nabla_\theta \log \pi_\theta(a|s)$ as basis functions for advantage approximation. The term "*compatible*" refers to the property that these basis functions satisfy the condition $\nabla_w A_w(s, a) = \psi(s, a)$, ensuring alignment with the natural policy gradient structure (Sutton et al., 1999). This formulation is grounded in the policy gradient theorem,

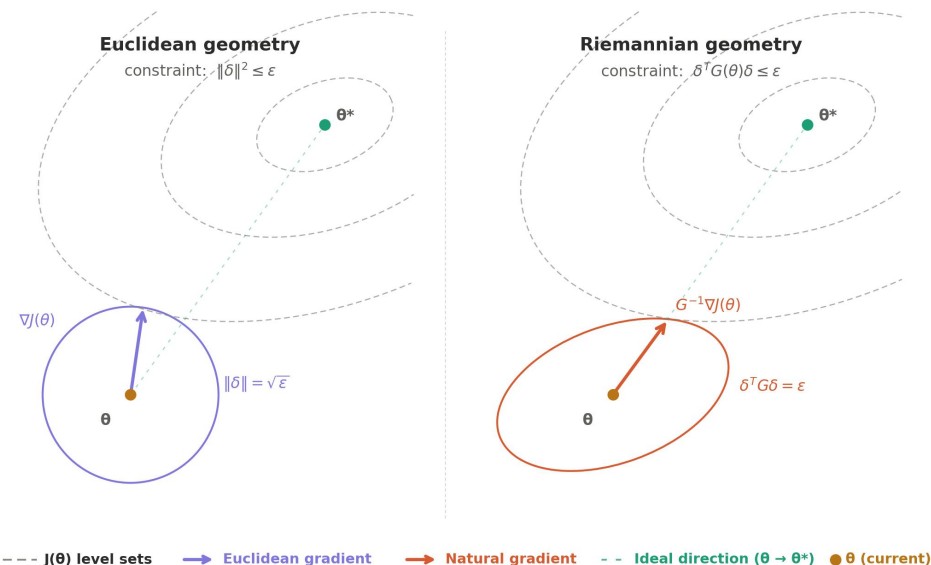

Figure 1: Comparison of Euclidean and natural gradient update geometry over identical $J(\theta)$ level sets (dashed). [ *Left* ] the Euclidean gradient $\nabla J(\theta)$ is constrained to a circular ball $\|\delta\|^2 \leq \varepsilon$, yielding a step that is nearly orthogonal to the direction toward the optimum $\theta^*$. [ *Right* ] the natural gradient $G(\theta)^{-1}\nabla J(\theta)$ is constrained to the KL ball $\delta^\top G(\theta)\delta \leq \varepsilon$, whose ellipsoidal geometry aligns with the curvature of the policy manifold, producing a step that more closely tracks the ideal direction toward $\theta^*$. Both steps consume the same budget $\varepsilon$, illustrating that the Fisher metric $G(\theta)$ redistributes the update along directions of meaningful policy change rather than raw parameter displacement.

which establishes that the score function inherently appears in gradient expressions. By employing $\psi(s, a)$ as compatible features, the resulting parameter estimates automatically yield natural gradient directions when solved through least-squares minimization as follows:

$$\min_{\overline{w}} \mathbb{E}_{(\mathbf{s},\mathbf{a}) \sim d^{\pi_\theta} \times \pi_\theta} \left[ \left( Q^{\pi_\theta}(\mathbf{s}, \mathbf{a}) - \widehat{Q}^{\pi_\theta}(\mathbf{s}, \mathbf{a}) \right)^2 \right], \tag{11}$$

where the expectation is taken with respect to the state-action distribution under the policy $\pi_\theta$ derived from the discounted state visitation distribution $d^{\pi_\theta}(s) = (1 - \gamma) \sum_{t=0}^{\infty} \gamma^t \mathbb{P}_{\pi_\theta}(\mathbf{s}_t = s)$. And,

$$Q^{\pi_\theta}(s, a) = A^{\pi_\theta}(s, a) + V^{\pi_\theta}(s)$$
$$\approx \psi(s, a)^\top w + \phi(s)^\top v = \widehat{\phi}(s, a)^\top \overline{w}, \tag{12}$$

with $\widehat{\phi}(s, a) = [\psi(s, a)^\top, \phi(s)^\top]^\top$ and $\overline{w} = [w^\top, v^\top]^\top$ are the augmented features and parameters, respectively. Thus, (11) seeks the best linear approximation of the Q-function in the augmented feature space (which includes the score function and the value function features). Taking the gradient of the objective and setting it to zero yields:

$$\nabla_{\overline{w}} \mathbb{E}_{(\mathbf{s},\mathbf{a}) \sim d^{\pi_\theta} \times \pi_\theta} \left[ \left( Q^{\pi_\theta}(\mathbf{s}, \mathbf{a}) - \widehat{\phi}(\mathbf{s}, \mathbf{a})^\top \overline{w} \right)^2 \right] = 0.$$

This gives:

$$\mathbb{E}_{(\mathbf{s},\mathbf{a})} \left[ \left( Q^{\pi_\theta}(\mathbf{s}, \mathbf{a}) - \widehat{\phi}(\mathbf{s}, \mathbf{a})^\top \overline{w} \right) \widehat{\phi}(\mathbf{s}, \mathbf{a}) \right] = 0. \tag{13}$$

Using the Bellman equation $Q^{\pi_\theta}(s, a) = r(s, a) + \gamma V^{\pi_\theta}(s')$ and the approximation $V^{\pi_\theta}(s') \approx \phi(s')^\top v$, we substitute into (13):

$$\mathbb{E}_{\substack{(\mathbf{s},\mathbf{a}) \sim d^{\pi_\theta} \times \pi_\theta \\ \mathbf{s}' \sim \mathbb{P}(\cdot|\mathbf{s},\mathbf{a})}} \left[ \left( r(\mathbf{s}, \mathbf{a}) + \gamma \phi(\mathbf{s}')^\top v - \widehat{\phi}(\mathbf{s}, \mathbf{a})^\top \overline{w} \right) \widehat{\phi}(\mathbf{s}, \mathbf{a}) \right] = 0.$$

Rewriting in terms of $\bar{w}$:

$$\mathbb{E}_{(\mathbf{s},\mathbf{a},\mathbf{s}')}\left[\widehat{\phi}(\mathbf{s},\mathbf{a})\left(\widehat{\phi}(\mathbf{s},\mathbf{a})^\top \overline{w} - \gamma\widetilde{\phi}(\mathbf{s}')^\top \overline{w}\right)\right] = \mathbb{E}_{(\mathbf{s},\mathbf{a})\sim d^{\pi_\theta}\times\pi_\theta}\left[r(\mathbf{s},\mathbf{a})\widehat{\phi}(\mathbf{s},\mathbf{a})\right],$$

where $\widetilde{\phi}(s') = [\mathbf{0}^\top, \phi(s')^\top]^\top \in \mathbb{R}^{q+d}$ and $\mathbf{0}$ is a zero vector with the same dimension as $w$. This yields the linear system:

$$\Xi\,\overline{w} = b, \tag{14}$$

where

$$\Xi = \mathbb{E}_{\substack{(\mathbf{s},\mathbf{a})\sim d^{\pi_\theta}\times\pi_\theta \\ \mathbf{s}'\sim\mathbb{P}(\cdot|\mathbf{s},\mathbf{a})}}\left[\widehat{\phi}(\mathbf{s},\mathbf{a})\left(\widehat{\phi}(\mathbf{s},\mathbf{a}) - \gamma\,\widetilde{\phi}(\mathbf{s}')\right)^\top\right], \text{ and } b = \mathbb{E}_{(\mathbf{s},\mathbf{a})\sim d^{\pi_\theta}\times\pi_\theta}\left[r(\mathbf{s},\mathbf{a})\widehat{\phi}(\mathbf{s},\mathbf{a})\right]. \tag{15}$$

Solving the linear system $\Xi\,\bar{w} = b$ avoids explicit Fisher matrix inversion by casting natural actor–critic estimation as a single least-squares problem over compatible features. The solution $\bar{w}^* = (w^*, v^*)$ separates naturally into a component $w^*$ that recovers the natural gradient $G(\theta)^{-1}\nabla_\theta J(\theta)$ and a component $v^*$ corresponding to the critic parameters. This formulation therefore couples natural-gradient estimation and value-function learning within a single regression.

For notational convenience, we write $\boldsymbol{\psi} = \boldsymbol{\psi}(s,a), \phi = \phi(s), \phi' = \phi(s'), \widehat{\phi} = \widehat{\phi}(s,a)$ and $\widetilde{\phi}' = \widetilde{\phi}(s')$, where $s' \sim \mathbb{P}(\cdot \mid s, a)$.

**Remark 2.** In the above, we have a compatible critic of the form $f_w(s,a) = \boldsymbol{\psi}(s,a)^\top w$ and is fit by least squares to a target $\widehat{Q}^{\pi_\theta}$ as follows: $w_\theta = \arg\min_w \mathbb{E}[(\widehat{Q}^{\pi_\theta} - \boldsymbol{\psi}^\top w)^2]$, with $\widehat{Q}^{\pi_\theta}(s,a) = r(s,a) + \gamma\,\phi(s')^\top v$. By the first-order optimality conditions of the least-squares problem, the solution $w_\theta$ satisfies the normal equations $\mathbb{E}[\boldsymbol{\psi}(\widehat{Q}^{\pi_\theta} - \boldsymbol{\psi}^\top w_\theta)] = \mathbf{0}$. This means $\widehat{Q}^{\pi_\theta}$ decomposes into two orthogonal pieces in $L^2(d^{\pi_\theta}\times\pi_\theta)$:

$$\widehat{Q}^{\pi_\theta} = \underbrace{\boldsymbol{\psi}^\top w_\theta}_{\substack{\text{projection onto}\\\text{span}\{\boldsymbol{\psi}\}}} + \underbrace{\zeta,}_{\substack{\text{orthogonal}\\\text{residual}}}$$

with $\mathbb{E}[\boldsymbol{\psi}\,\zeta] = \mathbf{0}$. So the residual $\zeta$ has no component along any score direction, precisely what "compatible" enforces. To further understand why this zeros the harmful part, consider the actor update using the compatible critic $f_{w_\theta}$:

$$G^{-1}\,\mathbb{E}[\boldsymbol{\psi}\,f_{w_\theta}] = G^{-1}\,\mathbb{E}[\boldsymbol{\psi}\,\boldsymbol{\psi}^\top]\,w_\theta = G^{-1}G\,w_\theta = w_\theta.$$

If instead one uses the raw target $\widehat{Q}^{\pi_\theta}$, then

$$\begin{aligned}
G^{-1}\,\mathbb{E}[\boldsymbol{\psi}\,\widehat{Q}^{\pi_\theta}] &= G^{-1}\,\mathbb{E}[\boldsymbol{\psi}\,(\boldsymbol{\psi}^\top w_\theta + \zeta)] \\
&= G^{-1}\,(G\,w_\theta + \underbrace{\mathbb{E}[\boldsymbol{\psi}\,\zeta]}_{=\,\mathbf{0}}) = w_\theta.
\end{aligned}$$

Thus, both ways, we obtain the same step $w_\theta$. Crucially, the only component that could bias the actor, $\mathbb{E}[\boldsymbol{\psi}(\widehat{Q}^{\pi_\theta} - \boldsymbol{\psi}^\top w_\theta)]$, is forced to zero by the regression optimality. The actor "sees" only directions in the score span $\text{span}\{\boldsymbol{\psi}\}$. Compatibility projects the (possibly noisy) target $\widehat{Q}^{\pi_\theta}$ onto that span, discarding the orthogonal component the actor cannot use anyway. This projection is exactly what is needed to compute the natural step, with no explicit Fisher inverse required.

To ensure the computational feasibility of the natural actor-critic framework, it is essential that the linear system derived from compatible function approximation admits a unique solution. The following theorem establishes sufficient conditions for the invertibility of the matrix $\Xi$, thereby guaranteeing the well-posedness of the linear system.

**Theorem 2.** *Let* $(\mathbf{s}, \mathbf{a}) \sim d^{\pi_\theta} \times \pi_\theta$ *and* $\mathbf{s}' \sim \mathbb{P}(\cdot \mid \mathbf{s}, \mathbf{a})$. *Let* $\Xi_1 = \mathbb{E}\big[\widehat{\phi}(\mathbf{s}, \mathbf{a})\widehat{\phi}(\mathbf{s}, \mathbf{a})^\top\big]$ *and* $\Xi_2 = \mathbb{E}\big[\widehat{\phi}(\mathbf{s}, \mathbf{a})\,\widetilde{\phi}(\mathbf{s}')^\top\big]$. *Hence, under Assumptions 1–6 and for* $\gamma < \lambda_{\min}(\Xi_1)/C_{\widehat{\phi}}$, *with* $C_{\widehat{\phi}} = \max\{C_\psi, C_\phi\}$, *we have* $\Xi \succ 0$, *and the linear system* $\Xi\,\overline{w} = b$ *has the unique solution* $\overline{w}^* = \Xi^{-1}b$.

*Proof.* By Assumption 6, we obtain

$$z^\top \Xi_1 z = \mathbb{E}\Big[\big(z^\top \widehat{\phi}(\mathbf{s}, \mathbf{a})\big)^2\Big] > 0, \quad \text{for all } z \neq 0, \tag{16}$$

hence $\Xi_1$ is positive definite. Hence, $\lambda_{\min}(\Xi_1) > 0$. Now using Cauchy–Schwarz and boundedness of the features (Assumption 2),

$$\|\Xi_2\| = \big\|\mathbb{E}\big[\widehat{\phi}(\mathbf{s}, \mathbf{a})\,\widetilde{\phi}(\mathbf{s}')^\top\big]\big\| \leq \mathbb{E}\big[\|\widehat{\phi}\|\,\|\widetilde{\phi}\|\big] \leq C_{\widehat{\phi}}^2, \text{ where } C_{\widehat{\phi}} := \max\{C_\psi, C_\phi\}.$$

Further, for any vector $z \in \mathbb{R}^{q+d}$, we have

$$z^\top \Xi_2 z = z^\top \left(\tfrac{\Xi_2 + \Xi_2^\top}{2}\right) z$$
$$\leq \left\|\tfrac{\Xi_2 + \Xi_2^\top}{2}\right\| \|z\|^2$$
$$\leq \tfrac{1}{2}\big(\|\Xi_2\| + \|A_2^\top\|\big) \|z\|^2$$
$$= \|\Xi_2\| \|z\|^2, \text{ since } \|\Xi_2^\top\| = \|\Xi_2\|.$$

Now for any unit vector $z$,

$$z^\top \Xi z = z^\top \Xi_1 z - \gamma\, z^\top \Xi_2 z$$
$$\geq \lambda_{\min}(\Xi_1)\|z\|^2 - \gamma\,\|\Xi_2\|$$
$$\geq \lambda_{\min}(\Xi_1) - \gamma\, C_{\widehat{\phi}}^2.$$

Thus if $\gamma < \lambda_{\min}(\Xi_1)/C_{\widehat{\phi}}^2$, we have $z^\top \Xi z > 0$ for all $z \neq 0$, *i.e.*, $\Xi \succ 0$. Hence $\Xi$ is invertible, so the normal equations $\Xi\,\overline{w} = b$ admit the unique solution $\overline{w}^* = \Xi^{-1}b$. $\qquad\square$

The above results establish that the matrix $\Xi$ is positive definite, and thus invertible, provided the discount factor $\gamma$ is bounded by $\gamma < \lambda_{\min}(\Xi_1)/C_{\widehat{\phi}}^2$. Technically, this condition ensures that the covariance structure $\Xi_1$ dominates the temporal difference term $\gamma \Xi_2$ in the matrix $\Xi = \Xi_1 - \gamma \Xi_2$, preserving the positive definiteness inherited from $\Xi_1$. This guarantees the numerical stability of solving the linear system $\Xi\overline{w} = \mathbf{b}$ and is a prerequisite for any meaningful policy update. Under these guarantees, the following result characterizes the solution itself. It demonstrates that under perfect value function approximation, the component $w$ of the solution vector $\overline{w}^*$ corresponds exactly to the natural policy gradient. It establishes the validity of the compatible function approximation.

**Theorem 3.** *Let* $V^{\pi_\theta}$ *be perfectly represented by the linear approximation* $V^{\pi_\theta}(s) = \phi(s)^\top v$ *for all* $s \in \mathcal{S}$, *for some* $v \in \mathbb{R}^d$. *The solution* $\overline{w}^* = [w^\top, v^\top]^\top$ *to the linear system (14) satisfies* $w = G(\theta)^{-1}\nabla_\theta J(\theta)$.

*Proof.* We first expand the expectation on the left-hand side of (14)

$$\mathbb{E}[\widehat{\phi}(\widehat{\phi} - \gamma\widetilde{\phi}')^\top] = \mathbb{E}\left[\begin{bmatrix} \psi\psi^\top & \psi(\phi^\top - \gamma\phi'^\top) \\ \phi\psi^\top & \phi(\phi^\top - \gamma\phi'^\top) \end{bmatrix}\right] = \begin{bmatrix} \mathbb{E}[\psi\psi^\top] & \mathbb{E}[\psi(\phi^\top - \gamma\phi'^\top)] \\ \mathbb{E}[\phi\psi^\top] & \mathbb{E}[\phi(\phi^\top - \gamma\phi'^\top)] \end{bmatrix}$$

Similarly, the right-hand side of (14) expands to

$$\mathbb{E}[\mathbf{r}\widehat{\phi}] = \begin{bmatrix} \mathbb{E}[\mathbf{r}\psi] \\ \mathbb{E}[\mathbf{r}\phi] \end{bmatrix}$$

The linear system can thus be written as two equations

$$\mathbb{E}[\boldsymbol{\psi\psi}^\top]w + \mathbb{E}[\boldsymbol{\psi}(\boldsymbol{\phi}^\top - \gamma\boldsymbol{\phi}'^\top)]v = \mathbb{E}[\mathbf{r}\boldsymbol{\psi}] \tag{17}$$

$$\mathbb{E}[\boldsymbol{\phi\psi}^\top]w + \mathbb{E}[\boldsymbol{\phi}(\boldsymbol{\phi}^\top - \gamma\boldsymbol{\phi}'^\top)]v = \mathbb{E}[\mathbf{r}\boldsymbol{\phi}] \tag{18}$$

From the definition of the Fisher information matrix, we have $\mathbb{E}[\boldsymbol{\psi\psi}^\top] = G(\theta)$. Now, under the assumption of perfect value function approximation (i.e., $V^{\pi_\theta}(s) = \boldsymbol{\phi}(s)^\top v$), we have from the Bellman equation:

$$Q^{\pi_\theta}(s,a) = r(s,a) + \gamma V^{\pi_\theta}(s')$$
$$= r(s,a) + \gamma\boldsymbol{\phi}(s')^\top v$$
$$\text{and } A^{\pi_\theta}(s,a) = Q^{\pi_\theta}(s,a) - V^{\pi_\theta}(s)$$
$$= r(s,a) + \gamma\boldsymbol{\phi}(s')^\top v - \boldsymbol{\phi}(s)^\top v$$

Thus, we can rewrite (17) as

$$G(\theta)w + \mathbb{E}[\boldsymbol{\psi}(\mathbf{r} + \gamma\boldsymbol{\phi}'^\top v - \boldsymbol{\phi}^\top v - \mathbf{r})] = \mathbb{E}[\mathbf{r}\boldsymbol{\psi}]$$
$$\Rightarrow G(\theta)w + \mathbb{E}[\boldsymbol{\psi}(\gamma\boldsymbol{\phi}'^\top v - \boldsymbol{\phi}^\top v)] = \mathbb{E}[\mathbf{r}\boldsymbol{\psi}] - \mathbb{E}[\boldsymbol{\psi}\mathbf{r}]$$
$$\Rightarrow G(\theta)w - \mathbb{E}[\boldsymbol{\psi}A^{\pi_\theta}(\mathbf{s},\mathbf{a})] = 0 \tag{19}$$

From the policy gradient theorem, we have

$$\nabla_\theta J(\theta) = \mathbb{E}[\boldsymbol{\psi}Q^{\pi_\theta}(\mathbf{s},\mathbf{a})]$$
$$= \mathbb{E}[\boldsymbol{\psi}A^{\pi_\theta}(\mathbf{s},\mathbf{a})] + \mathbb{E}[\boldsymbol{\psi}V^{\pi_\theta}(\mathbf{s})]$$

Since $\mathbb{E}[\boldsymbol{\psi}|\mathbf{s}] = 0$ (as $\boldsymbol{\psi}$ is the score function), we have:

$$\mathbb{E}[\boldsymbol{\psi}V^{\pi_\theta}(\mathbf{s})] = \mathbb{E}[\mathbb{E}[\boldsymbol{\psi}|\mathbf{s}]V^{\pi_\theta}(\mathbf{s})] = 0$$

Therefore, $\nabla_\theta J(\theta) = \mathbb{E}[\boldsymbol{\psi}A^{\pi_\theta}(\mathbf{s},\mathbf{a})]$. Substituting back into (19), we get,

$$G(\theta)w - \nabla_\theta J(\theta) = 0 \quad \Rightarrow \quad w = G(\theta)^{-1}\nabla_\theta J(\theta)$$

$$\square$$

The above result confirms that NAC coincides with the exact natural policy gradient when the critic is perfect. However, if the value function class cannot fit $V^{\pi_\theta}$ exactly, it incurs a bias. We now provide a fundamental theoretical characterization of the bias in natural actor-critic methods under function approximation, showing that imperfect value function representation introduces a systematic bias term that projects the Bellman error onto the natural gradient direction. We let $\mathbb{E}_{\pi_\theta}[\cdot] = \mathbb{E}_{\substack{\mathbf{s}\sim d^{\pi_\theta},\mathbf{a}\sim\pi_\theta \\ \mathbf{s}'\sim\mathbb{P}(\cdot|\mathbf{s},\mathbf{a})}}[\cdot]$.

**Theorem 4.** *Let $V^{\pi_\theta}(s)$ be approximated by a linear function $V_v(s) = \boldsymbol{\phi}(s)^\top v$, which is not a perfect representation. The solution $\overline{w} = [w^\top, v^\top]^\top$ to the linear system (14) satisfies:*

$$w = G(\boldsymbol{\theta})^{-1}\nabla_{\boldsymbol{\theta}}J(\boldsymbol{\theta}) + G(\boldsymbol{\theta})^{-1}\mathbb{E}_{\pi_\theta}[\boldsymbol{\psi}(\mathbf{s},\mathbf{a})\cdot\varepsilon(\mathbf{s},\mathbf{a},\mathbf{s}')]$$

*where $\varepsilon(s,a,s')$ is the Bellman error for transition $(s,a,s')$, defined as $\varepsilon(s,a,s') = (r(s,a)+\gamma V_{\mathbf{v}}(s')) - Q^{\pi_\theta}(s,a)$*

*Proof.* From the linear system (14), we have

$$\mathbb{E}_{\pi_\theta}[\widehat{\boldsymbol{\phi}}(\widehat{\boldsymbol{\phi}} - \gamma\widetilde{\boldsymbol{\phi}}')^\top]\overline{w} = \mathbb{E}_{\pi_\theta}[\mathbf{r}\widehat{\boldsymbol{\phi}}].$$

Then,

$$\mathbb{E}[\boldsymbol{\psi\psi}^\top]w + \mathbb{E}[\boldsymbol{\psi}(\boldsymbol{\phi}^\top - \gamma\boldsymbol{\phi}'^\top)]v = \mathbb{E}[\mathbf{r}\boldsymbol{\psi}] \tag{20}$$

The true Bellman equation for $Q^{\pi_\theta}$ is

$$Q^{\pi_\theta}(s,a) = r(s,a) + \gamma \mathbb{E}_{\mathbf{s}' \sim \mathbb{P}(\cdot|\mathbf{s},\mathbf{a})}[V^{\pi_\theta}(\mathbf{s}')]$$

Our approximation is $\widehat{Q}(s,a) = \boldsymbol{\psi}(s,a)^\top w + \boldsymbol{\phi}(s)^\top v$, and we have an approximated value function $V_v(s) = \boldsymbol{\phi}(s)^\top v$. For a specific transition $(s,a,s')$, the Bellman error is:

$$\varepsilon(s,a,s') = (r(s,a) + \gamma V_v(s')) - Q^{\pi_\theta}(s,a)$$
$$\Rightarrow r(s,a) = Q^{\pi_\theta}(s,a) - \gamma V_v(s') + \varepsilon(s,a,s')$$

Now, substitute this expression into the RHS of (20).

$$\mathbb{E}[r\boldsymbol{\psi}] = \mathbb{E}[(Q^{\pi_\theta}(\mathbf{s},\mathbf{a}) - \gamma V_v(\mathbf{s}') + \varepsilon(\mathbf{s},\mathbf{a},\mathbf{s}'))\boldsymbol{\psi}]$$
$$= \mathbb{E}[Q^{\pi_\theta}(\mathbf{s},\mathbf{a})\boldsymbol{\psi}] - \gamma \mathbb{E}[V_v(\mathbf{s}')\boldsymbol{\psi}] + \mathbb{E}[\varepsilon(\mathbf{s},\mathbf{a},\mathbf{s}')\boldsymbol{\psi}]. \tag{21}$$

Now rewrite the LHS of (20) as

$$\text{LHS} = \mathbb{E}[\boldsymbol{\psi}\boldsymbol{\psi}^\top]w + \mathbb{E}[\boldsymbol{\psi}\boldsymbol{\phi}^\top v] - \gamma \mathbb{E}[\boldsymbol{\psi}\boldsymbol{\phi}'^\top v]$$
$$= \mathbb{E}[\boldsymbol{\psi}\boldsymbol{\psi}^\top]w + \mathbb{E}[\boldsymbol{\psi}V_v(\mathbf{s})] - \gamma \mathbb{E}[\boldsymbol{\psi}V_v(\mathbf{s}')] \tag{22}$$

Now, equating the expanded LHS and RHS:

$$\mathbb{E}[\boldsymbol{\psi}\boldsymbol{\psi}_t^\top]w + \mathbb{E}[\boldsymbol{\psi}V_v(\mathbf{s})] - \gamma \mathbb{E}[\boldsymbol{\psi}V_v(\mathbf{s}')] = \mathbb{E}[Q^{\pi_\theta}(s,a)\boldsymbol{\psi}] - \gamma \mathbb{E}[V_{\mathbf{v}}(s')\boldsymbol{\psi}] + \mathbb{E}[\varepsilon(\mathbf{s},\mathbf{a},\mathbf{s}')\boldsymbol{\psi}]$$
$$\Rightarrow \mathbb{E}[\boldsymbol{\psi}\boldsymbol{\psi}^\top]w + \mathbb{E}[\boldsymbol{\psi}V_v(\mathbf{s})] = \mathbb{E}[Q^{\pi_\theta}(\mathbf{s},\mathbf{a})\boldsymbol{\psi}] + \mathbb{E}[\varepsilon(\mathbf{s},\mathbf{a},\mathbf{s}')\boldsymbol{\psi}].$$

Now, add and subtract $\mathbb{E}[V_v(\mathbf{s})\boldsymbol{\psi}]$

$$\mathbb{E}[\boldsymbol{\psi}\boldsymbol{\psi}^\top]w + \mathbb{E}[\boldsymbol{\psi}V_v(\mathbf{s})] = \mathbb{E}[(Q^{\pi_\theta}(\mathbf{s},\mathbf{a}) - V_v(\mathbf{s}))\boldsymbol{\psi}] + \mathbb{E}[V_v(\mathbf{s})\boldsymbol{\psi}] + \mathbb{E}[\varepsilon(\mathbf{s},\mathbf{a},\mathbf{s}')\boldsymbol{\psi}]$$
$$\Rightarrow \mathbb{E}[\boldsymbol{\psi}\boldsymbol{\psi}^\top]w = \mathbb{E}[(Q^{\pi_\theta}(\mathbf{s},\mathbf{a}) - V_v(\mathbf{s}))\boldsymbol{\psi}] + \mathbb{E}[\varepsilon(\mathbf{s},\mathbf{a},\mathbf{s}')\boldsymbol{\psi}]$$
$$\Rightarrow G(\boldsymbol{\theta})w = \mathbb{E}[(Q^{\pi_\theta}(\mathbf{s},\mathbf{a}) - V_v(\mathbf{s}))\boldsymbol{\psi}] + \mathbb{E}[\varepsilon(\mathbf{s},\mathbf{a},\mathbf{s}')\boldsymbol{\psi}]. \tag{23}$$

Define the approximate advantage function:

$$\widetilde{A}(s,a) = Q^{\pi_\theta}(s,a) - V_v(s)$$
$$= A^{\pi_\theta}(s,a) + (V^{\pi_\theta}(s) - V_v(s))$$

Therefore,

$$\mathbb{E}\left[\widetilde{A}(\mathbf{s},\mathbf{a})\boldsymbol{\psi}\right] = \mathbb{E}[A^{\pi_\theta}(\mathbf{s},\mathbf{a})\boldsymbol{\psi}] + \mathbb{E}[(V^{\pi_\theta}(\mathbf{s}) - V_v(\mathbf{s}))\boldsymbol{\psi}]$$

Since $\mathbb{E}[\boldsymbol{\psi}\,|\,\mathbf{s}] = 0$,

$$\mathbb{E}[(V^{\pi_\theta}(\mathbf{s}) - V_v(\mathbf{s}))\boldsymbol{\psi}_t] = \mathbb{E}[\mathbb{E}[(V^{\pi_\theta}(\mathbf{s}) - V_v(\mathbf{s}))\boldsymbol{\psi}_t\,|\,\mathbf{s}]]$$
$$= \mathbb{E}[(V^{\pi_\theta}(\mathbf{s}_t) - V_v(\mathbf{s})) \cdot \mathbb{E}[\boldsymbol{\psi}\,|\,\mathbf{s}]]$$
$$= 0.$$

Thus, $\mathbb{E}\left[\widetilde{A}(\mathbf{s},\mathbf{a})\boldsymbol{\psi}\right] = \mathbb{E}[A^{\pi_\theta}(\mathbf{s},\mathbf{a})\boldsymbol{\psi}]$. By the policy gradient theorem, we also have,

$$\nabla_{\boldsymbol{\theta}}J(\boldsymbol{\theta}) = \mathbb{E}[\boldsymbol{\psi}(\mathbf{s},\mathbf{a})Q^{\pi_\theta}(\mathbf{s},\mathbf{a})] = \mathbb{E}[\boldsymbol{\psi}(\mathbf{s},\mathbf{a})A^{\pi_\theta}(\mathbf{s},\mathbf{a})]$$

Therefore, $\mathbb{E}\left[\widetilde{A}(\mathbf{s},\mathbf{a})\boldsymbol{\psi}\right] = \nabla_{\boldsymbol{\theta}}J(\boldsymbol{\theta})$. Substituting back into (23), we get,

$$G(\boldsymbol{\theta})w = \nabla_{\boldsymbol{\theta}}J(\boldsymbol{\theta}) + \mathbb{E}[\varepsilon(\mathbf{s},\mathbf{a},\mathbf{s}')\boldsymbol{\psi}]$$
$$\Rightarrow w = G(\boldsymbol{\theta})^{-1}\nabla_{\boldsymbol{\theta}}J(\boldsymbol{\theta}) + G(\boldsymbol{\theta})^{-1}\mathbb{E}[\varepsilon(\mathbf{s},\mathbf{a},\mathbf{s}')\boldsymbol{\psi}].$$

$\square$

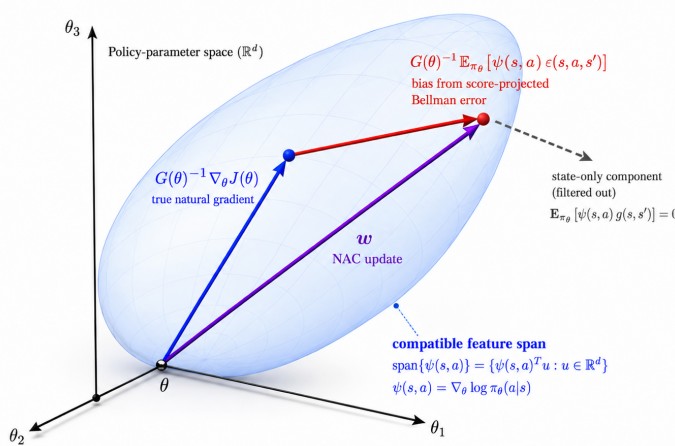

Figure 2: Under imperfect critic approximation, the NAC direction $w$ decomposes into the exact natural policy gradient $G(\theta)^{-1}\nabla_\theta J(\theta)$ and the Fisher-preconditioned score-projected Bellman residual $G(\theta)^{-1}\mathbb{E}_{\pi_\theta}[\psi(s,a)\epsilon(s,a,s')]$. The state-only component is filtered out since $\mathbb{E}_{\pi_\theta}[\psi(s,a)g(s,s')] = 0$, whereas the score-correlated Bellman error remains and biases the update direction.

The above result reveals that the bias in natural gradient estimation arises from the score-projected Bellman residual $P^*\varepsilon = \mathbb{E}[\boldsymbol{\psi}(\mathbf{s},\mathbf{a})\varepsilon(\mathbf{s},\mathbf{a},\mathbf{s}')]$ (where $P^*f = \mathbb{E}[\boldsymbol{\psi}f]$ is the score projector), which extracts precisely the components of function approximation error that correlate with policy sensitivity directions. While state-only approximation errors are automatically filtered out due to the fundamental property $P^*h = \mathbb{E}[\boldsymbol{\psi}h(\mathbf{s})] = 0$ for any state-dependent function $h$ (due to $\mathbb{E}[\boldsymbol{\psi}|s] = 0$), the Bellman error $\varepsilon(s,a,s') = (r(s,a) + \gamma V_v(s')) - Q^{\pi_\theta}(s,a)$ inherently contains action-dependent components that survive this projection. This occurs because imperfect value approximation creates Bellman errors that vary with action selection through the MDP dynamics—different actions lead to different next-state values and rewards, causing the approximation error to align with policy sensitivity. Consequently, policy updates are systematically distorted toward regions where value function errors correlate with the score function, rather than following the true steepest ascent direction of expected return. This creates a problematic feedback loop: biased gradients alter the state visitation distribution, rendering previously learned value functions increasingly inaccurate, which further amplifies the bias in subsequent iterations. The result is either convergence to suboptimal policies where the biased gradient estimate vanishes despite non-zero true gradients, or catastrophic divergence as the policy is progressively misled by accumulating approximation errors that survive the score projection filter.

## 4  Natural Actor–Critic with Array-RQMC Sampling

The theoretical guarantees of the natural actor-critic framework are dependent upon the accuracy of the estimated natural gradient. The compatible-feature solution is affected by stochastic estimation noise due to finite sampling. Both the bias term and the stochastic fluctuations are filtered through Fisher preconditioning and jointly influence the stability and accuracy of the natural-gradient update. Consequently, controlling estimator variance is essential to limit the impact of both bias and stochastic noise on the natural-gradient update. Standard Monte Carlo rollouts rely on independent uniforms, which leads to high-variance estimators and makes the natural-gradient update sensitive to noise in the critic. To reduce this variance while preserving unbiasedness, we now incorporate Randomized Quasi-Monte Carlo (RQMC) methods to sample trajectories more efficiently from $d^{\pi_\theta}$. We assume access to a generative model (simulator) represented by a deterministic oracle $\Lambda$, such that given the current state $\mathbf{s}_t \in \mathcal{S}$, an action $\mathbf{a}_t \sim \pi_\theta(\cdot|\mathbf{s}_t)$, and a uniform random seed $\mathbf{u}_t \sim \mathrm{Unif}([0,1)^{\dim(\mathcal{S})})$, the next state can be generated as: $\mathbf{s}_{t+1} = \Lambda(\mathbf{s}_t, \mathbf{a}_t, \mathbf{u}_t)$. Here $\Lambda$ encodes the probability transition $\mathbb{P}(\cdot|\mathbf{s}_t, \mathbf{a}_t)$ via the inverse transform sampling, i.e., $\Lambda(\mathbf{s}_t, \mathbf{a}_t, \mathbf{u}_t) = F^{-1}_{\mathbf{s}_t, \mathbf{a}_t}(\mathbf{u}_t)$, where $F_{\mathbf{s}_t, \mathbf{a}_t}$ is the CDF associated with $\mathbb{P}(\cdot|\mathbf{s}_t, \mathbf{a}_t)$.

## 4.1 Randomized Quasi Monte-Carlo

Quasi–Monte Carlo (QMC) methods are based on the construction of *low–discrepancy point sets*, which are deterministic sequences designed to cover the unit hypercube $[0, 1)^d$ more uniformly than pseudo–random samples. Informally, low discrepancy means that the points are distributed so as to avoid large gaps or clusters, ensuring that no region of the domain is systematically over– or under–represented. Such constructions are particularly well suited for numerical integration in moderate dimensions. A common formalization of low–discrepancy point sets is given by $(t, m, d)$–*nets*. A $(t, m, d)$–net in base $b$ is a set of $N = b^m$ points in $[0, 1)^d$ with the property that every axis–aligned $b$–adic box of volume $b^{t-m}$ contains exactly $b^t$ points. Digital $(t, m, d)$–nets provide a principled way to construct highly uniform point sets and form the foundation of many QMC sequences used in practice.

In this paper, we consider the *Sobol sequence* (Sobol', 1967), which is a widely used digital net in base 2. Sobol sequences are generated using a family of polynomials that are irreducible over the finite field $\mathbb{F}_2$, together with associated *direction numbers*. These direction numbers determine a set of generating matrices that map integers to points in $[0, 1)^d$ via binary expansions. Each point in the sequence is constructed through bitwise operations (notably exclusive–or operations) on these binary representations, resulting in a deterministic point set with strong uniformity properties. Various constructions of direction numbers have been proposed; in our implementation we follow the approach of (Joe & Kuo, 2008). While QMC point sets are deterministic and highly uniform, their performance can be sensitive to correlations between the integrand and the fixed point set. To address this issue, *randomized quasi–Monte Carlo* (RQMC) methods introduce randomness into the construction while preserving the low–discrepancy structure. A standard approach is *digital scrambling*, which applies random transformations to the generating matrices of a digital net, often combined with a *random digital shift* (Matoušek, 1998). In particular, left matrix scrambling replaces each generating matrix of the Sobol sequence with a randomly transformed version, yielding a randomized point set with uniform marginals. Random digital shifts are then applied via a bitwise exclusive–or with a random binary vector, ensuring that the resulting estimator is unbiased while retaining the variance–reduction properties of the underlying low–discrepancy construction (L'Ecuyer, 2018). These scrambled nets form the basic building blocks used in our Array–RQMC construction.

Array-RQMC is an extension of RQMC specifically designed to simulate Markov chains by running $N$ realizations in parallel. Let $\mathbf{s}_{i,j}$ be the state of the $i^{th}$ chain at time $j$, and let $\Omega_j = \{\mathbf{u}_{1,j}, \mathbf{u}_{2,j}, \ldots, \mathbf{u}_{N,j}\}$ be an independent RQMC point set with $\mathbf{u}_{i,j} \in [0, 1)^{\dim(\mathcal{S})}$. We have $N$ parallel chains $\{\mathbf{s}_{i,j}\}_{i=1}^{N}$. At each step $j$, we proceed as follows:

1. Sort the chain states $\{\mathbf{s}_{i,j}\}$ by a sort function $h$.

2. Generate and scramble a RQMC point set $\Omega_j$ of size $N$, then sort those points by their first coordinate.

3. Pair each chain with one RQMC point in that sorted order to sample the next state $\mathbf{s}_{i,j+1}$ as $\mathbf{s}_{i,j+1} = \Lambda(\mathbf{s}_{i,j}, \mathbf{a}_{i,j}, \mathbf{u}_{i,j})$, and the corresponding reward $\mathbf{r}_{i,j}$, where $\Lambda$ denotes the inverse transform of the transition function $\mathbb{P}$ of the Markov chain.

The $N$ trajectories are deliberately coupled since the driving uniforms are dependent, typically yielding negative inter-trajectory correlation, which reduces the variance of sample-mean estimators relative to Monte Carlo. However, it does not alter the marginal distribution for each chain. In other words, each $\mathbf{s}_{i,j+1}$ has the same distribution as it would under an IID Monte Carlo simulation, ensuring there is no bias in the chain evolution. By sorting the states (so neighbors in the array are similar states) and sorting the RQMC points (neighbors are nearby uniforms), then pairing rank-$i$ with rank-$i$, the simulator's output across the array change smoothly across parallel trajectories. Subsequently on averaging over a smooth sequence, the fluctuations cancel out much more than in an irregular one resulting in reduced variance.

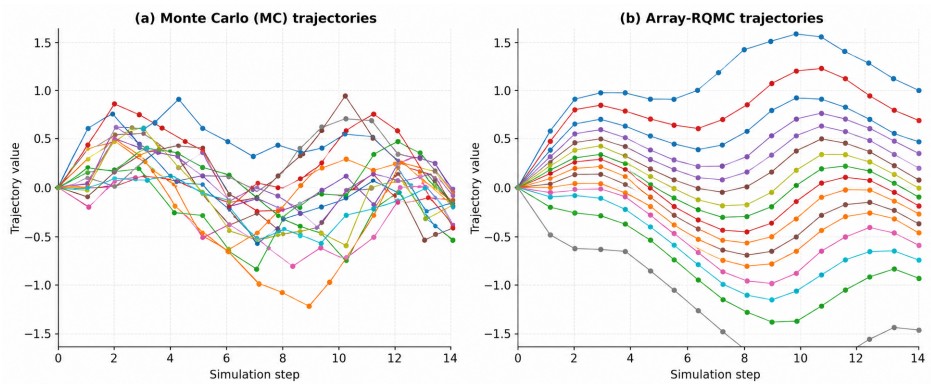

Figure 3: Monte Carlo [ *Left* ] shows irregular trajectories due to independent random sampling. Array RQMC [ *Right* ] uses randomized low-discrepancy sequences, producing more uniform and structured paths.

Finally, the Array–RQMC per-step pooled estimators are obtained as follows:

$$
\begin{aligned}
\widehat{\Xi}_N &= \frac{1}{N} \sum_{j=0}^{T-1} \left( \sum_{i=1}^{N} \widehat{\phi}_{i,j} \left( \widehat{\phi}_{i,j} - \gamma \, \widetilde{\phi}_{i,j} \right)^{\top} \right), \\
\widehat{\zeta}_N &= \frac{1}{N} \sum_{j=0}^{T-1} \left( \sum_{i=1}^{N} \mathbf{r}_{i,j} \, \widehat{\phi}_{i,j} \right).
\end{aligned}
\tag{24}
$$

**Theorem 5.** *Consider the Array-RQMC based estimation given in (24) using $N$ parallel chains and trajectory length $T$. Under the following assumptions:*

5.a. *For each $j$, the scrambled point set $\Omega_j = \{\mathbf{u}_{1,j}, \ldots, \mathbf{u}_{N,j}\} \subset [0,1)^{\dim(\mathcal{S})}$ satisfies $\mathbf{u}_{i,j} \sim \mathrm{Unif}([0,1)^{\dim(\mathcal{S})})$ for all $i$, and the sets $\{\Omega_j\}_{j=0}^{T-1}$ are independent across $j$.*

5.b. *For each $j$, the random vector $(\mathbf{u}_{1,j}, \ldots, \mathbf{u}_{N,j})$ is exchangeable, i.e., for every permutation $\sigma$ of $\{1, \ldots, N\}$, $(\mathbf{u}_{\sigma(1),j}, \ldots, \mathbf{u}_{\sigma(N),j}) \overset{d}{=} (\mathbf{u}_{1,j}, \ldots, \mathbf{u}_{N,j})$. Thus the joint distribution of the RQMC points is invariant under reordering.*

5.c. *For each $j$, the point set $\Omega_j$ is independent of $\{\mathcal{F}_j\}_{j \geq 0}$, the natural filtration generated by the state–action process and all past randomness up to time $j$, i.e., $\mathcal{F}_j := \sigma\left( \{(s_{i,t}, a_{i,t})\}_{i=1}^{N}, \Omega_0, \ldots, \Omega_{t-1} : 0 \leq t \leq j \right)$.*

5.d. *The transition $\Lambda$ satisfies $\Lambda(\mathbf{s}, \mathbf{a}, \mathbf{u}) \overset{d}{=} \mathbb{P}(\cdot \mid \mathbf{s}, \mathbf{a})$ for $\mathbf{u} \sim \mathrm{Unif}([0,1)^{\dim(\mathcal{S})})$, so that using $u \in [0,1)^{\dim(\mathcal{S})}$ via inverse transform reproduces the Markov kernel $\mathbb{P}(\cdot \mid s, a)$.*

5.e. *At each time $j$, the permutation $\sigma_j$ used to reorder the chains depends only on the current states: $\sigma_j = \sigma_j(\mathbf{s}_{1,j}, \ldots, \mathbf{s}_{N,j})$, and never on the RQMC points $\Omega_j$. After sorting, chain $i$ occupies position $k = \sigma_j(i)$ and receives the RQMC point $\mathbf{u}'_{k,j}$, the $k$-th element of the sorted point set.*

*Then $\widehat{\Xi}_N$ and $\widehat{b}_T$ the estimators are unbiased, i.e., $\mathbb{E}[\widehat{\Xi}_N] = \Xi$ and $\mathbb{E}[\widehat{\zeta}_N] = \zeta$.*

*Proof.* First we show that, despite the dependence induced across chains by Array–RQMC, the marginal law of each state–action–next-state triple $(\mathbf{s}_{i,j}, \mathbf{a}_{i,j}, \mathbf{s}_{i,j+1})$ coincides with that of a standard Monte Carlo simulation of the Markov chain. This further implies unbiasedness of $\widehat{\Xi}_N$ and $\widehat{\zeta}_N$. Let $\{(\mathbf{s}_{i,j}^{\mathrm{MC}}, \mathbf{a}_{i,j}^{\mathrm{MC}})\}$ be $N$ independent copies of the Markov chain driven by IID uniforms, i.e., $\mathbf{s}_{i,0}^{\mathrm{MC}} \sim \mu_0$ IID. across $i$, $\mathbf{a}_{i,j}^{\mathrm{MC}} \sim \pi_\theta(\cdot \mid$

$\mathbf{s}_{i,j}^{\mathrm{MC}}$), and $\mathbf{s}_{i,j+1}^{\mathrm{MC}} \sim \mathbb{P}(\cdot \mid \mathbf{s}_{i,j}^{\mathrm{MC}}, \mathbf{a}_{i,j}^{\mathrm{MC}})$. This is the standard MC simulator for the same MDP and policy. We will show that for all $i$ and $j$,

$$(\mathbf{s}_{i,j}, \mathbf{a}_{i,j}, \mathbf{s}_{i,j+1}) \overset{d}{=} (\mathbf{s}_{i,j}^{\mathrm{MC}}, \mathbf{a}_{i,j}^{\mathrm{MC}}, \mathbf{s}_{i,j+1}^{\mathrm{MC}}), \tag{25}$$

where $\overset{d}{=}$ denotes equality in distribution. We proceed by induction on $j$. For $j = 0$, we have $\mathbf{s}_{i,0} \sim \mu_0$ by initialization of the Array–RQMC chains, so $(\mathbf{s}_{i,0}, \mathbf{a}_{i,0}) \overset{d}{=} (\mathbf{s}_{i,0}^{\mathrm{MC}}, \mathbf{a}_{i,0}^{\mathrm{MC}})$ once actions are sampled from $\pi_\theta(\cdot \mid \mathbf{s}_{i,0})$, since both procedures use the same policy. Assume now that for some fixed $j \geq 0$,

$$(\mathbf{s}_{i,j}, \mathbf{a}_{i,j}) \overset{d}{=} (\mathbf{s}_{i,j}^{\mathrm{MC}}, \mathbf{a}_{i,j}^{\mathrm{MC}}) \quad \text{for all } i.$$

Fix a time $j$ and condition on the $\sigma$-algebra $\mathcal{F}_j$ generated by all states and actions up to time $j$ (and all past RQMC sets up to time $j - 1$). By 5.c, the scrambled point set $\Omega_j$ is independent of $\mathcal{F}_j$. By 5.a, each $\mathbf{u}_{i,j}$ is marginally $\mathrm{Unif}([0,1)^d)$, and by 5.b, the vector $(\mathbf{u}_{1,j}, \ldots, \mathbf{u}_{N,j})$ is exchangeable. The Array–RQMC procedure next computes a permutation $\sigma_j$ that depends only on $(\mathbf{s}_{1,j}, \ldots, \mathbf{s}_{N,j})$ (5.e), and reassigns the RQMC points by ordering them into a sorted set and pairing the chain with rank $k$ to point $\mathbf{u}_{k,j}'$. Since the sorting of $\Omega_j$ depends only on the RQMC points themselves, and $\sigma_j$ depends only on states, the composition $(\mathbf{u}_{1,j}', \ldots, \mathbf{u}_{N,j}')$ is still an exchangeable random vector with the same multiset of values as $\Omega_j$, and remains independent of $\mathcal{F}_j$. In particular, for any Borel set $B \subset [0,1)^{\dim(\mathcal{S})}$ and any index $k$, $\mathbb{P}(\mathbf{u}_{k,j}' \in B \mid \mathcal{F}_j) = \mathbb{P}(\mathbf{u}_{1,j} \in B) = \mathrm{Unif}(B)$, where the last equality uses marginal uniformity. Thus, conditionally on $\mathcal{F}_j$, each assigned point $\mathbf{u}_{k,j}'$ is $\mathrm{Unif}([0,1)^{\dim(\mathcal{S})})$. Now consider a fixed chain index $i$. After sorting, it occupies some rank $k = \sigma_j(i)$, and receives the point $\mathbf{u}_{k,j}'$. Hence,

$$\mathbf{u}_{\sigma_j(i),j}' \,\big|\, \mathcal{F}_j \;\sim\; \mathrm{Unif}([0,1)^{\dim(\mathcal{S})}). \tag{26}$$

The next state of chain $i$ is generated via $\mathbf{s}_{i,j+1} = \Lambda(\mathbf{s}_{i,j}, \mathbf{a}_{i,j}, \mathbf{u}_{\sigma_j(i),j}')$. By (26) and 5.d, $\mathbf{s}_{i,j+1} \,\big|\, \mathbf{s}_{i,j}, \mathbf{a}_{i,j} \overset{d}{=} \mathbb{P}(\cdot \mid \mathbf{s}_{i,j}, \mathbf{a}_{i,j})$, which is exactly the same conditional law as in the MC reference process. Together with the induction hypothesis $(\mathbf{s}_{i,j}, \mathbf{a}_{i,j}) \overset{d}{=} (\mathbf{s}_{i,j}^{\mathrm{MC}}, \mathbf{a}_{i,j}^{\mathrm{MC}})$, we get $(\mathbf{s}_{i,j+1}, \mathbf{a}_{i,j+1}) \overset{d}{=} (\mathbf{s}_{i,j+1}^{\mathrm{MC}}, \mathbf{a}_{i,j+1}^{\mathrm{MC}})$, where $\mathbf{a}_{i,j+1} \sim \pi_\theta(\cdot \mid \mathbf{s}_{i,j+1})$ in both processes. Hence (25) holds for all $j$ by induction.

Now define the measurable function $f_\Xi(\mathbf{s}, \mathbf{a}, \mathbf{s}') = \widehat{\phi}(\mathbf{s}, \mathbf{a})(\widehat{\phi}(\mathbf{s}, \mathbf{a}) - \gamma \widetilde{\phi}(\mathbf{s}'))^\top$. By construction,

$$\widehat{\Xi}_N = \frac{1}{N} \sum_{j=0}^{T-1} \sum_{i=1}^{N} f_\Xi(\mathbf{s}_{i,j}, \mathbf{a}_{i,j}, \mathbf{s}_{i,j+1}).$$

Using the coupling (25) and linearity of expectation,

$$\mathbb{E}[\widehat{\Xi}_N] = \frac{1}{N} \sum_{j=0}^{T-1} \sum_{i=1}^{N} \mathbb{E}\big[f_\Xi(\mathbf{s}_{i,j}, \mathbf{a}_{i,j}, \mathbf{s}_{i,j+1})\big] = \frac{1}{N} \sum_{j=0}^{T-1} \sum_{i=1}^{N} \mathbb{E}\big[f_\Xi(\mathbf{s}_{i,j}^{\mathrm{MC}}, \mathbf{a}_{i,j}^{\mathrm{MC}}, \mathbf{s}_{i,j+1}^{\mathrm{MC}})\big].$$

Each term in the double sum has the same distribution, and for large $j$ we are in the stationary regime with $(\mathbf{s}_{i,j}^{\mathrm{MC}}, \mathbf{a}_{i,j}^{\mathrm{MC}}, \mathbf{s}_{i,j+1}^{\mathrm{MC}}) \sim d_{\pi_\theta} \times \pi_\theta \times \mathbb{P}$, so that

$$\mathbb{E}\big[f_\Xi(\mathbf{s}_{i,j}^{\mathrm{MC}}, \mathbf{s}_{i,j}^{\mathrm{MC}}, \mathbf{s}_{i,j+1}^{\mathrm{MC}})\big] = \mathbb{E}_{(\mathbf{s}, \mathbf{a}, \mathbf{s}') \sim d_{\pi_\theta} \times \pi_\theta \times \mathbb{P}}\big[f_\Xi(\mathbf{s}, \mathbf{a}, \mathbf{s}')\big] = \Xi.$$

Thus $\mathbb{E}[\widehat{\Xi}_N] = \Xi$.

The unbiasedness of $\widehat{\zeta}_N$ can be proven identically. Define $f_\zeta(s, a) = r(s, a)\widehat{\phi}(s, a)$, so that

$$\widehat{\zeta}_N = \frac{1}{N} \sum_{j=0}^{T-1} \sum_{i=1}^{N} f_\zeta(\mathbf{s}_{i,j}, \mathbf{a}_{i,j}).$$

Now using the same coupling and stationarity argument, we obtain $\mathbb{E}[\widehat{\zeta}_N] = \mathbb{E}_{(\mathbf{s}, \mathbf{a}) \sim d_{\pi_\theta} \times \pi_\theta}\big[f_\zeta(\mathbf{s}, \mathbf{a})\big] = \zeta.$ $\quad\square$

**Remark 3.** Assumption 5.*a* ensures that each RQMC point is correctly distributed, exchangeability guarantees that sorting doesn't introduce bias, temporal independence prevents pathological correlations across time, and state-RQMC independence is crucial for the conditional uniformity argument. In practice, Sobol sequences with Owen scrambling and random digital shift satisfy these properties. Violation of any sub-assumption can lead to biased estimators, with state-RQMC dependence being particularly pernicious as it directly breaks the proof technique. The restriction in Assumption 5.b that sorting depends only on states is essential for preserving exchangeability. If sorting used both state and RQMC information, the conditional distribution of assigned RQMC points could become non-uniform. In practice, common sorting functions like state value estimates or coordinate-based ordering satisfy this assumption. Adaptive sorting strategies that incorporate RQMC information would violate this assumption and require separate theoretical analysis. Assumption 5.*d* connects the theoretical transition kernel to the practical implementation. It requires that the inverse transform method correctly simulates the Markov chain transitions. In discrete state spaces, this is achieved through the inverse CDF method. Violation of this assumption leads to incorrect transition distributions and biased gradient estimates. Assumption 5.c ensures that the only dependence introduced in Array-RQMC comes from the deterministic sorting step, not from interactions between the chain evolution and the quasi-random uniforms.

The complete pseudo-code of our proposed algorithm, termed aRNAC, under the discounted reward criteria is summarized in Algorithm 1.

---

**Algorithm 1 Array-RQMC Natural Actor-Critic (aRNAC)**

---

**Function** `aRNAC`($N$, $h$, $\alpha$, $\gamma$, $\epsilon$)

    **Input:** $N$: number of parallel chains; $h$: state sorting function for Array-RQMC; $\alpha$: learning rates for actor; $\gamma$: discount factor; $\epsilon$: convergence threshold for gradient direction

    **Initialize:** $\theta \leftarrow 0$, $\bar{w} \leftarrow 0$, $\bar{A} \leftarrow 0$, $\bar{b} \leftarrow 0$, $s_0 \sim$ random

    $\mathbf{s}_{i,0} \leftarrow s_0, \forall i \in \{0, 1, \ldots, N-1\}$               `// Initialize parallel chains`

1    **for** $j = 0, 1, 2, \ldots$ **do**

2       $i \leftarrow 0$, $\widehat{A}_j \leftarrow 0$, $\widehat{b}_j \leftarrow 0$            `// Reset per-iteration accumulators`

3       Generate new RQMC point set $\Omega_j$             `// Low-discrepancy sampling`

4       **for** $i = 0, 1, 2, \ldots U - 1$ **do**

          `// Policy execution with RQMC-enhanced transitions`

5          $\mathbf{a}_{i,j} \sim \pi_\theta(\cdot | \mathbf{s}_{i,j})$            `// Sample action from current policy`

6          $\mathbf{s}_{i,j+1} \leftarrow \Lambda(\mathbf{s}_{i,j}, \mathbf{a}_{i,j}, \widetilde{P}_i)$, $\mathbf{r}_{i,j} \leftarrow \mathbf{r}(\mathbf{s}_{i,j}, \mathbf{a}_{i,j})$ `// Feature computation for natural gradient estimation`

7          $\widehat{\phi}_{i,j} \leftarrow [\psi_{i,j}^T, \phi_{i,j}^T]^T$          `// Augmented features: score + value`

8          $\tilde{\phi}_{i,j} \leftarrow [\mathbf{0}^T, \phi_{i,j+1}^T]^T$          `// Next-state value features`

          `// Accumulate LSTD-Q statistics`

9          $\widehat{A}_{i+1,j} \leftarrow \widehat{A}_{i,j} + \widehat{\phi}_{i,j}(\widehat{\phi}_{i,j} - \gamma\tilde{\phi}_{i,j})^T$,

         $\widehat{b}_{i+1,j} \leftarrow \widehat{b}_{i,j} + \mathbf{r}_{i,j}\widehat{\phi}_{i,j}$

10      Sort $\mathbf{s}_{0,j+1}, \ldots, \mathbf{s}_{N-1,j+1}$ according to $h(\mathbf{s}_{i,j+1})$    `// Maintain low-discrepancy structure across chains`

11      **Critic Update: (Natural gradient estimation via LSTD-Q)**

12      $\overline{A}_{j+1} \leftarrow (1 - \frac{1}{j+1})\overline{A}_j + \frac{1}{j+1}\frac{1}{N}\widehat{A}_{N,j}$     `// Update Fisher matrix estimate`

13      $\overline{b} \leftarrow (1 - \frac{1}{j+1})\overline{b} + \frac{1}{j+1}\widehat{b}_j$         `// Update policy gradient estimate`

14      $\overline{w}_j \leftarrow \bar{A}^{-1}\bar{b}$            `// Solve for natural gradient (use SVD)`

15      **Actor Update: (Policy improvement) if** $\angle(\overline{w}_j, \overline{w}_{j-1}) \leq \epsilon$ **then**

16         $\theta_{j+1} \leftarrow \theta_j + \alpha\overline{w}_j$         `// Update policy parameters`

---

## 4.2 Analysis of Natural Actor-Critic

We analyze the finite-sample performance of the natural actor-critic algorithm under standard Monte Carlo sampling. The goal is to bound the error in estimating the natural policy gradient $\nabla_G J(\theta) = G(\theta)^{-1}\nabla J(\theta)$, where $G(\theta)$ is the Fisher information matrix and $\nabla J(\theta)$ is the policy gradient. We consider the setting where trajectories are generated under the current policy $\pi_\theta$, and estimates are constructed from $N$ independent sample trajectories. In the following analysis, we consider an idealized setting in which the action–value function $Q_{\pi_\theta}$ is available exactly when constructing the empirical estimators. This assumption isolates the sampling variability of the gradient estimators by removing approximation error arising from a learned critic. This allows to characterize the intrinsic sampling variance of natural-gradient estimation. This idealized variance-only setting serves as the baseline to which approximation error is later added.

We define the empirical estimators using $N$ independent samples $\{(\mathbf{s}_i, \mathbf{a}_i)\}_{i=1}^N$ drawn from $d^{\pi_\theta} \times \pi_\theta$:

$$\widehat{G} = \frac{1}{N}\sum_{i=1}^N \boldsymbol{\psi}(\mathbf{s}_i, \mathbf{a}_i)\boldsymbol{\psi}(\mathbf{s}_i, \mathbf{a}_i)^\top, \quad \widehat{\nabla}J(\theta) = \frac{1}{N}\sum_{i=1}^N \boldsymbol{\psi}(\mathbf{s}_i, \mathbf{a}_i)Q^{\pi_\theta}(\mathbf{s}_i, \mathbf{a}_i), \quad \widehat{\nabla}_G J(\theta) = \widehat{G}^{-1}\widehat{\nabla}J(\theta). \tag{27}$$

The following lemma bounds the estimation error of the empirical Fisher information matrix in Frobenius norm which for any matrix $A \in \mathbb{R}^{q \times q}$ is defined as $\|A\|_F = \sqrt{\sum_{i=1}^q \sum_{j=1}^q |A_{ij}|^2}$.

**Lemma 6.** *The empirical Fisher information matrix estimator $\widehat{G}$ satisfies:*

$$\mathbb{E}\left[\|\widehat{G} - G(\theta)\|^2\right] \leq \frac{C_G}{N}, \quad \text{where } C_G = C_{\boldsymbol{\psi}}^4.$$

*Proof.* We define the random matrix $\mathbf{x}_i = \boldsymbol{\psi}(s_i, a_i)\boldsymbol{\psi}(s_i, a_i)^\top$ for each sample $i$. Then, $\widehat{G} = \frac{1}{N}\sum_{i=1}^N \mathbf{x}_i$, and $G(\theta) = \mathbb{E}[\mathbf{x}_i]$. The estimation error in Frobenius norm squared decomposes as follows:

$$\|\widehat{G} - G(\theta)\|_F^2 = \sum_{j=1}^q \sum_{k=1}^q \left((\widehat{G} - G(\theta))_{jk}\right)^2.$$

Taking expectation and using linearity, we obtain

$$\mathbb{E}\left[\|\widehat{G} - G(\theta)\|_F^2\right] = \sum_{j,k}\mathbb{E}\left[\left((\widehat{G} - G(\theta))_{jk}\right)^2\right]. \tag{28}$$

Now, observe that $(\widehat{G} - G(\theta))_{jk} = \frac{1}{N}\sum_{i=1}^N ((\mathbf{x}_i)_{jk} - \mathbb{E}[(\mathbf{x}_i)_{jk}])$. Since the samples $\{(\mathbf{s}_i, \mathbf{a}_i)\}$ are i.i.d., the cross-terms vanish in expectation, and we obtain,

$$\mathbb{E}\left[\left((\widehat{G} - G(\theta))_{jk}\right)^2\right] = \frac{1}{N}\text{Var}\left((\mathbf{x}_i)_{jk}\right) \leq \frac{1}{N}\mathbb{E}\left[(\mathbf{x}_i)_{jk}^2\right]. \tag{29}$$

The $(j,k)$-th entry of $\mathbf{x}_i$ is $(\mathbf{x}_i)_{jk} = \boldsymbol{\psi}_j(s_i, a_i)\boldsymbol{\psi}_k(s_i, a_i)$. By the Cauchy–Schwarz inequality, $(\mathbf{x}_i)_{jk}^2 = \boldsymbol{\psi}_j^2\boldsymbol{\psi}_k^2 \leq \|\boldsymbol{\psi}\|^4$. From Assumption 2, the feature vector $\boldsymbol{\psi}(s, a)$ is bounded: $\|\boldsymbol{\psi}(s, a)\| \leq C_{\boldsymbol{\psi}}$ for all $s, a$. Therefore, $\mathbb{E}\left[(\mathbf{x}_i)_{jk}^2\right] \leq \mathbb{E}\left[\|\boldsymbol{\psi}\|^4\right] \leq C_{\boldsymbol{\psi}}^4$. Summing over all $q^2$ entries, we get, $\sum_{j,k}\mathbb{E}\left[(\mathbf{x}_i)_{jk}^2\right] \leq q^2 C_{\boldsymbol{\psi}}^4$. Further, note that $\sum_{j,k}(\mathbf{x}_i)_{jk}^2 = \|\mathbf{x}_i\|_F^2 = \|\boldsymbol{\psi}\boldsymbol{\psi}^\top\|_F^2$. Now for any vector $\boldsymbol{\psi} \in \mathbb{R}^q$, the outer product $\boldsymbol{\psi}\boldsymbol{\psi}^\top$ has the following Frobenius norm:

$$\|\boldsymbol{\psi}\boldsymbol{\psi}^\top\|_F^2 = \text{trace}\left((\boldsymbol{\psi}\boldsymbol{\psi}^\top)(\boldsymbol{\psi}\boldsymbol{\psi}^\top)\right) = \text{trace}\left(\boldsymbol{\psi}(\boldsymbol{\psi}^\top\boldsymbol{\psi})\boldsymbol{\psi}^\top\right) = (\boldsymbol{\psi}^\top\boldsymbol{\psi})\text{trace}(\boldsymbol{\psi}\boldsymbol{\psi}^\top) = \|\boldsymbol{\psi}\|^2\|\boldsymbol{\psi}\|^2 = \|\boldsymbol{\psi}\|^4.$$

Thus, $\|\mathbf{x}_i\|_F^2 = \|\boldsymbol{\psi}\|^4 \leq C_{\boldsymbol{\psi}}^4$, and consequently,

$$\mathbb{E}\left[\|\mathbf{x}_i\|_F^2\right] \leq C_{\boldsymbol{\psi}}^4. \tag{30}$$

Combining all together (28)-(30), we obtain,

$$\mathbb{E}\left[\|\widehat{G} - G(\theta)\|_F^2\right] = \sum_{j,k} \mathbb{E}\left[\left((\widehat{G} - G(\theta))_{jk}\right)^2\right] \leq \frac{1}{N}\sum_{j,k}\mathbb{E}\left[(\mathbf{x}_i)_{jk}^2\right] = \frac{1}{N}\mathbb{E}\left[\|\mathbf{x}_i\|_F^2\right] \leq \frac{C_\psi^4}{N}.$$

Therefore, $\mathbb{E}\left[\|\widehat{G} - G(\theta)\|^2\right] \leq \frac{C_\psi^4}{N}$, which completes the proof with $C_G = C_\psi^4$. $\qquad\square$

We now analyze the empirical policy gradient estimator constructed from sampled trajectories.

**Lemma 7.** *The empirical policy gradient estimator $\widehat{\nabla}J(\theta)$ satisfies the following:*

$$\mathbb{E}\left[\|\widehat{\nabla}J(\theta) - \nabla J(\theta)\|^2\right] \leq \frac{C_\nabla}{N}, \quad \text{where } C_\nabla = C_\psi^2 C_\mathcal{Q}^2 \text{ with } C_\mathcal{Q} = \frac{C_r}{1-\gamma}.$$

*Proof.* We define the random vector $\mathbf{y}_i = \boldsymbol{\psi}(\mathbf{s}_i, \mathbf{a}_i)Q^{\pi_\theta}(\mathbf{s}_i, \mathbf{a}_i)$. Then, $\widehat{\nabla}J(\theta) = \frac{1}{N}\sum_{i=1}^N \mathbf{y}_i$, and $\nabla J(\theta) = \mathbb{E}[\mathbf{y}_i]$. We now use exactly the same variance calculation as in the previous lemma, but with the Euclidean norm. The estimation error is now given by,

$$\|\widehat{\nabla}J(\theta) - \nabla J(\theta)\|^2 = \sum_{j=1}^q \left((\widehat{\nabla}J(\theta) - \nabla J(\theta))_j\right)^2.$$

Taking expectation and using linearity, we obtain,

$$\mathbb{E}\left[\|\widehat{\nabla}J(\theta) - \nabla J(\theta)\|^2\right] = \sum_{j=1}^q \mathbb{E}\left[\left((\widehat{\nabla}J(\theta) - \nabla J(\theta))_j\right)^2\right]. \tag{31}$$

Since $(\widehat{\nabla}J(\theta) - \nabla J(\theta))_j = \frac{1}{N}\sum_{i=1}^N \left((\mathbf{y}_i)_j - \mathbb{E}[(\mathbf{y}_i)_j]\right)$ and samples are i.i.d., we have

$$\mathbb{E}\left[\left((\widehat{\nabla}J(\theta) - \nabla J(\theta))_j\right)^2\right] = \frac{1}{N}\text{Var}\left((\mathbf{y}_i)_j\right) \leq \frac{1}{N}\mathbb{E}\left[(\mathbf{y}_i)_j^2\right]. \tag{32}$$

The $j$-th component is $(Y_i)_j = \boldsymbol{\psi}_j(\mathbf{s}_i, \mathbf{a}_i)Q^{\pi_\theta}(\mathbf{s}_i, \mathbf{a}_i)$. From Assumption 2, $|\boldsymbol{\psi}_j(s, a)| \leq C_\psi$, $\forall s, a$, and the $Q$-function is bounded by $C_\mathcal{Q} = C_r/1 - \gamma$ . Therefore,

$$(\mathbf{y}_i)_j^2 = \boldsymbol{\psi}_j(\mathbf{s}_i, \mathbf{a}_i)^2 \left(Q^{\pi_\theta}(\mathbf{s}_i, \mathbf{a}_i)\right)^2 \leq C_\psi^2 C_\mathcal{Q}^2.$$

Thus, $\mathbb{E}\left[(\mathbf{y}_i)_j^2\right] \leq C_\psi^2 C_\mathcal{Q}^2$. Now consider the squared norm of $\mathbf{y}_i$.

$$\|\mathbf{y}_i\|^2 = \sum_{j=1}^q (\mathbf{y}_i)_j^2 = \sum_{j=1}^q \boldsymbol{\psi}_j(\mathbf{s}_i, \mathbf{a}_i)^2 \left(Q^{\pi_\theta}(\mathbf{s}_i, \mathbf{a}_i)\right)^2 = \left(Q^{\pi_\theta}(\mathbf{s}_i, \mathbf{a}_i)\right)^2 \sum_{j=1}^q \boldsymbol{\psi}_j(\mathbf{s}_i, \mathbf{a}_i)^2$$

$$= \left(Q^{\pi_\theta}(\mathbf{s}_i, \mathbf{a}_i)\right)^2 \|\boldsymbol{\psi}(\mathbf{s}_i, \mathbf{a}_i)\|^2 \leq C_\mathcal{Q}^2 C_\psi^2. \tag{33}$$

Therefore, $\mathbb{E}\left[\|\mathbf{y}_i\|^2\right] \leq C_\psi^2 C_\mathcal{Q}^2$. Finally by combining all the above (31–33), we obtain,

$$\mathbb{E}\left[\|\widehat{\nabla}J(\theta) - \nabla J(\theta)\|^2\right] = \sum_{j=1}^q \mathbb{E}\left[\left((\widehat{\nabla}J(\theta) - \nabla J(\theta))_j\right)^2\right] \leq \frac{1}{N}\sum_{j=1}^q \mathbb{E}\left[(\mathbf{y}_i)_j^2\right] = \frac{1}{N}\mathbb{E}\left[\|\mathbf{y}_i\|^2\right] \leq \frac{C_\psi^2 C_\mathcal{Q}^2}{N}.$$

This completes the proof with $C_\nabla = C_\psi^2 C_\mathcal{Q}^2$. $\qquad\square$

We now establish the uniform boundedness of the natural gradient estimator under Tikhonov regularization

**Lemma 8.** *Under Assumptions 2–5, and using a Tikhonov regularized empirical Fisher matrix $\widehat{G}_{reg} = \widehat{G} + \epsilon I$ for some $\epsilon > 0$, there exists a constant $B_\epsilon > 0$ such that*

$$\left\| \widehat{\nabla}_G J(\theta) - \nabla_G J(\theta) \right\| \leq B_\epsilon \quad \text{almost surely,}$$

*where*

$$B_\epsilon = \left( \frac{2C_\psi C_\mathcal{Q}}{\epsilon} \left( 1 + \frac{C_\psi^2}{\lambda_{\min}} \right) + \frac{C_\psi C_\mathcal{Q}}{\lambda_{\min}} \right)^2$$

.

*Proof.* Observe that for the Tikhonov regularized empirical Fisher matrix $\widehat{G}_{\text{reg}} = \widehat{G} + \epsilon I$ for some $\epsilon > 0$, the empirical natural gradient is defined as $\widehat{\nabla}_G J(\theta) = \widehat{G}_{\text{reg}}^{-1} \widehat{\nabla} J(\theta)$. Now, we obtain the following bounds:

$$\|\widehat{\nabla} J(\theta)\| = \left\| \frac{1}{N} \sum_{i=1}^{N} \psi(\mathbf{s}_i, \mathbf{a}_i) Q^{\pi_\theta}(\mathbf{s}_i, \mathbf{a}_i) \right\| \leq \frac{1}{N} \sum_{i=1}^{N} \|\psi(\mathbf{s}_i, \mathbf{a}_i)\| |Q^{\pi_\theta}(\mathbf{s}_i, \mathbf{a}_i)| \leq C_\psi C_\mathcal{Q} \tag{34}$$

Similarly, $\|\nabla J(\theta)\| \leq C_\psi C_\mathcal{Q}$. Further,

$$\|\widehat{G}\| = \left\| \frac{1}{N} \sum_{i=1}^{N} \psi(\mathbf{s}_i, \mathbf{a}_i) \psi(\mathbf{s}_i, \mathbf{a}_i)^\top \right\| \leq \frac{1}{N} \sum_{i=1}^{N} \|\psi(\mathbf{s}_i, \mathbf{a}_i)\|^2 \leq C_\psi^2.$$

Similarly, $\|G(\theta)\| \leq C_\psi^2$. Also, from Assumption 5, we have $\|G(\theta)^{-1}\| \leq \frac{1}{\lambda_{\min}}$.

Now, for the regularized empirical Fisher matrix $\widehat{G}_{\text{reg}} = \widehat{G} + \epsilon I$, we have $\widehat{G}_{\text{reg}} \succeq \epsilon I \Rightarrow \|\widehat{G}_{\text{reg}}^{-1}\| \leq \frac{1}{\epsilon}$. Now, consider the following difference:

$$\widehat{\nabla}_G J(\theta) - \nabla_G J(\theta) = \widehat{G}_{\text{reg}}^{-1} \widehat{\nabla} J(\theta) - G(\theta)^{-1} \nabla J(\theta)$$
$$= \widehat{G}_{\text{reg}}^{-1} (\widehat{\nabla} J(\theta) - \nabla J(\theta)) + (\widehat{G}_{\text{reg}}^{-1} - G(\theta)^{-1}) \nabla J(\theta)$$

Taking norms, we get

$$\left\| \widehat{\nabla}_G J(\theta) - \nabla_G J(\theta) \right\| \leq \|\widehat{G}_{\text{reg}}^{-1}\| \|\widehat{\nabla} J(\theta) - \nabla J(\theta)\| + \|\widehat{G}_{\text{reg}}^{-1} - G(\theta)^{-1}\| \|\nabla J(\theta)\|$$

Using the bounds established above, we obtain

$$\left\| \widehat{\nabla}_G J(\theta) - \nabla_G J(\theta) \right\| \leq \frac{1}{\epsilon} \cdot 2C_\psi C_\mathcal{Q} + \|\widehat{G}_{\text{reg}}^{-1} - G(\theta)^{-1}\| \cdot C_\psi C_\mathcal{Q}$$
$$\leq \frac{1}{\epsilon} \cdot 2C_\psi C_\mathcal{Q} + \|\widehat{G}_{\text{reg}}^{-1}\| \|\widehat{G}_{\text{reg}} - G(\theta)\| \|G(\theta)^{-1}\| \cdot C_\psi C_\mathcal{Q} \tag{35}$$

Also, we have - $\|\widehat{G}_{\text{reg}}^{-1}\| \leq \frac{1}{\epsilon}$ - $\|G(\theta)^{-1}\| \leq \frac{1}{\lambda_{\min}}$ - $\|\widehat{G}_{\text{reg}} - G(\theta)\| \leq \|\widehat{G} - G(\theta)\| + \epsilon \|I\| \leq 2C_\psi^2 + \epsilon$. Therefore,

$$\|\widehat{G}_{\text{reg}}^{-1} - G(\theta)^{-1}\| \leq \frac{1}{\epsilon} \cdot (2C_\psi^2 + \epsilon) \cdot \frac{1}{\lambda_{\min}} = \frac{2C_\psi^2 + \epsilon}{\epsilon \lambda_{\min}} \tag{36}$$

Now, putting (36) in (35), we get

$$\left\| \widehat{\nabla}_G J(\theta) - \nabla_G J(\theta) \right\| \leq \frac{2C_\psi C_\mathcal{Q}}{\epsilon} + \frac{2C_\psi^2 + \epsilon}{\epsilon \lambda_{\min}} \cdot C_\psi C_\mathcal{Q}$$
$$\leq \frac{2C_\psi C_\mathcal{Q}}{\epsilon} \left( 1 + \frac{C_\psi^2}{\lambda_{\min}} \right) + \frac{C_\psi C_\mathcal{Q}}{\lambda_{\min}}$$

Hence,

$$\left\| \widehat{\nabla}_G J(\theta) - \nabla_G J(\theta) \right\|^2 \leq B_\epsilon, \text{ where } B_\epsilon = \left( \frac{2C_\psi C_\mathcal{Q}}{\epsilon} \left( 1 + \frac{C_\psi^2}{\lambda_{\min}} \right) + \frac{C_\psi C_\mathcal{Q}}{\lambda_{\min}} \right)^2.$$

$\square$

The following theorem quantifies non-asymptotically the mean squared error convergence rate of the regularized natural gradient estimator $\widehat{\nabla}_G J(\theta)$, characterizing how its estimation variance decays with the sample size $N$.

**Theorem 9.** *Under Assumptions 2–5, and using a Tikhonov regularized empirical Fisher matrix $\widehat{G}_{reg} = \widehat{G} + \epsilon I$ with $\epsilon > 0$, the regularized natural gradient estimator $\widehat{\nabla}_G J(\theta) = \widehat{G}_{reg}^{-1} \widehat{\nabla} J(\theta)$ satisfies:*

$$\mathbb{E}\left[\left\|\widehat{\nabla}_G J(\theta) - \nabla_G J(\theta)\right\|^2\right] \leq \frac{C_{gvar}}{N} + \frac{12\epsilon^2 M_{\nabla_G}^2}{\lambda_{\min}^4},$$

*where*

$$C_{gvar} = \frac{8}{\lambda_{\min}^2}\left(C_\nabla + M_{\nabla_G}^2 C_G\right) + \frac{4B_\epsilon C_G}{\lambda_{\min}^2},$$

*$M_{\nabla_G}$ bounds $\|\nabla_G J(\theta)\|$ (Remark 1), and $B_\epsilon$ is the almost-sure bound from Lemma 8 for the specific regularization parameter $\epsilon$.*

*Proof.* We let $\widehat{g} = \widehat{\nabla} J(\theta)$ and $g = \nabla J(\theta)$. For the regularized natural gradient estimator $\widehat{\nabla}_G J(\theta)$, the estimation error now decomposes as follows:

$$\widehat{\nabla}_G J(\theta) - \nabla_G J(\theta) = \widehat{G}_{reg}^{-1}\widehat{g} - G^{-1}g = \widehat{G}_{reg}^{-1}(\widehat{g} - g) + (\widehat{G}_{reg}^{-1} - G^{-1})g.$$

Using the identity $\widehat{G}_{reg}^{-1} - G^{-1} = \widehat{G}_{reg}^{-1}(G - \widehat{G}_{reg})G^{-1}$ and noting that $G - \widehat{G}_{reg} = (G - \widehat{G}) - \epsilon I$, we obtain:

$$\begin{aligned}
\widehat{\nabla}_G J(\theta) - \nabla_G J(\theta) &= \widehat{G}_{reg}^{-1}(\widehat{g} - g) + \widehat{G}_{reg}^{-1}[(G - \widehat{G}) - \epsilon I]G^{-1}g \\
&= \widehat{G}_{reg}^{-1}(\widehat{g} - g) + \widehat{G}_{reg}^{-1}(G - \widehat{G})\nabla_G J(\theta) - \epsilon\widehat{G}_{reg}^{-1}G^{-1}\nabla_G J(\theta).
\end{aligned} \tag{37}$$

Let $\mathcal{E}$ denote the well-conditioned event: $\mathcal{E} = \left\{\left\|\widehat{G} - G\right\| < \frac{\lambda_{\min}}{2}\right\}$. On event $\mathcal{E}$, Weyl's inequality ensures $\lambda_{\min}(\widehat{G}) \geq \lambda_{\min} - \left\|\widehat{G} - G\right\| > \frac{\lambda_{\min}}{2}$, which implies

$$\left\|\widehat{G}_{reg}^{-1}\right\| \leq \frac{1}{\lambda_{\min}(\widehat{G}_{reg})} \leq \frac{1}{\lambda_{\min}(\widehat{G}) + \epsilon} < \frac{1}{\lambda_{\min}/2 + \epsilon} \leq \frac{2}{\lambda_{\min}}, \tag{38}$$

where the last inequality follows since $\epsilon > 0$. Now taking norms in (37) and applying the triangle inequality and sub-multiplicativity on $\mathcal{E}$, we get

$$\left\|\widehat{\nabla}_G J(\theta) - \nabla_G J(\theta)\right\| \leq \left\|\widehat{G}_{reg}^{-1}\right\|\|\widehat{g} - g\| + \left\|\widehat{G}_{reg}^{-1}\right\|\left\|G - \widehat{G}\right\|\|\nabla_G J(\theta)\| + \epsilon\left\|\widehat{G}_{reg}^{-1}\right\|\left\|G^{-1}\right\|\|\nabla_G J(\theta)\|.$$

Using the bounds $\left\|\widehat{G}_{reg}^{-1}\right\| \leq \frac{2}{\lambda_{\min}}$, $\left\|G^{-1}\right\| \leq \frac{1}{\lambda_{\min}}$, and $\|\nabla_G J(\theta)\| \leq M_{\nabla_G}$ (Remark 1), we get,

$$\left\|\widehat{\nabla}_G J(\theta) - \nabla_G J(\theta)\right\| \leq \frac{2}{\lambda_{\min}}\|\widehat{g} - g\| + \frac{2M_{\nabla_G}}{\lambda_{\min}}\left\|G - \widehat{G}\right\| + \frac{2\epsilon M_{\nabla_G}}{\lambda_{\min}^2}.$$

Squaring both sides and using the inequality $(a + b + c)^2 \leq 3(a^2 + b^2 + c^2)$:

$$\left\|\widehat{\nabla}_G J(\theta) - \nabla_G J(\theta)\right\|^2 \leq \frac{12}{\lambda_{\min}^2}\|\widehat{g} - g\|^2 + \frac{12M_{\nabla_G}^2}{\lambda_{\min}^2}\left\|G - \widehat{G}\right\|^2 + \frac{12\epsilon^2 M_{\nabla_G}^2}{\lambda_{\min}^4}. \tag{39}$$

We now take the total expectation and split it over events $\mathcal{E}$ and $\mathcal{E}^c$:

$$\mathbb{E}\left[\left\|\widehat{\nabla}_G J(\theta) - \nabla_G J(\theta)\right\|^2\right] = \mathbb{E}\left[\left\|\widehat{\nabla}_G J(\theta) - \nabla_G J(\theta)\right\|^2 I_{\mathcal{E}}\right] + \mathbb{E}\left[\left\|\widehat{\nabla}_G J(\theta) - \nabla_G J(\theta)\right\|^2 I_{\mathcal{E}^c}\right]. \tag{40}$$

We bound each term separately.

First we bound the error on the well-conditioned event $\mathcal{E}$. Taking expectation of inequality (39) over $\mathcal{E}$ and applying Lemmas 6 and 7, we get

$$
\begin{aligned}
\mathbb{E}\left[\left\|\widehat{\nabla}_G J(\theta) - \nabla_G J(\theta)\right\|^2 I_{\mathcal{E}}\right] &\leq \frac{12}{\lambda_{\min}^2}\mathbb{E}\left[\|\widehat{g} - g\|^2\right] + \frac{12M_{\nabla_G}^2}{\lambda_{\min}^2}\mathbb{E}\left[\left\|G - \widehat{G}\right\|^2\right] + \frac{12\epsilon^2 M_{\nabla_G}^2}{\lambda_{\min}^4} \\
&\leq \frac{12}{\lambda_{\min}^2}\left(\frac{C_\nabla}{N} + M_{\nabla_G}^2 \frac{C_G}{N}\right) + \frac{12\epsilon^2 M_{\nabla_G}^2}{\lambda_{\min}^4} \\
&= \frac{12}{\lambda_{\min}^2}\left(C_\nabla + M_{\nabla_G}^2 C_G\right)\frac{1}{N} + \frac{12\epsilon^2 M_{\nabla_G}^2}{\lambda_{\min}^4}.
\end{aligned}
\tag{41}
$$

Now we bound the error on the ill-conditioned event $\mathcal{E}^c$. Using Lemma 8, we have

$$
\mathbb{E}\left[\left\|\widehat{\nabla}_G J(\theta) - \nabla_G J(\theta)\right\|^2 I_{\mathcal{E}^c}\right] \leq B_\epsilon \cdot \mathbb{P}(\mathcal{E}^c).
\tag{42}
$$

Using Chebyshev's inequality and Lemma 6, we get

$$
\mathbb{P}(\mathcal{E}^c) = \mathbb{P}\left(\left\|\widehat{G} - G\right\| \geq \frac{\lambda_{\min}}{2}\right) \leq \frac{\mathbb{E}\left[\left\|\widehat{G} - G\right\|^2\right]}{(\lambda_{\min}/2)^2} \leq \frac{4C_G}{\lambda_{\min}^2 N}.
\tag{43}
$$

Substituting (43) into (42):

$$
\mathbb{E}\left[\left\|\widehat{\nabla}_G J(\theta) - \nabla_G J(\theta)\right\|^2 I_{\mathcal{E}^c}\right] \leq \frac{4B_\epsilon C_G}{\lambda_{\min}^2 N}.
\tag{44}
$$

Finally, the total expectation (40) can be bounded by substituting bounds (41) and (44) as follows

$$
\mathbb{E}\left[\left\|\widehat{\nabla}_G J(\theta) - \nabla_G J(\theta)\right\|^2\right] \leq \frac{12}{\lambda_{\min}^2}\left(C_\nabla + M_{\nabla_G}^2 C_G\right)\frac{1}{N} + \frac{12\epsilon^2 M_{\nabla_G}^2}{\lambda_{\min}^4} + \frac{4B_\epsilon C_G}{\lambda_{\min}^2 N}.
$$

$\square$

The above theorem establishes a non-asymptotic mean-squared error bound for the regularized natural-gradient estimator. It decomposes the estimation error into a stochastic variance term of order $O(N^{-1})$ and a deterministic bias term scaling as $O(\epsilon^2)$. The variance constant $C_{\text{gvar}}$ depends quadratically on $\lambda_{\min}^{-1}$, the inverse of the smallest eigenvalue of the Fisher information matrix, which explicitly quantifies the amplification of sampling noise during preconditioning induced by ill-conditioned policy curvature. Meanwhile, the bias component arises exclusively from the Tikhonov regularization $\epsilon I$, inducing an explicit bias–stability trade-off: regularization guarantees invertibility and numerical robustness at the cost of a fixed approximation error. Overall, the theorem identifies the implicit $O(N^{-1})$ Monte Carlo variance floor for natural-gradient estimation and provides a principled statistical baseline against which variance-reduction strategies can be assessed.

We now analyze the convergence behavior of natural policy gradient estimation when value functions are approximated using linear architectures. The following theorem characterizes the error dynamics under standard Monte Carlo sampling.

**Theorem 10.** *Let $w_{\text{true}} = G(\theta)^{-1}\nabla_\theta J(\theta) + G(\theta)^{-1}\mathbb{E}[\psi(\mathbf{s}, \mathbf{a})\varepsilon(\mathbf{s}, \mathbf{a}, \mathbf{s}')]$ be the biased natural gradient from Theorem 4, where $\varepsilon(s, a, s')$ is the Bellman error. Let $\widehat{w}_\epsilon = \widehat{G}_{\text{reg}}^{-1}\widehat{g}$ be the regularized empirical estimator with $\widehat{G}_{\text{reg}} = \widehat{G} + \epsilon I$, $\epsilon > 0$, using $N$ Monte Carlo samples. Under Assumptions 2–5, there exist $C_{wvar} > 0$ such that*

$$
\mathbb{E}\left[\|\widehat{w}_\epsilon - w_{\text{true}}\|^2\right] \leq \frac{C_{wvar}}{N} + \frac{24}{\lambda_{\min}^2}C_\varepsilon + \frac{12\epsilon^2 M_w^2}{\lambda_{\min}^2},
$$

*where*

$$C_{wvar} = \frac{24}{\lambda_{\min}^2}(C_\nabla + M_w^2 C_G) + \frac{4B_\epsilon C_G}{\lambda_{\min}^2}, \ \ with \ C_\varepsilon = \|\mathbb{E}[\boldsymbol{\psi}(\mathbf{s},\mathbf{a})\varepsilon(\mathbf{s},\mathbf{a},\mathbf{s}')]\|^2,$$

$$M_w = M_{\nabla_G} + \frac{\sqrt{C_\varepsilon}}{\lambda_{\min}}, \ \ and \ B_\epsilon = \Big(\frac{C_\psi C_Q}{\epsilon} + M_w\Big)^2.$$

*Proof.* We again let $G = G(\theta)$, $g = \nabla_\theta J(\theta)$, $b = \mathbb{E}[\boldsymbol{\psi}(\mathbf{s},\mathbf{a})\varepsilon(\mathbf{s},\mathbf{a},\mathbf{s}')]$, $C_\varepsilon = \|b\|^2$, $\widehat{g} = \widehat{\nabla} J(\theta)$, and $g_{\text{true}} = \nabla_\theta J(\theta) + \mathbb{E}[\boldsymbol{\psi}(\mathbf{s},\mathbf{a})\varepsilon(\mathbf{s},\mathbf{a},\mathbf{s}')]$. Then $g_{\text{true}} = g + b$, and $w_{\text{true}} = G^{-1}g_{\text{true}}$. We also define the regularized estimator as $\widehat{w}_\epsilon = \widehat{G}_{\text{reg}}^{-1}\widehat{g}$ with $\widehat{G}_{\text{reg}} = \widehat{G} + \epsilon I$.

We decomposition the error on $\widehat{w}_\epsilon$ as follows:

$$\widehat{w}_\epsilon - w_{\text{true}} = \widehat{G}_{\text{reg}}^{-1}\widehat{g} - G^{-1}g_{\text{true}}$$

$$= \widehat{G}_{\text{reg}}^{-1}(\widehat{g} - g_{\text{true}}) + \big(\widehat{G}_{\text{reg}}^{-1} - G^{-1}\big)g_{\text{true}}.$$

Using the identity $\widehat{G}_{\text{reg}}^{-1} - G^{-1} = \widehat{G}_{\text{reg}}^{-1}(G - \widehat{G}_{\text{reg}})G^{-1}$ and noting $G - \widehat{G}_{\text{reg}} = (G - \widehat{G}) - \epsilon I$, we obtain

$$\widehat{w}_\epsilon - w_{\text{true}} = \widehat{G}_{\text{reg}}^{-1}(\widehat{g} - g_{\text{true}}) + \widehat{G}_{\text{reg}}^{-1}(G - \widehat{G})w_{\text{true}} - \epsilon \widehat{G}_{\text{reg}}^{-1}w_{\text{true}}. \tag{45}$$

Similar to Theorem 9, we define the well-conditioned event $\mathcal{E} = \Big\{\|\widehat{G} - G\| < \frac{\lambda_{\min}}{2}\Big\}$. On event $\mathcal{E}$, we have $\lambda_{\min}(\widehat{G}) \geq \lambda_{\min}/2$ (by Assumption 5), which implies

$$\|\widehat{G}_{\text{reg}}^{-1}\| \leq \frac{2}{\lambda_{\min}}. \tag{46}$$

From Remark 1 and the definition of $b$, we have

$$\|w_{\text{true}}\| \leq \|G^{-1}g\| + \|G^{-1}b\| \leq M_{\nabla_G} + \frac{\|b\|}{\lambda_{\min}} = M_{\nabla_G} + \frac{\sqrt{C_\varepsilon}}{\lambda_{\min}} =: M_w. \tag{47}$$

Taking norms in (45) and applying triangle inequality and using (46–47), we obtain on $\mathcal{E}$

$$\|\widehat{w}_\epsilon - w_{\text{true}}\| \leq \frac{2}{\lambda_{\min}}\|\widehat{g} - g_{\text{true}}\| + \frac{2}{\lambda_{\min}}\|G - \widehat{G}\| M_w + \frac{2\epsilon}{\lambda_{\min}}M_w. \tag{48}$$

Now by definition, we have $\|\widehat{g} - g_{\text{true}}\| \leq \|\widehat{g} - g\| + \|b\|$ and $\|b\| = \sqrt{C_\varepsilon}$. By applying the inequality $(a + b + c)^2 \leq 3(a^2 + b^2 + c^2)$ on (48), we obtain

$$\|\widehat{w}_\epsilon - w_{\text{true}}\|^2 \leq \frac{12}{\lambda_{\min}^2}\big(\|\widehat{g} - g\| + \sqrt{C_\varepsilon}\big)^2 + \frac{12M_w^2}{\lambda_{\min}^2}\|G - \widehat{G}\|^2 + \frac{12\epsilon^2 M_w^2}{\lambda_{\min}^2}$$

$$\leq \frac{24}{\lambda_{\min}^2}\|\widehat{g} - g\|^2 + \frac{24C_\varepsilon}{\lambda_{\min}^2} + \frac{12M_w^2}{\lambda_{\min}^2}\|G - \widehat{G}\|^2 + \frac{12\epsilon^2 M_w^2}{\lambda_{\min}^2}, \tag{49}$$

where we used $(x + y)^2 \leq 2(x^2 + y^2)$. By taking the expectation of (49) conditioned on $\mathcal{E}$ and applying Lemmas 6 and 7, we get

$$\mathbb{E}\big[\|\widehat{w}_\epsilon - w_{\text{true}}\|^2\mathbf{1}_\mathcal{E}\big] \leq \frac{24C_\nabla}{\lambda_{\min}^2 N} + \frac{24C_\varepsilon}{\lambda_{\min}^2} + \frac{12M_w^2 C_G}{\lambda_{\min}^2 N} + \frac{12\epsilon^2 M_w^2}{\lambda_{\min}^2}. \tag{50}$$

Now we bound the expectation on the complement event $\mathcal{E}^c$. Since $\widehat{G}_{\text{reg}} \succeq \epsilon I$, we have $\|\widehat{G}_{\text{reg}}^{-1}\| \leq 1/\epsilon$. Moreover, $\|\widehat{g}\| \leq C_\psi C_Q$ (from (34)). Hence

$$\|\widehat{w}_\epsilon\| \leq \frac{C_\psi C_Q}{\epsilon}, \qquad \|\widehat{w}_\epsilon - w_{\text{true}}\| \leq \frac{C_\psi C_Q}{\epsilon} + M_w =: \sqrt{B_\epsilon}.$$

Thus $\|\widehat{w}_\epsilon - w_{\text{true}}\|^2 \le B_\epsilon$ almost surely. Now by appealing to Chebyshev's Inequality and Lemma 6,

$$\mathbb{P}(\mathcal{E}^c) = \mathbb{P}\big(\|\widehat{G} - G\| \ge \lambda_{\min}/2\big) \le \frac{4\,\mathbb{E}[\|\widehat{G} - G\|^2]}{\lambda_{\min}^2} \le \frac{4C_G}{\lambda_{\min}^2 N}.$$

Consequently,

$$\mathbb{E}\big[\|\widehat{w}_\epsilon - w_{\text{true}}\|^2 \mathbf{1}_{\mathcal{E}^c}\big] \le B_\epsilon\,\mathbb{P}(\mathcal{E}^c) \le \frac{4B_\epsilon C_G}{\lambda_{\min}^2 N}. \tag{51}$$

Finally, by adding (50) and (51) gives the final bound.

$$\mathbb{E}\big[\|\widehat{w}_\epsilon - w_{\text{true}}\|^2\big] \le \frac{1}{N}\Big(\frac{24C_\nabla}{\lambda_{\min}^2} + \frac{12M_w^2 C_G}{\lambda_{\min}^2} + \frac{4B_\epsilon C_G}{\lambda_{\min}^2}\Big) + \frac{24C_\varepsilon}{\lambda_{\min}^2} + \frac{12\epsilon^2 M_w^2}{\lambda_{\min}^2}.$$

$\square$

Theorem 10 extends the analysis to the practical setting where value functions are approximated, introducing an additional, irreducible error source: the score-projected Bellman residual $C_\varepsilon = |\mathbb{E}[\boldsymbol{\psi}(s,a)\varepsilon(s,a,s')]|^2$. This constant bias term persists even with infinite samples and zero regularization, representing an implicit limit on policy improvement achievable with a given function class. The geometry of this bias is particularly revealing: state-dependent approximation errors vanish under projection due to the score function property $\mathbb{E}[\boldsymbol{\psi}|s] = 0$, while action-dependent errors survive and systematically skew policy updates along directions that correlate with policy sensitivity. The combined error bound demonstrates that both sampling variance and approximation bias are amplified by $\lambda_{\min}^{-1}$, creating a three-way trade-off between sample efficiency ($O(1/N)$), numerical stability ($\epsilon^2$), and representation power ($C_\varepsilon$). Thus, the above two results provide the statistical foundation for understanding why natural actor-critic methods can struggle with high-variance gradients while simultaneously explaining the effectiveness of techniques that address either sampling efficiency (through variance reduction) or representation quality (through richer function approximators).

## 4.3 Array-RQMC Analysis

We now analyze the corresponding estimators when trajectory generation is driven by Array–RQMC which offers more structured sampling, with particular emphasis on how structured dependence across sample trajectories propagates through the natural actor–critic linear system. To establish the theoretical foundation for the variance reduction achieved by randomized quasi-Monte Carlo methods within our framework, we now present a central result on the mean-squared error of estimators constructed from scrambled $(t, m, d)$-nets. This theorem (Owen, 1997b;a) provides a rigorous non-asymptotic bound on the integration error when a deterministic low-discrepancy point set is randomized via nested uniform scrambling and a digital shift. The bound explicitly relates the error to the Hardy–Krause variation of the integrand, denoted $V_{\text{HK}}(f)$. Formally, for a function $f : [0,1)^d \to \mathbb{R}$, let $\partial^u f$ denote the mixed partial derivative with respect to the coordinates in $u \subseteq \{1, \ldots, d\}$, and define the Vitali variation of $f$ on a $k$-dimensional face as the integral of $|\partial^u f|$ over that face. The Hardy–Krause variation is then defined as:

$$V_{\text{HK}}(f) = \sum_{\emptyset \ne u \subseteq \{1,\ldots,d\}} \int_{[0,1]^{|u|}} |\partial^u f(x_u, \mathbf{1}_{-u})|\; dx_u,$$

where $(x_u, \mathbf{1}_{-u})$ denotes the point in $[0,1]^d$ with coordinates in $u$ set to $x_u$ and the remaining coordinates set to 1 (Owen, 1997a). For functions with $V_{\text{HK}}(f) < \infty$, the theorem shows that the RQMC estimator achieves a mean-squared error of order $O((\log N)^{2d}/N^2)$, a strict improvement over the standard Monte Carlo rate of $O(1/N)$. This result is critical for our subsequent analysis, as it formally guarantees that the RQMC-based estimators for the Fisher information matrix and the policy gradient—components of the natural actor-critic linear system—inherit this accelerated convergence, thereby directly reducing the variance propagated into the natural-gradient update.

**Theorem 11.** *(Owen, 1997a; 2013) Let $\{z_i\}_{i=1}^N \subset [0,1)^d$ be a randomly scrambled $(t,m,d)$-net in base $b$, with $N = b^m$, constructed via Owen's nested uniform scrambling (optionally followed by a random digital shift). Let $f : [0,1)^d \to \mathbb{R}$ be integrable with finite Hardy–Krause variation $V_{\mathrm{HK}}(f) < \infty$, and define*

$$\widehat{I}_N := \frac{1}{N} \sum_{i=1}^N f(z_i), \qquad I := \int_{[0,1)^d} f(z)\,dz.$$

*Then $\mathbb{E}[\widehat{I}_N] = I$ and there exists a constant $c_d > 0$ (depending only on $d$ and the net construction) such that*

$$\mathbb{E}\big[(\widehat{I}_N - I)^2\big] \;\leq\; c_d\, V_{\mathrm{HK}}(f)^2\, \frac{(\log N)^{2d}}{N^2}.$$

We now apply this result to bound the estimation errors of the Fisher information matrix and the policy gradient within the natural actor-critic framework. Specifically, we consider the Array-RQMC estimators $\widehat{G}$ and $\widehat{g}$ defined in (27), which are constructed by averaging over parallel trajectories generated via low-discrepancy point sets.

**Lemma 12.** *Under Assumptions 2–5, the Array–RQMC estimators $\widehat{G}$ and $\widehat{g}$ in (27) satisfy*

$$\mathbb{E}\big[\|\widehat{G} - G\|^2\big] \leq C_G \frac{(\log N)^{2d}}{N^2}, \qquad \mathbb{E}\big[\|\widehat{g} - g\|^2\big] \leq C_g \frac{(\log N)^{2d}}{N^2},$$

*where $d = d_s + d_a$ is the total dimension of the uniforms driving transition and action selection, and $C_G, C_g > 0$ depend only on problem constants (in particular on $C_\psi, C_r, \gamma, |\mathcal{S}|, |\mathcal{A}|$) and the net construction.*

*Proof.* We first prove the bound for $\widehat{G}$; the argument for $\widehat{g}$ is identical. Fix a time step $j$ and let $\mathcal{F}_j$ be the $\sigma$-field generated by all states/actions up to time $j$ and all scrambled nets up to time $j-1$. Conditioning on $\mathcal{F}_j$ freezes the current array $\{(s_{i,j}, a_{i,j})\}_{i=1}^N$. Now given $(s_{i,j}, a_{i,j})$, the next pair $(s_{i,j+1}, a_{i,j+1})$ is generated by inverse transforms $s_{i,j+1} = \Lambda(s_{i,j}, a_{i,j}, u)$, $a_{i,j+1} = \Gamma(s_{i,j+1}, v)$. Let $z = (u,v) \in [0,1)^d$. Because $\mathcal{S}, \mathcal{A}$ are finite, both $\Lambda$ and $\Gamma$ are inverse-CDF maps for discrete distributions with finite support; hence $[0,1)^d$ can be partitioned into finitely many axis-aligned boxes on which $(s_{i,j+1}(z), a_{i,j+1}(z))$ is constant. Consequently, for each $(k, \ell)$,

$$f_{i,j}^{k\ell}(z) := \psi_k(s_{i,j}, a_{i,j})\psi_\ell(s_{i,j}, a_{i,j})$$

is a bounded step function with finitely many axis-aligned discontinuities, and thus has finite Hardy–Krause variation $V_{\mathrm{HK}}(f_{i,j}^{k\ell}) < \infty$ (Niederreiter, 1978).

Owen's scrambled-net result gives a mean-squared error bound of order $(\log N)^{2d}/N^2$ for estimating an integral of a function with finite Hardy–Krause variation by averaging the function over the scrambled net points. However, the estimator is not average of one function $f$ over the net. Instead, it is the average of many functions $f_1, \ldots, f_N$, where $f_i$ depends on $(s_{i,j}, a_{i,j})$, evaluated at corresponding net points $z_{i,j}$. So we need a "multi-integrand" extension. Let $\{z_{i,j}\}_{i=1}^N$ denote the scrambled $(t,d)$-net used at time $j$ (after the state-dependent permutation; a permutation does not change the empirical average). We define the following centered errors:

$$\Delta_i := f_{i,j}^{k\ell}(z_{i,j}) - \int_{[0,1)^d} f_{i,j}^{k\ell}(z)\,dz.$$

Then

$$(\widehat{G}_{k\ell} - G_{k\ell})^2 = \left(\frac{1}{N}\sum_{i=1}^N \Delta_i\right)^2 = \frac{1}{N^2}\sum_{i=1}^N \Delta_i^2 + \frac{2}{N^2}\sum_{i<r} \Delta_i \Delta_r.$$

By taking conditional expectation given the $\sigma$-field $\mathcal{F}_j$ and applying Cauchy–Schwarz inequality, we get $\mathbb{E}[\Delta_i \Delta_r \mid \mathcal{F}_j] \leq \sqrt{\mathbb{E}[\Delta_i^2 \mid \mathcal{F}_j]\mathbb{E}[\Delta_r^2 \mid \mathcal{F}_j]}$. Moreover, conditional on $\mathcal{F}_j$, each $f_{i,j}^{k\ell}$ is deterministic with finite $V_{\mathrm{HK}}$, so Theorem 11 implies

$$\mathbb{E}[\Delta_i^2 \mid \mathcal{F}_j] \leq c_d\, V_{\mathrm{HK}}(f_{i,j}^{k\ell})^2\, \frac{(\log N)^{2d}}{N^2}.$$

Now combining all of the above and summing the diagonal and off-diagonal terms gives

$$\mathbb{E}\Big[(\widehat{G}_{k\ell} - G_{k\ell})^2 \mid \mathcal{F}_j\Big] \le c_d \left(\frac{1}{N}\sum_{i=1}^{N} V_{\mathrm{HK}}(f_{i,j}^{k\ell})^2\right) \frac{(\log N)^{2d}}{N^2},$$

where we used $(\sum_{i=1}^{N} a_i)^2 \le N \sum_{i=1}^{N} a_i^2$ with $a_i = V_{\mathrm{HK}}(f_{i,j}^{k\ell})$. Since $\mathcal{S} \times \mathcal{A}$ is finite, the collection of possible functions $\{f_{i,j}^{k\ell}(\cdot)\}$ over all $(s_{i,j}, a_{i,j})$ is finite, and therefore

$$\bar{V}_{k\ell}^2 := \sup_{(s,a)\in\mathcal{S}\times\mathcal{A}} V_{\mathrm{HK}}(f_{s,a}^{k\ell})^2 < \infty.$$

Hence $\frac{1}{N}\sum_{i=1}^{N} V_{\mathrm{HK}}(f_{i,j}^{k\ell})^2 \le \bar{V}_{k\ell}^2$ uniformly over $j$, which yields

$$\mathbb{E}\Big[(\widehat{G}_{k\ell} - G_{k\ell})^2 \mid \mathcal{F}_j\Big] \le c_d \, \bar{V}_{k\ell}^2 \, \frac{(\log N)^{2d}}{N^2}.$$

Summing over $(k,\ell)$ and using $\|\cdot\|^2 \le \|\cdot\|_F^2$ gives

$$\mathbb{E}\Big[\|\widehat{G} - G\|^2 \mid \mathcal{F}_j\Big] \le c_d \Big(\sum_{k,\ell} \bar{V}_{k\ell}^2\Big) \frac{(\log N)^{2d}}{N^2} =: C_G \frac{(\log N)^{2d}}{N^2}.$$

Finally, take expectation over $\mathcal{F}_j$. Averaging over the $T$ steps preserves the bound because the scrambled nets are independent across $j$ and the bound is uniform in $j$. The identical argument applied to $\widehat{g}$ yields the analogous bound with constant $C_g$. □

We now derive a finite-sample error bound for the resulting regularized natural gradient estimator. This theorem quantifies the combined effect of low-discrepancy sampling and Tikhonov regularization on the accuracy of natural gradient updates. Specifically, we analyze the estimator $\widehat{G}_{\mathrm{reg}}^{-1}\widehat{\nabla}J(\theta)$, where $\widehat{G}_{\mathrm{reg}} = \widehat{G} + \epsilon I$ with $\epsilon > 0$.

**Theorem 13.** *Under Assumptions 2–5 and for the Tikhonov regularized Fisher matrix, $\widehat{G}_{\mathrm{reg}} = \widehat{G} + \epsilon I, \ \epsilon > 0$, we have the following:*

$$\mathbb{E}\Big[\big\|\widehat{G}_{\mathrm{reg}}^{-1}\widehat{\nabla}J(\theta) - \nabla_G J(\theta)\big\|^2\Big] \le C_{\mathrm{rqvar}} \cdot \frac{(\log N)^{2d}}{N^2} + \frac{12 M_{\nabla_G}^2}{\lambda_{\min}^4} \cdot \epsilon^2,$$

*where*

$$C_{\mathrm{rqvar}} = \frac{12}{\lambda_{\min}^2}\big(C_g + M_{\nabla_G}^2 C_G\big) + \frac{4\tilde{B}(\epsilon)C_G}{\lambda_{\min}^2}, \quad and \quad \tilde{B}(\epsilon) = \Big(\frac{C_\psi C_{\mathcal{Q}}}{\epsilon} + M_{\nabla_G}\Big)^2.$$

*Proof.* Let $\widehat{\nabla}_{G,\epsilon}J(\theta) = \widehat{G}_{\mathrm{reg}}^{-1}\widehat{\nabla}J(\theta)$, $G = G(\theta)$, $g = \nabla J(\theta)$, $\widehat{g} = \widehat{\nabla}J(\theta)$, and denote the (unregularized) natural gradient by $w = G^{-1}g = \nabla_G J(\theta)$.

We decompose the estimation error as follows:

$$\widehat{\nabla}_{G,\epsilon}J(\theta) - w = \widehat{G}_{\mathrm{reg}}^{-1}\widehat{g} - G^{-1}g = \underbrace{\widehat{G}_{\mathrm{reg}}^{-1}(\widehat{g} - g)}_{\text{gradient estimation error}} + \underbrace{(\widehat{G}_{\mathrm{reg}}^{-1} - G^{-1})g}_{\text{matrix inversion error}}. \tag{52}$$

Using the identity $A^{-1} - B^{-1} = A^{-1}(B - A)B^{-1}$ with $A = \widehat{G}_{\mathrm{reg}}$ and $B = G$, we get

$$\widehat{G}_{\mathrm{reg}}^{-1} - G^{-1} = \widehat{G}_{\mathrm{reg}}^{-1}(G - \widehat{G}_{\mathrm{reg}})G^{-1}.$$

Applying $G - \widehat{G}_{\mathrm{reg}} = G - (\widehat{G} + \epsilon I) = (G - \widehat{G}) - \epsilon I$ we get

$$\widehat{G}_{\mathrm{reg}}^{-1} - G^{-1} = \widehat{G}_{\mathrm{reg}}^{-1}(G - \widehat{G})G^{-1} - \epsilon \widehat{G}_{\mathrm{reg}}^{-1}G^{-1}. \tag{53}$$

Now from (52) and (53), we get

$$\widehat{\nabla}_{G,\epsilon}J(\theta) - w = \widehat{G}_{\text{reg}}^{-1}(\widehat{g} - g) + \widehat{G}_{\text{reg}}^{-1}(G - \widehat{G})w - \epsilon\widehat{G}_{\text{reg}}^{-1}G^{-1}w.$$

Taking norms and applying the triangle inequality, we obtain

$$\left\|\widehat{\nabla}_{G,\epsilon}J(\theta) - w\right\| \le \|\widehat{G}_{\text{reg}}^{-1}\| \|\widehat{g} - g\| + \|\widehat{G}_{\text{reg}}^{-1}\| \|G - \widehat{G}\| \|w\| + \epsilon \|\widehat{G}_{\text{reg}}^{-1}\| \|G^{-1}\| \|w\|. \tag{54}$$

Similar to Theorem 9, we now define the event $\mathcal{E} = \left\{\|\widehat{G} - G\| < \lambda_{\min}/2\right\}$. On $\mathcal{E}$, we have $\|\widehat{G}_{\text{reg}}^{-1}\| \le \frac{2}{\lambda_{\min}}$. On $\mathcal{E}$, by applying Assumption 5 and Remark 1 to (54), we get

$$\left\|\widehat{\nabla}_{G,\epsilon}J(\theta) - w\right\| \le \frac{2}{\lambda_{\min}}\|\widehat{g} - g\| + \frac{2M_{\nabla_G}}{\lambda_{\min}}\|G - \widehat{G}\| + \frac{2\epsilon M_{\nabla_G}}{\lambda_{\min}^2}.$$

Apply the inequality $(a + b + c)^2 \le 3(a^2 + b^2 + c^2)$ to the above, we obtain

$$\left\|\widehat{\nabla}_{G,\epsilon}J(\theta) - w\right\|^2 \le \frac{12}{\lambda_{\min}^2}\|\widehat{g} - g\|^2 + \frac{12M_{\nabla_G}^2}{\lambda_{\min}^2}\|G - \widehat{G}\|^2 + \frac{12\epsilon^2 M_{\nabla_G}^2}{\lambda_{\min}^4}.$$

Taking expectations and invoking Lemma 12, we get,

$$\mathbb{E}\left[\left\|\widehat{\nabla}_{G,\epsilon}J(\theta) - w\right\|^2 \mathbf{1}_{\mathcal{E}}\right] \le \frac{12}{\lambda_{\min}^2}\left(C_g + M_{\nabla_G}^2 C_G\right)\frac{(\log N)^{2d}}{N^2} + \frac{12\epsilon^2 M_{\nabla_G}^2}{\lambda_{\min}^4}. \tag{55}$$

Now we bound the expectation on $\mathcal{E}^c$. By Chebyshev's inequality and Lemma 12, we obtain

$$\mathbb{P}(\mathcal{E}^c) = \mathbb{P}\left(\|G - \widehat{G}\| \ge \lambda_{\min}/2\right) \le \frac{\mathbb{E}[\|G - \widehat{G}\|^2]}{(\lambda_{\min}/2)^2} \le \frac{4C_G}{\lambda_{\min}^2}\frac{(\log N)^{2d}}{N^2}. \tag{56}$$

Moreover, since $\widehat{G}_{\text{reg}} \succeq \epsilon I$, we have $\|\widehat{G}_{\text{reg}}^{-1}\| \le 1/\epsilon$, and by Assumption 2, $\|\widehat{g}\| \le C_\psi C_Q$ (from (34)). Hence

$$\|\widehat{\nabla}_{G,\epsilon}J(\theta)\| = \|\widehat{G}_{\text{reg}}^{-1}\widehat{g}\| \le \|\widehat{G}_{\text{reg}}^{-1}\|\|\widehat{g}\| \le \frac{C_\psi C_Q}{\epsilon}.$$

Therefore,

$$\left\|\widehat{\nabla}_{G,\epsilon}J(\theta) - w\right\| \le \frac{C_\psi C_Q}{\epsilon} + M_{\nabla_G} \qquad \implies \qquad \left\|\widehat{\nabla}_{G,\epsilon}J(\theta) - w\right\|^2 \le \tilde{B}(\epsilon), \tag{57}$$

where $\tilde{B}(\epsilon) := \left(\frac{C_\psi C_Q}{\epsilon} + M_{\nabla_G}\right)^2$. Thus, by taking expectation over $\mathcal{E}^c$ on (57), and applying (56), we obtain

$$\mathbb{E}\left[\left\|\widehat{\nabla}_{G,\epsilon}J(\theta) - w\right\|^2 \mathbf{1}_{\mathcal{E}^c}\right] \le \tilde{B}(\epsilon)\,\mathbb{P}(\mathcal{E}^c) \le \frac{4\tilde{B}(\epsilon)C_G}{\lambda_{\min}^2}\frac{(\log N)^{2d}}{N^2}. \tag{58}$$

Finally, by combining (55) and (58), we obtain the final bound as follows:

$$\mathbb{E}\left[\left\|\widehat{\nabla}_{G,\epsilon}J(\theta) - w\right\|^2\right] \le \left[\frac{12}{\lambda_{\min}^2}\left(C_g + M_{\nabla_G}^2 C_G\right) + \frac{4\tilde{B}(\epsilon)C_G}{\lambda_{\min}^2}\right]\frac{(\log N)^{2d}}{N^2} + \frac{12\epsilon^2 M_{\nabla_G}^2}{\lambda_{\min}^4}.$$

$$\square$$

The above result reveals two fundamentally distinct components: a *variance term* scaling as $O((\log N)^{2d}/N^2)$ and a *bias term* scaling as $O(\epsilon^2)$. The variance term inherits the accelerated convergence rate characteristic of scrambled $(t, m, d)$-nets, demonstrating that the structured dependence introduced by Array-RQMC effectively suppresses the stochastic noise in estimating both the Fisher information matrix $G(\theta)$ and the policy gradient $\nabla J(\theta)$. Crucially, this $O((\log N)^{2d}/N^2)$ rate represents a strict improvement over the canonical

Monte Carlo rate of $O(1/N)$, with the exponent 2 reflecting the quadratic error reduction from quasi-Monte Carlo integration of functions with finite Hardy–Krause variation. The bias term arises solely from the Tikhonov regularization $\epsilon I$, which guarantees numerical invertibility but introduces a deterministic perturbation that persists even as $N \to \infty$. The presence of $\lambda_{\min}^{-4}$ in the bias constant explicitly quantifies how ill-conditioned curvature (small $\lambda_{\min}$) amplifies regularization-induced distortion during preconditioning. This result therefore delineates a clear bias–variance trade-off: while smaller $\epsilon$ reduces approximation bias, it may compromise numerical stability; conversely, increasing $N$ reduces stochastic variance without affecting the bias floor. It also confirms that the geometric preconditioning inherent to natural gradients is fully compatible with the variance-reduction benefits of low-discrepancy sampling.

We now state the finite-sample error bound for the regularized Array-RQMC natural actor-critic estimator under linear function approximation.

**Theorem 14.** *Under Assumptions 2–5 and the conditions of Lemma 12, let $w_{\text{true}} = G(\theta)^{-1}\nabla_\theta J(\theta) + G(\theta)^{-1}\mathbb{E}[\psi(s,a)\varepsilon(s,a,s')]$ be the biased natural gradient, where $\varepsilon(s,a,s')$ is the Bellman error, and let $\widehat{G}_{\text{reg}} = \widehat{G} + \epsilon I$ with $\epsilon > 0$. Define the regularized Array-RQMC estimator $\widehat{w}_{\text{ARQMC}} = \widehat{G}_{\text{reg}}^{-1}\widehat{\nabla}J(\theta)$. Then there exist constants $C_{\text{arqmc}} > 0$, $C_\varepsilon > 0$ and $M_w > 0$ such that*

$$\mathbb{E}\big[\|\widehat{w}_{\text{ARQMC}} - w_{\text{true}}\|^2\big] \;\leq\; C_{\text{arqmc}}\frac{(\log N)^{2d}}{N^2} \;+\; \frac{16}{\lambda_{\min}^2}C_\varepsilon \;+\; \frac{16\epsilon^2 M_w^2}{\lambda_{\min}^2},$$

*where $d$ is the total dimension of the uniforms driving the transitions and actions, $\lambda_{\min}$ is the minimal eigenvalue of $G(\theta)$, $C_\varepsilon = \|\mathbb{E}[\psi(\mathbf{s},\mathbf{a})\varepsilon(\mathbf{s},\mathbf{a},\mathbf{s}')]\|^2$, and $M_w = M_{\nabla_G} + \sqrt{C_\varepsilon}/\lambda_{\min}$ with $M_{\nabla_G}$ the bound on the true natural gradient (Remark 1).*

*Proof.* For brevity, we denote $G = G(\theta)$, $g = \nabla_\theta J(\theta)$, $b = \mathbb{E}[\psi(\mathbf{s},\mathbf{a})\varepsilon(\mathbf{s},\mathbf{a},\mathbf{s}')]$, and $g_{\text{true}} = g + b$. Thus $w_{\text{true}} = G^{-1}g_{\text{true}}$.

We first decompose the error on $\widehat{w}_{\text{ARQMC}}$ as follows:

$$\widehat{w}_{\text{ARQMC}} - w_{\text{true}} = \widehat{G}_{\text{reg}}^{-1}(\widehat{g} - g_{\text{true}}) + (\widehat{G}_{\text{reg}}^{-1} - G^{-1})g_{\text{true}}.$$

Using the identity $\widehat{G}_{\text{reg}}^{-1} - G^{-1} = \widehat{G}_{\text{reg}}^{-1}(G - \widehat{G}_{\text{reg}})G^{-1}$ and $G - \widehat{G}_{\text{reg}} = (G - \widehat{G}) - \epsilon I$, we rewrite the second term as

$$\begin{aligned}(\widehat{G}_{\text{reg}}^{-1} - G^{-1})g_{\text{true}} &= \widehat{G}_{\text{reg}}^{-1}(G - \widehat{G})G^{-1}g_{\text{true}} - \epsilon\widehat{G}_{\text{reg}}^{-1}G^{-1}g_{\text{true}} \\ &= \widehat{G}_{\text{reg}}^{-1}(G - \widehat{G})w_{\text{true}} - \epsilon\widehat{G}_{\text{reg}}^{-1}w_{\text{true}}.\end{aligned}$$

Hence

$$\widehat{w}_{\text{ARQMC}} - w_{\text{true}} = \widehat{G}_{\text{reg}}^{-1}(\widehat{g} - g_{\text{true}}) + \widehat{G}_{\text{reg}}^{-1}(G - \widehat{G})w_{\text{true}} - \epsilon\widehat{G}_{\text{reg}}^{-1}w_{\text{true}}. \tag{59}$$

Similar to Theorem 9, we now define the event $\mathcal{E} = \{\|\widehat{G} - G\| \leq \lambda_{\min}/2\}$. On $\mathcal{E}$, we have $\lambda_{\min}(\widehat{G}_{\text{reg}}) \geq \lambda_{\min}/2$, so that

$$\|\widehat{G}_{\text{reg}}^{-1}\| \leq \frac{2}{\lambda_{\min}}. \tag{60}$$

By Assumption 5, $\|G^{-1}\| \leq 1/\lambda_{\min}$. From Remark 1 and the definition of $b$, we have

$$\|w_{\text{true}}\| \leq \|G^{-1}g\| + \|G^{-1}b\| \leq M_{\nabla_G} + \frac{\|b\|}{\lambda_{\min}} = M_{\nabla_G} + \frac{\sqrt{C_\varepsilon}}{\lambda_{\min}} =: M_w. \tag{61}$$

Taking norms in (59) and using (60)–(61),

$$\|\widehat{w}_{\text{ARQMC}} - w_{\text{true}}\| \leq \|\widehat{G}_{\text{reg}}^{-1}\|\,\|\widehat{g} - g_{\text{true}}\| + \|\widehat{G}_{\text{reg}}^{-1}\|\,\|G - \widehat{G}\|\,\|w_{\text{true}}\| + \epsilon\,\|\widehat{G}_{\text{reg}}^{-1}\|\,\|w_{\text{true}}\|$$

$$\leq \frac{2}{\lambda_{\min}}\big(\|\widehat{g} - g\| + \|b\|\big) + \frac{2M_w}{\lambda_{\min}}\|G - \widehat{G}\| + \frac{2\epsilon M_w}{\lambda_{\min}}.$$

Using identity $(a + b + c)^2 \leq 4(a^2 + b^2 + c^2)$ and applying Cauchy–Schwarz give

$$\|\widehat{w}_{\text{ARQMC}} - w_{\text{true}}\|^2 \leq \frac{16}{\lambda_{\min}^2}\|\widehat{g} - g\|^2 + \frac{16}{\lambda_{\min}^2}\|b\|^2 + \frac{16M_w^2}{\lambda_{\min}^2}\|G - \widehat{G}\|^2 + \frac{16\epsilon^2 M_w^2}{\lambda_{\min}^2}.$$

Now taking expectation conditioned on $\mathcal{E}$ and invoking Lemma 12, we get,

$$\mathbb{E}\big[\|\widehat{w}_{\text{ARQMC}} - w_{\text{true}}\|^2 \mathbf{1}_\mathcal{E}\big] \leq \frac{16}{\lambda_{\min}^2}C_g\frac{(\log N)^{2d}}{N^2} + \frac{16}{\lambda_{\min}^2}C_\varepsilon + \frac{16M_w^2}{\lambda_{\min}^2}C_G\frac{(\log N)^{2d}}{N^2} + \frac{16\epsilon^2 M_w^2}{\lambda_{\min}^2}. \tag{62}$$

Now we bound on the complement event $\mathcal{E}^c$. Similarly as in Theorem 10, we have $\|\widehat{G}_{\text{reg}}^{-1}\| \leq 1/\epsilon$ and $\|\widehat{g}\| \leq C_\psi C_\mathcal{Q}$. Thus

$$\|\widehat{w}_{\text{ARQMC}}\| = \|\widehat{G}_{\text{reg}}^{-1}\widehat{g}\| \leq \frac{C_\psi C_Q}{\epsilon}.$$

Together with (61) we obtain the following almost-sure bound

$$\|\widehat{w}_{\text{ARQMC}} - w_{\text{true}}\| \leq \frac{C_\psi C_Q}{\epsilon} + M_w =: B_\epsilon. \tag{63}$$

By Chebyshev's inequality and Lemma 12, we get

$$\mathbb{P}(\mathcal{E}^c) = \mathbb{P}\Big(\|\widehat{G} - G\| > \lambda_{\min}/2\Big) \leq \frac{4\,\mathbb{E}[\|\widehat{G} - G\|^2]}{\lambda_{\min}^2} \leq \frac{4C_G}{\lambda_{\min}^2}\frac{(\log N)^{2d}}{N^2}. \tag{64}$$

Hence,

$$\mathbb{E}\big[\|\widehat{w}_{\text{ARQMC}} - w_{\text{true}}\|^2 \mathbf{1}_{\mathcal{E}^c}\big] \leq B_\epsilon^2\,\mathbb{P}(\mathcal{E}^c) \leq \frac{4B_\epsilon^2 C_G}{\lambda_{\min}^2}\frac{(\log N)^{2d}}{N^2}. \tag{65}$$

Finally, by combining the expectations on $\mathcal{E}$ (62) and $\mathcal{E}^c$ (65), we get the required bound.

$$\mathbb{E}\big[\|\widehat{w}_{\text{ARQMC}} - w_{\text{true}}\|^2\big] \leq \left[\frac{16}{\lambda_{\min}^2}\big(C_g + M_w^2 C_G\big) + \frac{4B_\epsilon^2 C_G}{\lambda_{\min}^2}\right]\frac{(\log N)^{2d}}{N^2} + \frac{16}{\lambda_{\min}^2}C_\varepsilon + \frac{16\epsilon^2 M_w^2}{\lambda_{\min}^2}.$$

$\square$

The above result provides a complete non-asymptotic decomposition of the mean-squared error of the regularized Array-RQMC natural-actor-critic estimator. The bound explicitly separates three distinct sources of inaccuracy: a variance term scaling as $C_{\text{arqmc}}(\log N)^{2d}/N^2$, an irreducible approximation bias $\frac{16}{\lambda_{\min}^2}C_\varepsilon$, and a regularization bias $\frac{16\epsilon^2 M_w^2}{\lambda_{\min}^2}$. The variance term reflects the stochastic error induced by finite-sample estimation of the Fisher matrix and the policy gradient. The rate $O((\log N)^{2d}/N^2)$ is strictly faster than the canonical Monte-Carlo rate $O(1/N)$ and is inherited from the scrambled-net construction which arises from the quadratic error reduction of quasi-Monte-Carlo integration for functions of bounded Hardy–Krause variation. This acceleration is the core benefit of employing Array-RQMC: by replacing independent pseudo-random points with a low-discrepancy set, the estimator achieves a quadratically smaller variance as a function of the number $N$ of parallel chains. The approximation bias term originates from the projection of the Bellman residual $\varepsilon(s, a, s')$ onto the score-feature space, i.e., $C_\varepsilon = \|\mathbb{E}[\psi(\mathbf{s}, \mathbf{a})\varepsilon(\mathbf{s}, \mathbf{a}, \mathbf{s}')]\|^2$. This bias does not vanish with increased sampling or decreased regularization; it is a fundamental limitation of the chosen function class. The factor $\lambda_{\min}^{-2}$ indicates that ill-conditioned curvature (a small minimal Fisher eigenvalue) amplifies the effect of approximation errors, making the natural-gradient direction more sensitive to inaccuracies in the critic. The regularization bias term arises from the Tikhonov regularization $\epsilon I$ added to ensure numerical invertibility of the empirical Fisher matrix. Like the approximation bias, it is scaled by $\lambda_{\min}^{-2}$, again highlighting the vulnerability of preconditioned updates to poor conditioning. This term represents a deliberate bias–stability trade-off: a larger $\epsilon$ guarantees invertibility and controls the norm of $\widehat{G}_{\text{reg}}^{-1}$, but introduces a persistent bias that cannot be eliminated by collecting more samples. Collectively, the error bound delineates a three-way trade-off between sample efficiency (via the accelerated variance term), representation power (via the approximation bias), and numerical robustness (via the regularization bias). The

presence of $\lambda_{\min}^{-2}$ in both bias terms underscores the critical role of policy manifold curvature: when the Fisher matrix is ill-conditioned, not only does the sampling variance require more samples to reduce (as seen in the constant $C_{\mathrm{arqmc}}$), but the biases are also magnified. This result provides a rigorous foundation for understanding the interplay between low-discrepancy sampling, function approximation, and preconditioning in natural actor-critic methods, and it justifies the use of Array-RQMC as a variance-reduction technique that operates orthogonally to the approximation and regularization errors.

The theoretical rate improvement of Array-RQMC over standard Monte Carlo is most meaningful when the variance of the NAC estimator is dominated by simulation noise arising from trajectory generation rather than by irreducible environmental noise, function-approximation bias, or optimization error. In particular, Array-RQMC is expected to be practically beneficial when three conditions are approximately met. First, the transition simulator admits a sufficiently smooth transformation from low-dimensional uniform inputs to next states, so that nearby randomized low-discrepancy inputs produce nearby successor states. Second, the effective dimension of the trajectory-generation problem is moderate, or the dominant variation is concentrated in a small subset of coordinates. Third, the additional sorting and coupling cost incurred by Array-RQMC is small relative to the cost of simulating trajectories and solving the NAC linear system. Under these conditions, the more uniform coverage of the driving random variables reduces the dispersion of the empirical estimates of the NAC matrices and vectors, thereby improving the stability of the Fisher-preconditioned update. This is particularly relevant in settings where small errors in the compatible-feature linear system are amplified along directions of low Fisher curvature. Conversely, the practical advantage of Array-RQMC may diminish in very high-dimensional transition models, highly discontinuous simulators, environments with large irreducible stochasticity, or regimes where sorting overhead dominates rollout cost. Thus, Array-RQMC should not be viewed as uniformly superior to Monte Carlo in all NAC settings; rather, its advantage is expected in simulators with smooth, structured stochasticity and moderate effective dimension, where variance reduction in the trajectory estimator translates into more stable natural-gradient updates.

## 5 Experiments

Here, we demonstrate the effectiveness of low-discrepancy sampling for estimating the natural policy gradient. All experiments use Sobol sequences, randomized via a Left Matrix scramble combined with a digital shift. Points from the RQMC set are generated using the *Sobol Engine* (Virtanen et al., 2020). Our emphasis is on analysing the behaviour of the gradient estimator and the resulting optimisation dynamics, rather than relying solely on task-level performance metrics. Hence, we consider the Gradient Noise Scale (GNS) as a scale-aware proxy for estimator variance, together with the update magnitude $\log\|w_t\|$ as an indicator of stability and conditioning. Note that the within-window second moment of the update $w_t$ can be decomposed as follows:

$$\mathbb{E}\big[\|w_t\|^2\big] = \|\mathbb{E}[w_t]\|^2 + \mathrm{tr}\big(\mathrm{Cov}(w_t)\big),$$

which separates the contribution of the mean update direction from its stochastic dispersion. One can interpret GNS as a scale-normalized measure of noise, proportional to $\mathrm{tr}\big(\mathrm{Cov}(w_t)\big)/\|\mathbb{E}[w_t]\|^2$, thereby quantifying stochastic variability relative to the strength of the update signal. Intuitively, GNS measures the variance of the gradient relative to its signal: a lower GNS means either the gradient estimator's variance is low or its mean direction is strong (or both), indicating a high signal-to-noise ratio in $w_t$. In addition to GNS, we consider average return to assess task-level performance, as well as several complementary diagnostics that characterize optimization behavior. The update magnitude $\log\|w_t\|$ captures the scale and smoothness of parameter motion, with large fluctuations indicating poor conditioning or noisy gradients. We further report the KL divergence between successive policies, which measures the effective step size in policy space and serves as a proxy for trust-region behavior, and policy entropy, which reflects the evolution of exploration and guards against premature collapse. Together, these metrics provide a multi-faceted view of both performance and stability, enabling a principled comparison of variance-reduction effects beyond asymptotic return alone. All reported results are obtained by averaging over multiple independent runs with different random seeds. Curves show the seed-wise mean together with standard deviation (shaded), computed over fixed training windows to expose between-run stochasticity while attenuating non-stationarity within individual trajectories.

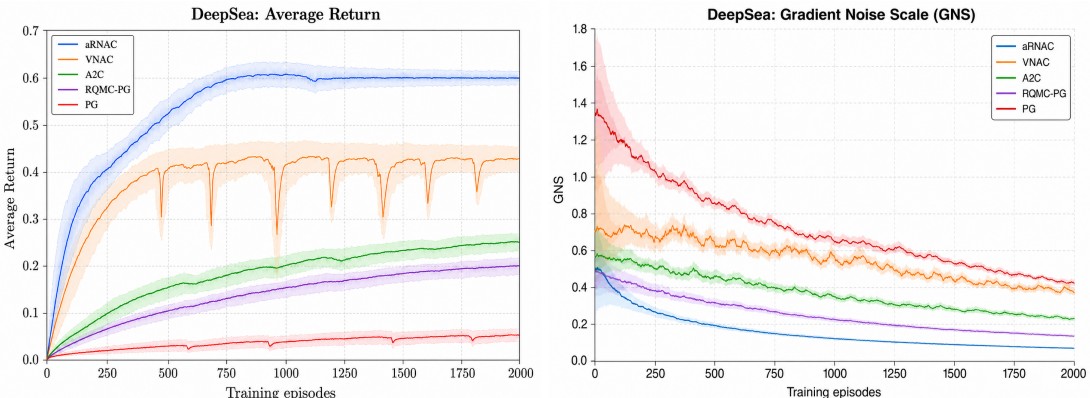

Figure 4: **DeepSea (BSuite):** DeepSea is a sparse-reward, long-horizon exploration task in which early policy-gradient estimates are dominated by noise. aRNAC attains the highest final return and exhibits the most stable post transient behaviour, while VNAC shows larger transient variability during training. The entropy curves indicate faster policy concentration for aRNAC, whereas PG and A2C maintain higher entropy over a longer horizon. The GNS profile suggests lower gradient-estimator variability for aRNAC, with VNAC remaining comparatively elevated. Shaded regions denote variability across independent runs.

**1. DeepSea:** DeepSea environment from Bsuite (Osband et al., 2019) constitutes a particularly stringent diagnostic for policy-gradient methods, as the learning signal is both sparse and exponentially weak in the horizon. This causes early gradient estimates to be dominated by stochastic noise. In this setting, we observe a clear separation between methods. As shown in Figure 4, our algorithm aRNAC achieves consistently higher average return while maintaining the lowest gradient noise scale (GNS). Although Vanilla NAC (VNAC) and A2C eventually reduce noise relative to vanilla policy gradients, both exhibit substantially higher variance during the critical early and mid-learning phases. Also A2C displays transient instability due to critic-induced noise. Meanwhile, the variance-reduced natural updates of aRNAC suppress stochastic fluctuations at the level of the gradient estimator itself, allowing weak reward signals to accumulate reliably over long horizons.

Figure 5 shows qualitative differences in optimisation dynamics between the methods. aRNAC consistently operates with smaller and smoother update magnitudes, as shown in the log|update| trace. VNAC and A2C exhibit irregular oscillations and vanilla policy gradients remain highly erratic. Additionally, aRNAC sustains controlled exploration without premature entropy collapse and induces significantly smaller KL divergence between successive policies, indicating conservative, trust-region-like steps in policy space. These results suggest that the benefits of aRNAC in this setting arise not merely from improved asymptotic performance, but from fundamentally more stable learning dynamics under extreme noise and delayed reward, which corroborates our variance-reduction and natural-gradient analysis.

**2. Frozen-Lake:** FrozenLake (Brockman et al., 2016) involves grid-based navigation where the agent aims to reach a goal while avoiding failure caused by slippery transitions. It represents a fundamentally different setting from DeepSea. Here, the state–action space is small and discrete, but the dynamics are highly stochastic and the reward signal is sparse, with a single terminal success and frequent catastrophic failures. In this setting, we observe that aRNAC consistently attains higher average returns than other methods. It also converges faster and more reliably to the optimal policy. As shown in Figure 6, vanilla policy gradients struggle to propagate sparse terminal rewards through the stochastic transitions, resulting in noisy and unstable learning trajectories. While VNAC and A2C partially mitigate this issue through curvature information or critic bootstrapping, both exhibit slower convergence and greater sensitivity to transition randomness. In contrast, the variance-reduced natural updates of aRNAC enable effective credit assignment even when successful trajectories are rare, leading to more dependable improvement across seeds.

FrozenLake setting also exhibits differences in optimization behavior. The log|update| traces indicate that aRNAC performs consistently smaller and more stable parameter updates. VNAC exhibits noticeable fluc-

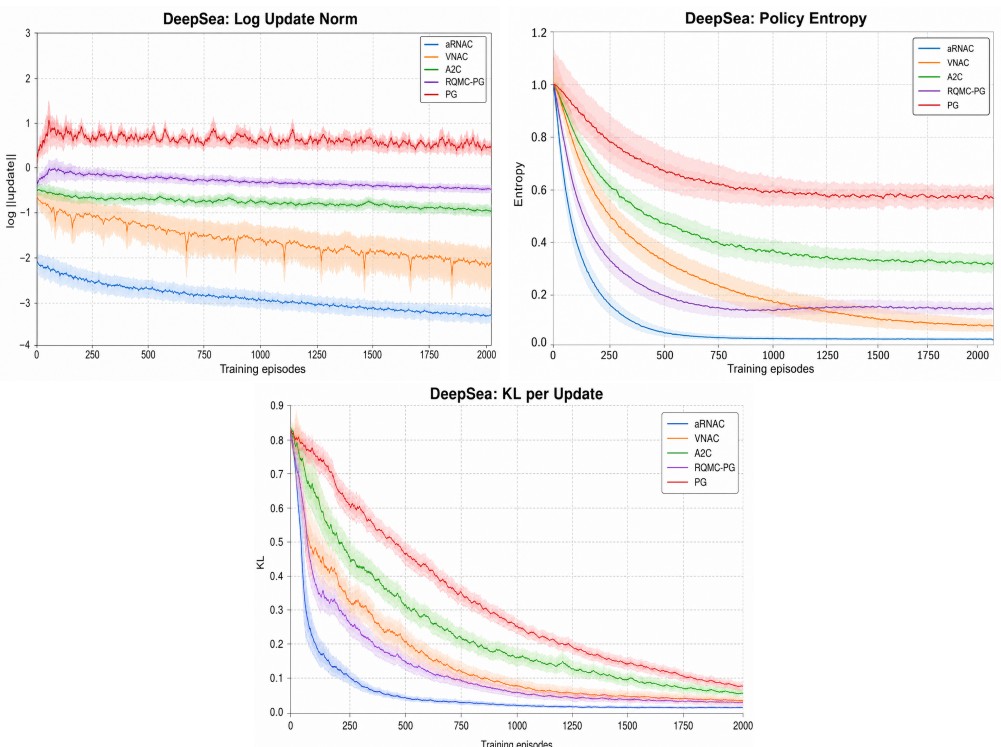

Figure 5: **DeepSea (BSuite):** The log ‖update‖ traces show that aRNAC performs consistently smaller and smoother parameter updates, whereas VNAC and A2C exhibit larger fluctuations and PG shows highly erratic behavior. Entropy evolution indicates that aRNAC preserves controlled exploration over long horizons without premature collapse. KL divergence between successive policies further confirms this stability: variance-reduced natural updates induce smaller policy shifts per iteration, approximating a trust-region behavior without explicit constraints.

tuations and PG shows erratic behavior throughout training. We observe similar ordering in the gradient noise scale, where aRNAC maintains the lowest noise level, followed by A2C and VNAC. The policy gradient remains the noisiest. These dynamics suggest that, even in low-dimensional settings, controlling estimator variance is crucial when transition stochasticity dominates the learning signal. The results show that variance reduction at the level of the natural-gradient estimator leads to more robust and reliable learning, in small, discrete environments with fragile reward structure.

**3. Acrobot:** Acrobot (Brockman et al., 2016) presents a continuous-state (discretized for the algorithm), under-actuated control problem in which successful learning requires coordinating torque-limited actions over long horizons to achieve an unstable upright configuration. It is characterised by a smooth but highly nonconvex control landscape in which progress is achieved not through local reward shaping, but by gradually accumulating mechanical energy via coordinated torque application. In this regime, learning is limited less by exploration failure and more by poor conditioning of gradient directions when the policy is far from the swing-up manifold. In this setting, we observe that aRNAC attains higher average return and converges more rapidly than all baselines (Figure 7). Vanilla policy gradients exhibit slow and erratic improvement, reflecting the high variance of Monte Carlo gradient estimates in the presence of delayed rewards and nonlinear dynamics. While A2C improves sample efficiency by introducing a learned baseline, its performance remains sensitive to critic noise, leading to flatter learning curves and increased variability across seeds. VNAC benefits from natural-gradient geometry and demonstrates improved stability relative to first-order methods, but still lags behind aRNAC in terms of convergence speed and final return. These results indicate that, for Acrobot, both curvature information and variance control are necessary to reliably exploit the weak reward signal induced by long-horizon swing-up behaviour.

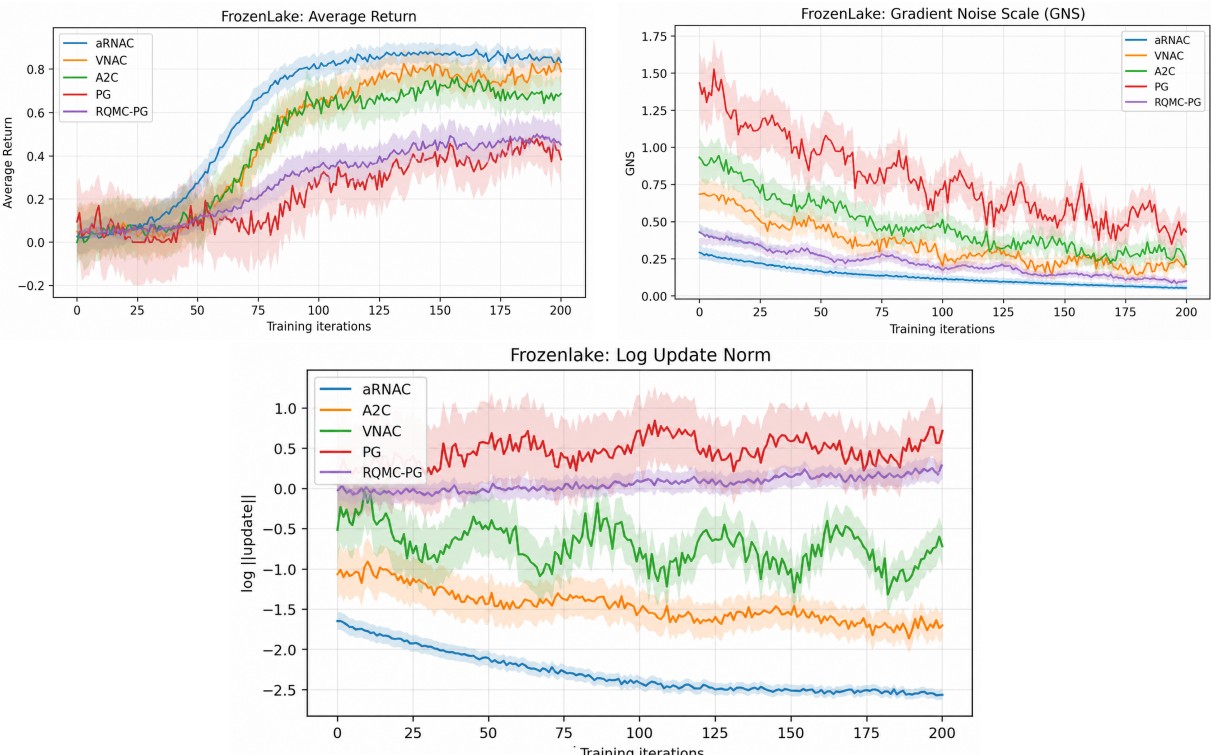

Figure 6: **FrozenLake:** [ *Top* ] average episodic return (higher is better). [ *Middle* ] gradient noise scale (GNS; lower indicates a less noisy update direction). [ *Bottom* ] log-norm of the update vector $\log \|\Delta_t\|$ (lower indicates smaller, more stable steps). Shaded regions denote standard deviation across runs. aRNAC achieves higher returns with consistently lower GNS and smaller update norms than VNAC, A2C, and PG, reflecting improved stability from randomized low-discrepancy sampling within the NAC estimator. VNAC exhibits noticeable oscillations, while PG remains the noisiest and least stable baseline. Here, for NAC-family methods (VNAC, aRNAC), $\Delta_t \equiv w_t$ (the compatible-critic / natural-step vector); for A2C and PG, $\Delta_t \equiv g_t$ (the actual policy-gradient update used by the optimiser).

The log|update| trace shows that aRNAC consistently callibrates with smaller and smoother parameter updates, whereas VNAC exhibits noticeable oscillations and A2C shows intermittent bursts associated with critic-induced noise. This behavior reflects in the gradient noise scale also, where aRNAC maintains the lowest noise level during training, followed by VNAC, and policy gradients remaining substantially noisier. Note that these stability advantages emerge early in training, when the policy is far from optimal and gradient estimates are most unreliable. The results demonstrate that variance-reduced natural updates do not merely improve asymptotic performance but fundamentally stabilize learning dynamics in nonlinear control tasks, enabling more reliable optimization under delayed rewards and complex dynamics.

**4. Taxi Domain:** The Taxi domain (Brockman et al., 2016) involves grid-based navigation with the goal of completing a passenger pickup and drop-off efficiently while avoiding illegal actions. It differs qualitatively from the other benchmarks considered. The rewards are neither purely sparse nor dense, but arise from a sequence of interdependent sub-decisions involving navigation, pickup, and drop-off under a discrete, factored state space. The results are summarised in Figure 8 In this setting, we observe that aRNAC achieves higher average return and reaches stable performance significantly earlier than other methods. Vanilla policy-gradient methods tend to have difficulty coordinating decisions across stages, leading to policies that perform well in navigation yet fail to reliably carry out the pickup–drop-off sequence. VNAC improves upon this behaviour by exploiting policy geometry, yet its progress remains uneven, reflecting sensitivity to noisy subtask-level gradients. A2C, while benefiting from bootstrapped value estimates, exhibits slower improvement due to critic variance propagating across the tightly coupled decision stages. In contrast,

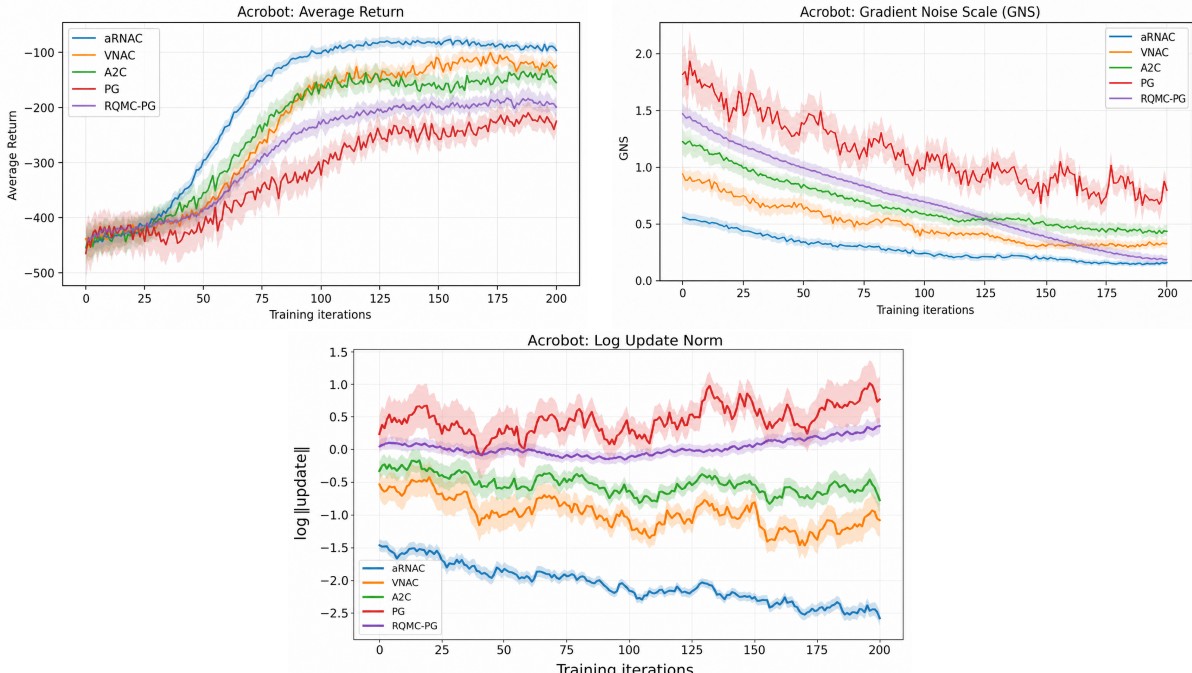

Figure 7: **Acrobot:** [ *Top* ] average episodic return. [ *Middle* ]: gradient noise scale (GNS; lower is better). [ *Bottom* ]: log-norm of the update vector $\log\|\Delta_t\|$. Shading indicates standard deviation across runs. aRNAC attains higher returns with markedly lower GNS and smaller update norms than VNAC, A2C, and PG, consistent with variance-reduction in the natural-actor update; VNAC fluctuates more, and PG is the noisiest baseline. Note the update ($y$-axis) $\Delta_t = w_t$ for NAC-family methods; $\Delta_t = g_t$ for A2C/PG.

aRNAC consistently integrates weak but structured learning signals across subtasks, resulting in more reliable end-to-end task completion.

Unlike long-horizon exploration tasks, this setting induces frequent but low-magnitude gradient signals whose utility depends on accurate aggregation across states with shared semantic roles (e.g., pickup locations versus drop-off destinations). The log |update| traces show that aRNAC performs smaller, well-regulated updates and avoids the oscillatory behavior observed in VNAC and the abrupt jumps characteristic of vanilla policy gradients. This stabilization is further mirrored in the gradient noise scale, where aRNAC maintains consistently lower noise, enabling more coherent policy refinement across related state clusters. Importantly, these effects emerge without sacrificing exploration, indicating that the advantage of aRNAC in Taxi stems not from aggressive policy changes but from improved statistical efficiency in assembling compositional decisions. Overall, the Taxi results show that variance-reduced natural gradients are especially effective in domains dominated by structured credit-assignment requirements.

**5. GridWorld:** In the Gridworld setting (Brockman et al., 2016), the agent operates over a discrete set of states and actions, aiming to reach a designated goal state while minimizing path length and avoiding inefficient transitions. Here, dynamics are mostly driven by dense but locally myopic rewards. The performance depends on the agent's ability to exploit spatial regularities while avoiding the accumulation of suboptimal local decisions. Unlike Taxi domain, where optimality requires coordinating a fixed sequence of subtasks, Gridworld requires consistent improvement across many short-horizon transitions whose effects compound over time. In this setting, aRNAC achieves higher average return and exhibits more reliable convergence than all baselines (Figure 9). Vanilla policy gradient oscillates between locally reasonable but globally inefficient paths. VNAC and A2C improve stability relative to first-order methods, however their learning curves remain more or less flatter. This shows their difficulty in aggregating weak improvements across neighbouring states. In contrast, aRNAC more effectively exploits the spatial smoothness of the environment, and it translates small, consistent improvements into long-term performance gains.

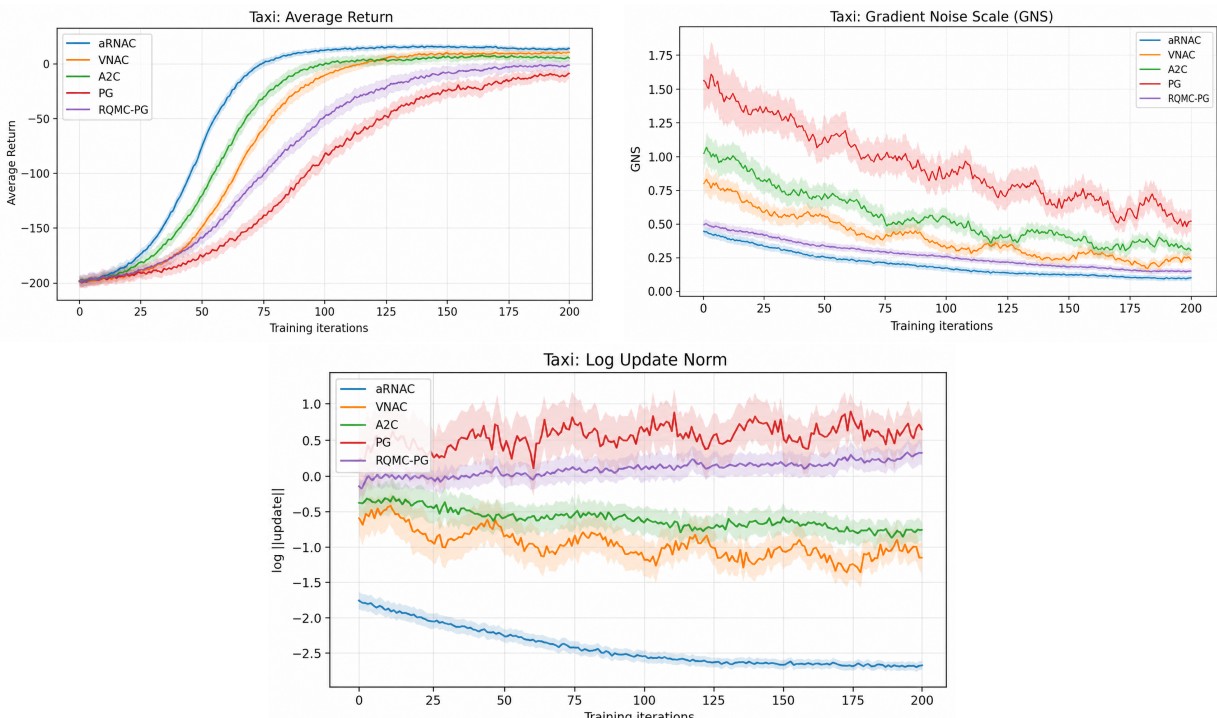

Figure 8: **Taxi Domain:** [ *Top* ] average episodic return (higher is better). [ *Middle* ] gradient noise scale (GNS; lower indicates a less noisy update direction). [ *Bottom* ] log-norm of the update vector $\log \|\Delta_t\|$ (lower indicates smaller, more stable steps). Shaded regions denote standard deviation across multiple runs. aRNAC achieves higher returns while maintaining lower GNS and smaller update norms than VNAC, A2C, and PG, consistent with variance reduction from randomized low-discrepancy sampling within the NAC linear-system estimator. VNAC exhibits larger oscillations, and PG remains the noisiest and least stable baseline. Note that for NAC-family methods (VNAC, aRNAC), $\Delta_t \equiv w_t$ (the compatible-critic / natural-step vector); for A2C and PG, $\Delta_t \equiv g_t$ (the actual policy-gradient update used by the optimiser).

Although individual transitions are short-horizon, the high number of updates amplifies estimator noise, making stability over many iterations indeed essential. The $\log|\text{update}|$ traces show that aRNAC maintains uniformly smaller and smoother updates, whereas VNAC displays moderate fluctuations and vanilla policy gradient remain highly erratic. This behaviour is reflected in the gradient noise scale, where aRNAC consistently attains the lowest noise levels, and obtain coherent policy refinement across adjacent states. Notably, these advantages emerge without suppressing exploration. This shows that aRNAC balances stability and adaptability even in densely connected state spaces. Overall, these results show that by controlling gradient noise, one can essentially improve performance not only in sparse-reward or long-horizon tasks, but also in structured environments where local errors can add up over time thus impacting cumulative performance.

**6. Continuous Control - Stochastic Hopper/v4:** Standard MuJoCo Hopper/v4 (Brockman et al., 2016) is a three-joint planar locomotion task with an 11-dimensional continuous state space and a 3-dimensional continuous action space, in which the dynamics are deterministic conditional on the current state and action. Since the Array-RQMC analysis assumes access to a stochastic simulator oracle of $\Lambda$, we consider stochastic Hopper/v4 by perturbing the deterministic MuJoCo transition map $f_{\mathrm{MJ}}$ with controlled Gaussian noise:

$$\mathbf{s}_{t+1} = \Lambda(\mathbf{s}_t, \mathbf{a}_t, \mathbf{u}_t) = f_{\mathrm{MJ}}(\mathbf{s}_t, \mathbf{a}_t) + \Sigma^{1/2}\Phi^{-1}(\mathbf{u}_t), \tag{66}$$

where $\mathbf{u}_t \in [0,1)^{11}$, $\Phi^{-1}$ is applied component-wise, and $\Sigma$ is a diagonal covariance matrix. The induced transition kernel is therefore

$$\mathbb{P}(\cdot \mid \mathbf{s}_t, \mathbf{a}_t) = \mathcal{N}(f_{\mathrm{MJ}}(\mathbf{s}_t, \mathbf{a}_t), \Sigma),$$

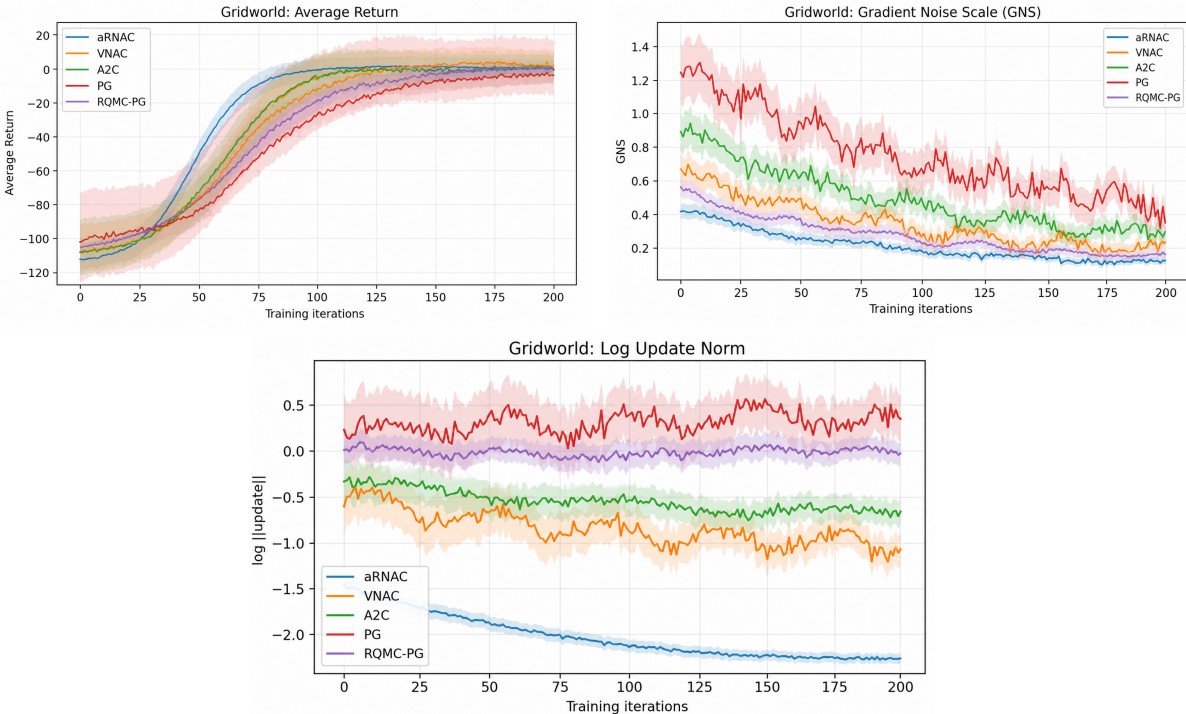

Figure 9: **Gridworld:** [ *Top* ] average episodic return. [ *Middle* ] gradient noise scale (GNS; lower is better). [ *Bottom* ] log-norm of the update vector $\log \|\Delta_t\|$. Shaded regions denote standard deviation across runs. aRNAC achieves the best return and the lowest noise/step magnitudes; VNAC exhibits noticeable oscillations; PG remains unstable and noisy; A2C is intermediate. Again, we consider the update $\Delta_t = w_t$ for NAC-family methods; $\Delta_t = g_t$ for A2C/PG.

and $\Lambda$ provides a deterministic inverse-transform simulator for this stochastic kernel. The noise is clipped to preserve numerical stability and avoid physically invalid states. This construction ensures that the environment satisfies the stochastic simulator oracle assumed in the Array-RQMC formulation, while retaining the continuous-control structure of the original benchmark. For aRNAC, we evolve $N$ parallel Hopper trajectories. At each time step, an independent scrambled Sobol point set $\Omega_t = \{\mathbf{z}_t^1, \ldots, \mathbf{z}_t^N\} \subset [0,1)^{14}$ is generated. Each point is decomposed as $\mathbf{z}_t^i = (\xi_t^i, \mathbf{u}_t^i)$, where $\xi_t^i$ drives action sampling and $\mathbf{u}_t^i$ drives the stochastic transition. Under a Gaussian policy,

$$\mathbf{a}_t^i = \mu_\theta(\mathbf{s}_t^i) + \sigma_\theta(\mathbf{s}_t^i)\, \Phi^{-1}(\xi_t^i),$$

and the next state is generated via (66).

As shown in Figure 10, A2C attains the strongest return profile and stabilizes near the best observed range, while aRNAC exhibits a smoother and more stable improvement trajectory than both PG and VNAC. RQMC-PG improves over PG by reducing sampling irregularity in the action-generation process, yet its performance remains below that of aRNAC after sufficient training. This ordering is consistent with the role of natural preconditioning: RQMC-PG reduces the variance of the policy-gradient estimator, whereas aRNAC additionally exploits the geometry induced by the compatible-feature system to control both the direction and the magnitude of the update. VNAC, despite incorporating natural-gradient information, exhibits large oscillations and wide confidence bands throughout training, confirming that the compatible-feature regression is sensitive to noisy trajectory estimates in the absence of low-discrepancy coupling. PG remains the weakest baseline owing to high estimator variance and the absence of both critic-based and Fisher-geometric correction.

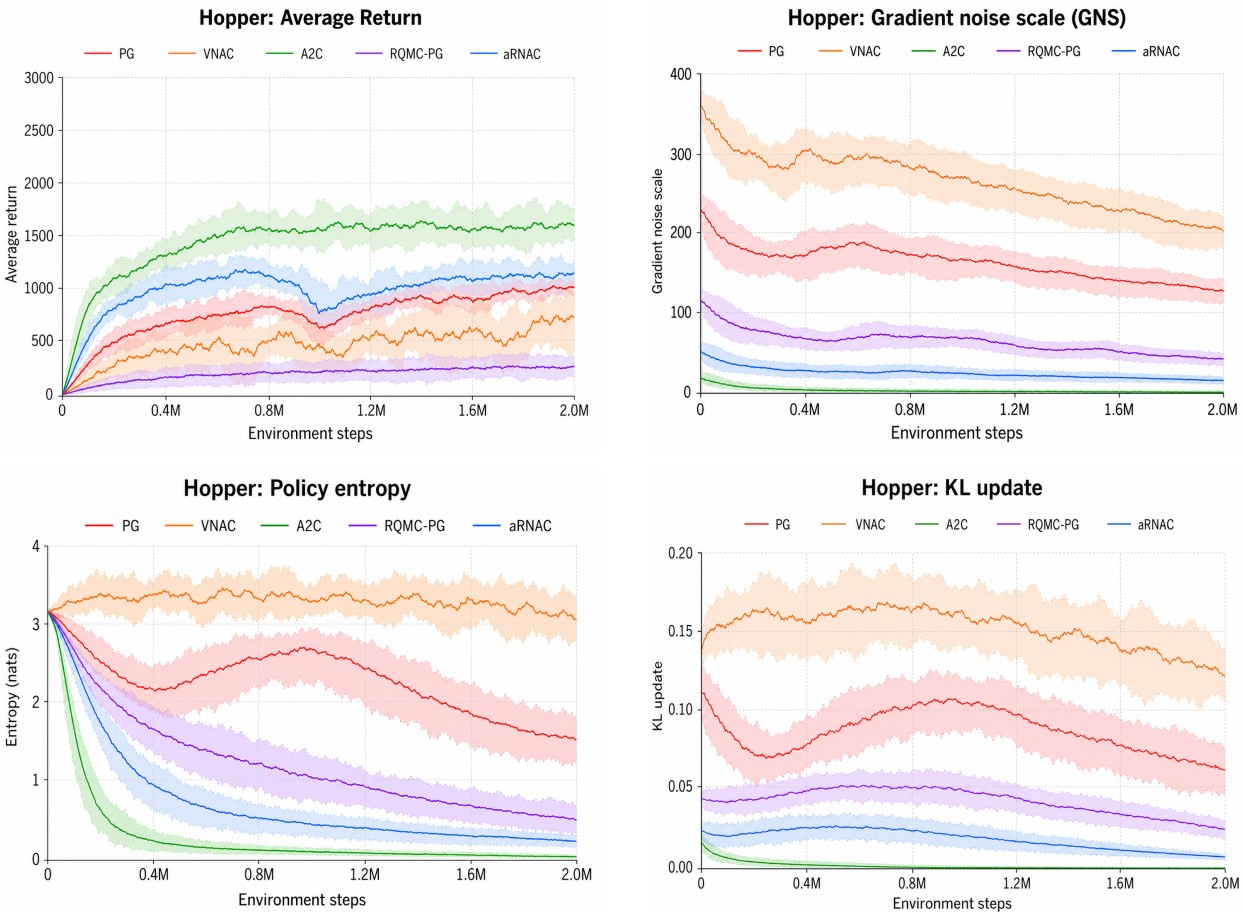

Figure 10: Stochastic Hopper-v4 comparison across average return, gradient noise scale [ *Top* ] and policy entropy, KL update [ *Bottom* ]. The average-return curve illustrates that aRNAC attains a higher and more stable learning profile than VNAC. The gradient-noise-scale curve shows that aRNAC maintains lower gradient noise than VNAC, consistent with the variance-reduction effect of Array-RQMC. The entropy and KL-update curves further indicate that aRNAC produces more concentrated and smaller policy updates, suggesting a more stable natural-gradient trajectory.

The GNS profile shows that VNAC maintains the highest update noise throughout training, reflecting the sensitivity of the natural-gradient direction to noisy Fisher-compatible regression in continuous control. PG also retains a high GNS, though its instability is less pronounced than VNAC. By contrast, RQMC-PG and aRNAC exhibit substantially lower GNS, with aRNAC remaining consistently below RQMC-PG after the initial transient phase. This supports the interpretation that randomized low-discrepancy sampling attenuates the sampling noise entering the policy update, and that the natural-gradient structure further moderates the effective update direction by projecting this noise onto the score-feature span. The entropy and KL-per-update diagnostics complement these observations. A2C rapidly concentrates its policy and achieves the strongest return, while aRNAC reduces entropy gradually and maintains a controlled exploration-to-exploitation transition. RQMC-PG displays an intermediate entropy profile, consistent with improved sampling regularity but without the additional stabilizing effect of the compatible natural-gradient system. VNAC sustains high entropy and large KL-per-update fluctuations, indicating erratic policy displacement rather than controlled exploration. The KL-per-update profile further confirms that aRNAC induces smaller and smoother policy shifts than PG, VNAC, and RQMC-PG, approximating a trust-region-like behaviour without explicit constraints, consistent with the results observed across the discrete benchmarks. In this setting, the dynamics are largely deterministic conditional on state and action, so the low-discrepancy inputs primarily affect policy-induced stochasticity and the injected transition noise rather than a tabular transi-

tion kernel. The results nonetheless support the practical applicability of the proposed variance-reduction mechanism to continuous control.

**7. Wall-clock Efficiency:** We now consider whether the variance reduction obtained by aRNAC translates into an end-to-end computational advantage. This comparison is important because Array-RQMC modifies not only the sampling structure but also the computational profile of each update. In particular, aRNAC requires randomized low-discrepancy point generation and state sorting/coupling across trajectories. Hence, it is not sufficient to compare variance or return alone; one must also account for the additional wall-clock cost introduced by these operations.

We evaluate this trade-off on a DeepSea instance with grid size 20, so that $|\mathcal{S}| \approx 400$. The trajectory length is set to $T = 20$, matching the horizon required to traverse the grid, and each update uses $N = 64$ trajectories. Both VNAC and aRNAC use the same policy architecture, critic update, trajectory budget, and hardware setting. This isolates the effect of replacing independent Monte Carlo trajectory generation with the Array-RQMC construction. Figure 11 summarizes the resulting wall-clock comparison. Figure 11(a) reports the per-update runtime decomposition. For VNAC, the dominant components are rollout generation and the critic/natural-gradient solve. For aRNAC, these same components remain present, while the additional cost is attributed to RQMC point generation and sorting/coupling of the trajectory array. The per-update runtime is therefore higher for aRNAC, as expected. However, the increase is small relative to the rollout and critic costs, indicating that the Array-RQMC overhead does not dominate computation in this setting. Figure 11(b) reports the number of updates required to reach the target return. aRNAC reaches this target in substantially fewer updates than VNAC. This reduction is consistent with the role of Array-RQMC as a variance-reduction mechanism: the trajectory array is more evenly distributed, which reduces estimator dispersion and yields a more reliable natural-gradient update. Thus, although an individual aRNAC update is slightly more expensive, it carries a more effective policy-improvement signal. Figure 11(c) combines these two effects by reporting the total time required to reach the target return. The lower time-to-target for aRNAC shows that the reduction in update count outweighs the additional per-update overhead. Thus the proposed sampling scheme should not only reduce estimator variance, but should also improve the computational efficiency of training when measured end to end. Overall, these results suggest that the Array-RQMC overhead is modest in relation to the main computational costs of NAC, while the reduction in required policy updates is substantial.

## 6 Conclusion

In this paper, we analyzed the Natural Actor–Critic method from the perspective of stochastic estimation error in the natural-gradient update. By examining the compatible-feature solution that defines the natural gradient, we showed how randomness inherent in this estimator propagates through Fisher preconditioning and affects the magnitude and conditioning of parameter updates. This analysis shows that the behavior of NAC is influenced not only by the geometric structure induced by the Fisher matrix but also by the variance properties of the estimator used to approximate the natural gradient. Within this framework, we showed that low-discrepancy sampling provides a principled way to reduce stochastic variability in the natural-gradient estimator while preserving unbiasedness. In particular, randomized low-discrepancy constructions reduce estimator dispersion relative to standard Monte Carlo sampling without altering the optimization objective or update structure. The theoretical analysis of our RQMC-based NAC estimators shows a reduction in estimator variance under reasonable conditions. Together, these results characterize the impact of estimator variance on natural-gradient methods and its implications for stability and conditioning in natural actor–critic algorithms.

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

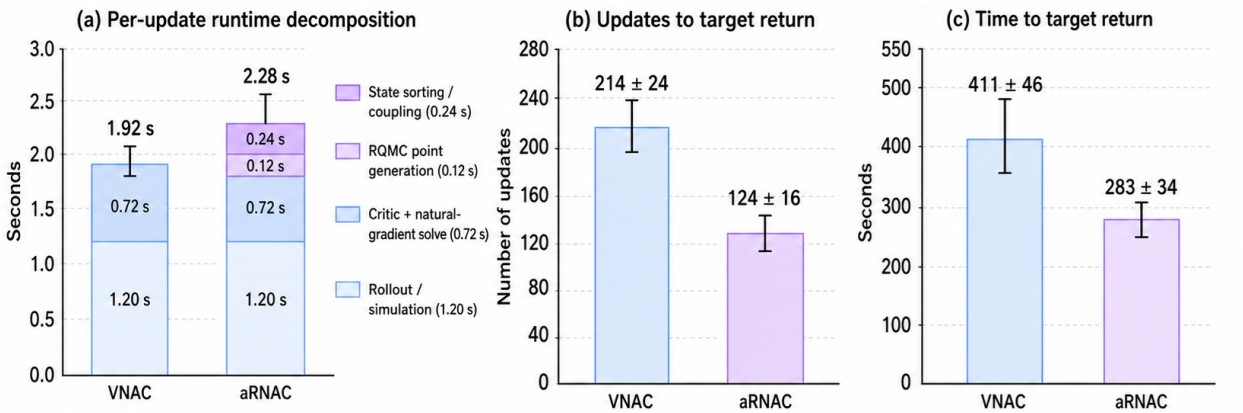

Figure 11: Wall-clock comparison on DeepSea instance with grid size 20, corresponding to $|\mathcal{S}| \approx 400$, using trajectory length $T = 20$ and $N = 64$ trajectories per policy update. **(a)** Decomposes the per-update runtime into rollout cost, critic plus natural-gradient solve cost, and the additional Array-RQMC overhead arising from low-discrepancy point generation and state sorting/coupling. Here, aRNAC incurs a modest increase in per-update cost relative to VNAC. **(b)** Shows the number of policy updates required to reach a fixed target return. The variance-reduced estimator used by aRNAC reaches the target in substantially fewer updates, indicating more sample-efficient and stable improvement. **(c)** Shows the resulting time-to-target return. Although aRNAC is slightly more expensive per update, the reduction in the number of updates more than compensates for this overhead, yielding a lower overall wall-clock time.

Sébastien MR Arnold, Pierre L'Ecuyer, Liyu Chen, Yi-Fan Chen, and Fei Sha. Policy learning and evaluation with randomized quasi-monte carlo. In *International Conference on Artificial Intelligence and Statistics*, pp. 1041–1061. PMLR, 2022.

Jonathan Baxter and Peter L Bartlett. Infinite-horizon policy-gradient estimation. *Journal of Artificial Intelligence Research*, 15:319–350, 2001.

Dimitri P Bertsekas. Neuro-dynamic programming. In *Encyclopedia of optimization*, pp. 1–6. Springer, 2025.

Shalabh Bhatnagar, Mohammad Ghavamzadeh, Mark Lee, and Richard S Sutton. Incremental natural actor-critic algorithms. *Advances in neural information processing systems*, 20, 2007.

Justin A Boyan. Least-squares temporal difference learning. In *ICML*, pp. 49–56, 1999.

Greg Brockman, Vicki Cheung, Ludwig Pettersson, Jonas Schneider, John Schulman, Jie Tang, and Wojciech Zaremba. Openai gym. *arXiv preprint arXiv:1606.01540*, 2016.

Henri Faure. Discrépance de suites associées à un système de numération (en dimension s). *Acta arithmetica*, 41(4):337–351, 1982.

Will Grathwohl, Dami Choi, Yuhuai Wu, Geoff Roeder, and David Duvenaud. Backpropagation through the void: Optimizing control variates for black-box gradient estimation. In *International Conference on Learning Representations*, 2018.

Tuomas Haarnoja, Aurick Zhou, Pieter Abbeel, and Sergey Levine. Soft actor-critic: Off-policy maximum entropy deep reinforcement learning with a stochastic actor. In *International conference on machine learning*, pp. 1861–1870. PMLR, 2018.

John H Halton. On the efficiency of certain quasi-random sequences of points in evaluating multi-dimensional integrals. *Numerische Mathematik*, 2:84–90, 1960.

Nan Jiang and Lihong Li. Doubly robust off-policy value evaluation for reinforcement learning. In *International conference on machine learning*, pp. 652–661. PMLR, 2016.

Stephen Joe and Frances Y Kuo. Constructing sobol sequences with better two-dimensional projections. *SIAM Journal on Scientific Computing*, 30(5):2635–2654, 2008.

Sham M Kakade. A natural policy gradient. *Advances in neural information processing systems*, 14, 2001.

Sajad Khodadadian, Prakirt Raj Jhunjhunwala, Sushil Mahavir Varma, and Siva Theja Maguluri. On the linear convergence of natural policy gradient algorithm. In *2021 60th IEEE Conference on Decision and Control (CDC)*, pp. 3794–3799. IEEE, 2021.

Vijay Konda and John Tsitsiklis. Actor-critic algorithms. *Advances in neural information processing systems*, 12, 1999.

Vijaymohan R Konda and Vivek S Borkar. Actor-critic–type learning algorithms for markov decision processes. *SIAM Journal on control and Optimization*, 38(1):94–123, 1999.

Pierre L'Ecuyer, Christian Lécot, and Bruno Tuffin. A randomized quasi-monte carlo simulation method for markov chains. *Operations research*, 56(4):958–975, 2008.

Pierre L'Ecuyer. *Randomized quasi-Monte Carlo: An introduction for practitioners*. Springer, 2018.

James Martens. New insights and perspectives on the natural gradient method. *The Journal of Machine Learning Research*, 21(1):5776–5851, 2020.

James Martens and Roger Grosse. Optimizing neural networks with kronecker-factored approximate curvature. In *International conference on machine learning*, pp. 2408–2417. PMLR, 2015.

Jiřı Matoušek. On thel2-discrepancy for anchored boxes. *Journal of Complexity*, 14(4):527–556, 1998.

Nicholas Metropolis and Stanislaw Ulam. The monte carlo method. *Journal of the American statistical association*, 44(247):335–341, 1949.

Shakir Mohamed, Mihaela Rosca, Michael Figurnov, and Andriy Mnih. Monte carlo gradient estimation in machine learning. *Journal of Machine Learning Research*, 21(132):1–62, 2020.

Harald Niederreiter. Quasi-monte carlo methods and pseudo-random numbers. *Bulletin of the American mathematical society*, 84(6):957–1041, 1978.

Ian Osband, Yotam Doron, Matteo Hessel, John Aslanides, Eren Sezener, Andre Saraiva, Katrina McKinney, Tor Lattimore, Csaba Szepesvari, Satinder Singh, et al. Behaviour suite for reinforcement learning. *arXiv preprint arXiv:1908.03568*, 2019.

Art B Owen. Randomly permuted (t, m, s)-nets and (t, s)-sequences. In *Monte Carlo and Quasi-Monte Carlo Methods in Scientific Computing: Proceedings of a conference at the University of Nevada, Las Vegas, Nevada, USA, June 23–25, 1994*, pp. 299–317. Springer, 1995.

Art B Owen. Monte carlo variance of scrambled net quadrature. *SIAM Journal on Numerical Analysis*, 34 (5):1884–1910, 1997a.

Art B Owen. Scrambled net variance for integrals of smooth functions. *The Annals of Statistics*, 25(4): 1541–1562, 1997b.

Art B. Owen. *Monte Carlo Theory, Methods and Examples*. Stanford University, 2013. URL https://statweb.stanford.edu/ owen/mc/. Unpublished manuscript.

Anargyros Papageorgiou. Sufficient conditions for fast quasi-monte carlo convergence. *Journal of Complexity*, 19(3):332–351, 2003.

Jan Peters, Sethu Vijayakumar, and Stefan Schaal. Natural actor-critic. In *Machine Learning: ECML 2005: 16th European Conference on Machine Learning, Porto, Portugal, October 3-7, 2005. Proceedings 16*, pp. 280–291. Springer, 2005.

Doina Precup. *Temporal abstraction in reinforcement learning*. University of Massachusetts Amherst, 2000.

Florian Puchhammer, Amal Ben Abdellah, and Pierre L'Ecuyer. Variance reduction with array-rqmc for tau-leaping simulation of stochastic biological and chemical reaction networks. *Bulletin of Mathematical Biology*, 83(8):91, 2021.

Martin L Puterman. *Markov decision processes: discrete stochastic dynamic programming*. John Wiley & Sons, 2014.

John Schulman, Sergey Levine, Pieter Abbeel, Michael Jordan, and Philipp Moritz. Trust region policy optimization. In *International conference on machine learning*, pp. 1889–1897. PMLR, 2015.

John Schulman, Philipp Moritz, Sergey Levine, Michael Jordan, and Pieter Abbeel. High-dimensional continuous control using generalized advantage estimation. In *International Conference on Learning Representations (ICLR)*, 2016.

Il'ya Meerovich Sobol'. On the distribution of points in a cube and the approximate evaluation of integrals. *Zhurnal Vychislitel'noi Matematiki i Matematicheskoi Fiziki*, 7(4):784–802, 1967.

Richard S Sutton, Andrew G Barto, et al. *Introduction to reinforcement learning*, volume 135. MIT press Cambridge, 1998.

Richard S Sutton, David McAllester, Satinder Singh, and Yishay Mansour. Policy gradient methods for reinforcement learning with function approximation. *Advances in neural information processing systems*, 12, 1999.

Philip Thomas. Bias in natural actor-critic algorithms. In *International conference on machine learning*, pp. 441–448. PMLR, 2014.

George Tucker, Andriy Mnih, Chris J Maddison, John Lawson, and Jascha Sohl-Dickstein. Rebar: Low-variance, unbiased gradient estimates for discrete latent variable models. *Advances in Neural Information Processing Systems*, 30, 2017.

Pauli Virtanen, Ralf Gommers, Travis E Oliphant, Matt Haberland, Tyler Reddy, David Cournapeau, Evgeni Burovski, Pearu Peterson, Warren Weckesser, Jonathan Bright, et al. Scipy 1.0: fundamental algorithms for scientific computing in python. *Nature methods*, 17(3):261–272, 2020.

L Weaver and Nigel Tao. The optimal reward baseline for gradient-based reinforcement learning. In *Conference on Uncertainty in Artificial Intelligence (UAI 2001)*, pp. 538–545. Morgan Kauffman Publishers, 2001.

Junfeng Wen, Saurabh Kumar, Ramki Gummadi, and Dale Schuurmans. Characterizing the gap between actor-critic and policy gradient. In *International conference on machine learning*, pp. 11101–11111. PMLR, 2021.

Ronald J Williams. Simple statistical gradient-following algorithms for connectionist reinforcement learning. *Machine learning*, 8(3):229–256, 1992.

Pan Xu, Felicia Gao, and Quanquan Gu. Sample efficient policy gradient methods with recursive variance reduction. In *International Conference on Learning Representations*, 2020a.

Tengyu Xu, Zhe Wang, and Yingbin Liang. Improving sample complexity bounds for (natural) actor-critic algorithms. In H. Larochelle, M. Ranzato, R. Hadsell, M.F. Balcan, and H. Lin (eds.), *Advances in Neural Information Processing Systems*, volume 33, pp. 4358–4369. Curran Associates, Inc., 2020b. URL $https://proceedings.neurips.cc/paper_files/paper/2020/file/2e1b24a664f5e9c18f407b2f9c73e821-Paper.pdf$.

Junzi Zhang, Jongho Kim, Brendan O'Donoghue, and Stephen Boyd. Sample efficient reinforcement learning with reinforce. In *Proceedings of the AAAI conference on artificial intelligence*, volume 35, pp. 10887–10895, 2021.

