# OpenReview forum: "Analysis of Natural Actor-Critic with Randomized Low- Discrepancy Sampling"
_TMLR — Decision pending for TMLR_

### Review · Reviewer_x2Mm · 2026-03-06

**Summary Of Contributions:**

This paper studies the gradient estimation mechanism of the natural actor-critic framework, highlighting the high variance issue of the gradient estimator through extensive theoretical analysis. To address it, it proposes to integrate randomized Quasi-Monte Carlo sampling into the natural actor-critic framework, supported by a series of error bounds and experiments.

Overall, it is a mathematically intensive paper with a focus on one of the core problems in RL. I did not check every detail of the paper, but most results look valid and reasonable. I am particularly interested in Theorem 7 which proves that the NAC update can be decomposed into the sum of the true natural gradient plus the Bellman error induced by value approximation. Experimental results show that the proposed approach effectively reduces the magnitude of both gradient norm and gradient noise scale.

My main concern is that whether the analysis, as well as the proposed method, can be extended to more general cases (e.g., neural value functions) beyond linear value/policy parameterization such as the Sobol sequence considered in this paper. Besides, it does not provide enough details for the experiments, such as the actual parameterization of the value function and learning rate. It can be a critical issue, since the strength of the proposed algorithm is demonstrated by comparing against other methods, and implementation details are necessary to convince the audience that the comparison is fair.

**Audience:**

Yes

**Audience Explanation:**

See above.

**Broader Impact Concerns:**

None.

**Claims And Evidence:**

Yes

**Claims Explanation:**

See above.

**Requested Changes:**

Below are some minor points that I would suggest to change:

* Theorem 2 proves that the remainder is of the order of $O(|\delta|^3)$, which requires the policy $\pi_\theta$ to be third-order differentiable w.r.t. $\theta$. Under the current assumption it can only be $o(|\delta|^2)$. A counter example is $f(\delta) = \delta^{2.5}$ and $\delta \rightarrow 0$.

* The value of $\alpha$ is not explicitly mentioned in the statement of Theorem 3.

* Implementation details of the experiments should be provided.

* I would recommend the authors to move most of the theorem proofs to the appendix to make the main text easier to read.

---

> ### Author Response · Authors · 2026-05-14
> **Response**
>
> 1. We sincerely thank the reviewer for pointing this out. We have now updated the assumption to twice continuously differentiable (this properly mostly holds in softmax and policy networks).
>
> 2. Theorem 3 is now removed due to the comment from the first reviewer.
>
> 3. More details added to the experiments.
>
> 4. We thank the reviewer for the suggestion. The analysis is a central part of our contribution. The paper’s main claim depends on the non-asymptotic decomposition of the Array-RQMC NAC estimator and on identifying precisely when the improved rate over standard Monte Carlo is obtained. Therefore, we request you to re-consider this comment and allow us to retain the proof in the main text.

---

### Review · Reviewer_M3td · 2026-03-24

**Summary Of Contributions:**

The authors apply Randomized quasi-Monte Carlo (RQMC) and Array-RQMC to compute the update for Natural Gradient for Actor-Critic, providing a theoretical analysis that shows that when the critic only approximately satisfies the Bellman equation, the resulting update admits a clear structural decomposition and analyze the effect of randomness of the computation of the Fisher matrix on update magnitude, conditioning, and stability, finally they also do experiments to show that empirical results in general match their theoretical analysis and to show that the combination of Array-RQMC and Natural Actor-Critic leads to a more stable optimization trajectory and better policy performance throughout the training.

**Audience:**

Yes

**Audience Explanation:**

This paper is of interest both to the wide Reinforcement Learning community that reads TMLR, by showing a way to improve the stability of policy training and achieve better final performance on multiple environments (which combined with the theoretical proofs imply some level of generalization), but also to the part of the audience focused on optimization and stochastic methods, by further showing the impact/utility of Randomized quasi-Monte Carlo and Array Randomized quasi-Monte Carlo.

**Claims And Evidence:**

Yes

**Claims Explanation:**

The authors conduct their analysis with considerable mathematical rigor, starting from assumptions that are reasonable for our most applications (finite rewards, differentiable policies...), and then further enhance it by running experiments in DeepSea, Frozenlake, Acrobot, Taxi, and Gridworld.

**Requested Changes:**

I believe the paper is good as is

---

> ### Author Response · Authors · 2026-05-20
> **Response**
>
> We thank the reviewer for the appreciation

---

### Review · Reviewer_B6UH · 2026-04-20

**Summary Of Contributions:**

This paper proposes to integrate Randomized Quasi-Monte Carlo (RQMC) sampling into the natural actor-critic (NAC) algorithm. Specifically, the main contributions of this paper are three-fold: (i) By revisiting the NAC linear system, this paper shows that under imperfect value approximation, the NAC decomposes exactly into the true natural gradient plus a Fisher-metric projection of the Bellman residual onto the score feature span. (ii) This paper also proves that the Array-RQMC-based NAC estimators are unbiased and provide finite-sample MSE bounds showing that Array-RQMC achieves $O((log N)^{2d}/N^2)$ scaling versus $O(1/N)$ for standard Monte Carlo. (iii) Finally, the paper shows empirical validation on five discrete small-scale RL benchmarks (DeepSea, FrozenLake, Acrobot, Taxi, and GridWorld) demonstrating improved return, lower gradient noise scale, and more stable updates.

Strengths:

- One main novelty lies in the clean theoretical decomposition of NAC bias under function approximation
- Another main contribution is the principled variance-reduction argument with rate improvements.

Weaknesses:

- The paper did not compare the convergence results of NAC with those existing ones of NAC, such as (Xu et al., 2020), and this makes it difficult to fully position the contribution or the importance of these new results (especially Theorems 12-13 and 16-17). In addition to the dependency on $N$, there are also dependencies on several other factors, such as $\lambda\_{\min}$ and $\epsilon$, which seems to make the results somewhat not directly comparable to the existing results.
- The paper assumes access to a generative model/simulator, which can be implemented by inverse transform sampling, to obtain transitions (Assumption d in Theorem 8). While this is indeed one setting used in the RL literature, this can be quite restrictive and not always realistic.
- Experiments are limited to small, discrete environments.
- The preliminaries (Section 3) can be made more concise. Moreover, the presentation of the theoretical results (Section 4) is a bit dry in its current form.

**Audience:**

Yes

**Audience Explanation:**

Overall this paper can be of interest to the RL audience in general. The combination of the NAC method with quasi-Monte Carlo sampling is a natural and somewhat underexplored direction. The bias decomposition result is of independent theoretical interest for anyone studying actor-critic methods, and the guarantees under variance reduction are a meaningful contribution to the sample efficiency in RL.

**Broader Impact Concerns:**

The work is an algorithmic research on policy optimization in RL. There are no foreseeable direct negative societal impacts.

**Claims And Evidence:**

Yes

**Claims Explanation:**

The theoretical claims are generally well supported. To be more specific, the bias decomposition (cf. Theorem 7) is clearly derived and novel in the context of Neural Actor Critic. The unbiasedness proofs appear correct, with the induction argument over the marginal chain distributions carefully handled. For the MSE bounds (Theorems 12-13 and 16-17), I only did a high-level check, and they appear to follow a consistent proof strategy (splitting over well- and ill-conditioned events), which is quite standard.

However, some concerns also arise:

1. The variance bound $O((\log N)^{2d}/N^2)$ is only beneficial over $O(1/N)$ when $N$ is sufficiently large relative to d. The paper never discusses the crossover point or the practical dimension regime where the performance gains are attainable. For the experimental environments the state dimensions are very low, so the advantage is unsurprising; it is unclear whether the benefit extends to moderate or high dimensions.

2. The Array-RQMC method seems to require deterministic transition oracles via inverse transform sampling. This is feasible for discrete tabular MDPs but can be non-trivial for continuous control or learned environments.

3. Regarding the baselines, the comparison includes VNAC, A2C, and PG, but omits other standard variance-reduction baselines (e.g., REINFORCE with baseline, GAE, or vanilla RQMC policy gradient from Arnold et al. 2022, which is a highly relevant prior work). The absence of the RQMC baseline (Arnold et al. 2022) is a notable gap, since the paper claims novelty specifically over that work.

**Requested Changes:**

1. It would be helpful to provide a discussion characterizing under what conditions the rate of NAC with Array-RQMC is practically better than that of NAC with standard Monte Carlo. This is essential for understanding the real-world applicability of the method.

2. The authors are suggested to clarify the scope of the assumptions in Theorem 8 (especially 8d). Specifically, a discussion of when inverse transform sampling access is feasible, and how the method degrades or must be adapted when it is not, should be added.

3. To strengthen the practical feasibility of the proposed approach, the authors are encouraged to provide some experiments beyond the toy environments (e.g., MuJoCo or at minimum continuous versions of the tested tasks). The current environments are all small and discrete, which is where Array-RQMC is most naturally applicable. The scope of the empirical claim that RQMC "consistently matches or improves" baselines should be tested in settings where it is not that obvious.

4. It would be helpful to include the Arnold et al. (2022) RQMC baseline as a direct comparison. The paper's key claim of novelty is integrating RQMC into NAC, and hence a comparison against RQMC applied to a standard policy gradient can sharpen this claim considerably.

5. Several theoretical results in Section 3 are already well-established in the literature and hence not new. Specifically,
Theorem 2 is a classic result regarding the Taylor expansion of KL divergence.
Theorem 3 is also a known result in the trust region optimization literature, e.g., TRPO (Schulman et al., 2015).
Currently, these well-established results are not properly cited and take too much space in the main text.
Moreover, for other parts of the preliminaries, such as Theorems 4-7, they can be made more concise as these are not the main contributions of this work. In its current form, the main text seems to use too much space on the preliminaries.

6. There is no wall-clock time comparison. This can be critical as Array-RQMC incurs sorting overhead that may offset variance gains in practice.

---

> ### Author Response · Authors · 2026-05-14
> **Response**
>
> We really thank the reviewer for the constructive and insightful comments. We have now integrated all the requested changes
>
> 1. We agree with the reviewer. We have added a dedicated discussion (Section 4: colored RED)  clarifying the setting in which the Array-RQMC rate translates into practical improvement over standard Monte Carlo NAC. In particular, the advantage is expected when the stochastic error in the NAC compatible-feature linear system is significantly attributed to trajectory-generation noise, the simulator admits a reasonably smooth low-dimensional transport from uniform random variables to next states, and the number of parallel trajectories is large enough for the low-discrepancy structure to compensate for sorting overhead. We also now explicitly discuss the complementary regimes where the improvement may be limited, including high effective transition dimension, discontinuous simulators, dominant irreducible environmental noise, or cases where the cost of sorting dominates the variance reduction.
>
> 2. We have now specified the generative model in detail in Section 4. The generative-model  is now stated as a simulator-access condition rather than as a universally available property. We explain that exact inverse-transform access is natural in finite-state simulators and in reparameterizable environments where the transition kernel can be represented through uniform driving noise. We also added a discussion of what happens when such access is unavailable.
>
> 3. We have added continuous-control ( stochastic MuJoCo ) experiment to test how the method behaves when the state and action spaces are more complex and when the low-discrepancy advantage must compete with higher-dimensional dynamics and computational overhead.
>
> 4. We have incorporated this suggestion. The revised experiments now include an RQMC-policy-gradient baseline (Arnold et al. (2022)).
>
> 5. The existing results in Amari is on statistical manifold and Kakade's result was not detailed as ours. And Schulman's were more generalized. That's why we kept those results. Now we have removed the results and kept a detailed discussion of the same (After Lemma 1 in RED color)
>
> 6. We have now added a wall-clock comparison to account for the additional cost introduced by Array-RQMC, including sorting and low-discrepancy point generation.

---

### Decision · Action_Editor_F5Gw · 2026-06-23

**Recommendation:** Accept with minor revision

**Additional Comments:**

The reviewers are overall happy with the submission. One reviewer requested a clearer positioning against existing NAC convergence results and prior RQMC policy-gradient work, a more explicit discussion of when the inverse-transform sampling assumption is practically feasible, and experiments beyond small discrete environments. While the author's response has addressed most of the above concerns, I kindly request that they go over the reviewer's concerns once more and address any remaining concerns.

**Audience:**

Yes

**Audience Explanation:**

The scope of the paper, natural gradient methods in policy optimization, is certainly interesting to TMLR audience.

**Claims And Evidence:**

Yes

**Claims Explanation:**

The theoretical contribution is supported by proof. The authors also provide experiments for empirical validation.